# Debiasing Conditional Stochastic Optimization

**Lie He**[*]
EPFL
lie.he@epfl.ch

Shiva Prasad Kasiviswanathan
Amazon
kasivisw@gmail.com

## Abstract

In this paper, we study the conditional stochastic optimization (CSO) problem which covers a variety of applications including portfolio selection, reinforcement learning, robust learning, causal inference, etc. The sample-averaged gradient of the CSO objective is biased due to its nested structure, and therefore requires a high sample complexity for convergence. We introduce a general *stochastic extrapolation* technique that effectively reduces the bias. We show that for non-convex smooth objectives, combining this extrapolation with variance reduction techniques can achieve a significantly better sample complexity than the existing bounds. Additionally, we develop new algorithms for the finite-sum variant of the CSO problem that also significantly improve upon existing results. Finally, we believe that our debiasing technique has the potential to be a useful tool for addressing similar challenges in other stochastic optimization problems.

## 1 Introduction

In this paper, we investigate the *conditional stochastic optimization* (CSO) problem as presented by Hu et al. [16], which is formulated as follows:

$$\min_{\boldsymbol{x}\in\mathbb{R}^d} F(\boldsymbol{x}) = \mathbb{E}_\xi[f_\xi(\mathbb{E}_{\eta|\xi}[g_\eta(\boldsymbol{x};\xi)])], \tag{CSO}$$

where $\xi$ and $\eta$ represent two random variables, with $\eta$ conditioned on $\xi$. The $f_\xi : \mathbb{R}^p \to \mathbb{R}$ and $g_\eta : \mathbb{R}^d \to \mathbb{R}^p$ denote a stochastic function and a mapping respectively. The inner expectation is calculated with respect to the conditional distribution of $\eta|\xi$. In line with the established CSO framework [16, 15], throughout this paper, we assume access to samples from the distribution $\mathbb{P}(\xi)$ and the conditional distribution $\mathbb{P}(\eta|\xi)$.

Many machine learning tasks can be formulated as a CSO problem, such as policy evaluation and control in reinforcement learning [6, 24], and linearly-solvable Markov decision process [5]. Other examples of the CSO problem include instrumental variable regression [23] and invariant learning [16]. Moreover, the widely-used Model-Agnostic Meta-Learning (MAML) framework, which seeks to determine a meta-initialization parameter using metadata for related learning tasks that are trained through gradient-based algorithms, is another example of a CSO problem. In this context, tasks $\xi$ are drawn randomly, followed by the drawing of samples $\eta|\xi$ from the specified task [11]. It is noteworthy that the standard stochastic optimization problem $\min_{\boldsymbol{x}} \mathbb{E}_\xi[f_\xi(\boldsymbol{x})]$ represents a degenerate case of the CSO problem, achieved by setting $g_\eta$ as an identity function.

In numerous prevalent CSO problems, such as first-order MAML (FO-MAML) [11], the outer random variable $\xi$ only takes value in a finite set (say in $\{1, \dots, n\}$). These problems can be reformulated to have a finite-sum structure in the outer loop and referred to as *Finite-sum Coupled Compositional Optimization (FCCO)* problem in [33, 19]. In this paper, we also study this problem, formulated as:

$$\min_{\boldsymbol{x}\in\mathbb{R}^d} F_n(\boldsymbol{x}) = \tfrac{1}{n}\sum_{i=1}^n f_i(\mathbb{E}_{\eta|i}[g_\eta(\boldsymbol{x};i)]). \tag{FCCO}$$

The FCCO problem also has broad applications in machine learning for optimizing average precision, listwise ranking losses, neighborhood component analysis, deep survival analysis, deep latent variable models [33, 19].

---

[*]Work initiated during internship at Amazon.

37th Conference on Neural Information Processing Systems (NeurIPS 2023).

Although the CSO and FCCO problems are widespread, they present challenges for optimization algorithms. Based on the special composition structure of CSO, using chain rule, under mild conditions, the full gradient of CSO is given by

$$\nabla F(\boldsymbol{x}) = \mathbb{E}_\xi \left[ \left( \mathbb{E}_{\eta|\xi}[\nabla g_\eta(\boldsymbol{x}; \xi)] \right)^\top \nabla f_\xi(\mathbb{E}_{\eta|\xi}[g_\eta(\boldsymbol{x}; \xi)]) \right].$$

Constructing an unbiased stochastic estimator for the gradient is generally computationally expensive (and even impossible). A straightforward estimation of $\nabla F(\boldsymbol{x})$ is to estimate $\mathbb{E}_\xi$ with 1 sample of $\xi$, estimate $\mathbb{E}_{\eta|\xi}[g_\eta(\cdot)]$ with a set $H_\xi$ of $m$ independent and identically distributed (i.i.d.) samples drawn from the conditional distribution $\mathbb{P}(\eta|\xi)$, and $\mathbb{E}_{\eta|\xi}[\nabla g_\eta(\cdot)]$ with a different set $\tilde{H}_\xi$ of $m$ i.i.d. samples drawn from the same conditional distribution, i.e.,

$$\nabla \hat{F}_m(\boldsymbol{x}) := \left( \tfrac{1}{m} \textstyle\sum_{\tilde\eta \in \tilde{H}_\xi} \nabla g_{\tilde\eta}(\boldsymbol{x}; \xi) \right)^\top \nabla f_\xi(\tfrac{1}{m} \textstyle\sum_{\eta \in H_\xi} g_\eta(\boldsymbol{x}; \xi)). \tag{1}$$

Note that $\nabla \hat{F}_m(\boldsymbol{x})$ consists of two terms. The first term, $(1/m)\sum_{\tilde\eta \in \tilde{H}_\xi} \nabla g_{\tilde\eta}(\mathbf{x}; \xi)$, is an unbiased estimate of $\mathbb{E}_{\eta|\xi}[\nabla g_\eta(\boldsymbol{x}; \xi)]$. However, the second term is generally biased, i.e.,

$$\mathbb{E}_{\eta|\xi}[\nabla f_\xi(\tfrac{1}{m} \textstyle\sum_{\eta \in H_\xi} g_\eta(\boldsymbol{x}; \xi))] \neq \nabla f_\xi(\mathbb{E}_{\eta|\xi}[g_\eta(\boldsymbol{x}; \xi)]).$$

Consequently, $\nabla \hat{F}_m(\boldsymbol{x})$ is a biased estimator of $\nabla F(\boldsymbol{x})$. To reach the $\varepsilon$-stationary point of $F(\boldsymbol{x})$ (Definition 1), the bias has to be sufficiently small.

Optimization with biased gradients converges only to a neighborhood of the stationary point. While the bias diminishes with increasing batch size, it also introduces additional sample complexity. For nonconvex objectives, Biased Stochastic Gradient Descent (BSGD) requires a total sample complexity of $\mathcal{O}(\varepsilon^{-6})$ to reach an $\varepsilon$-stationary point [16]. This contrasts with standard stochastic optimization, where sample-averaged gradients are unbiased with a sample complexity of $\mathcal{O}(\varepsilon^{-4})$ [12, 3]. This discrepancy has spurred a multitude of proposals aimed at reducing the sample complexities of both CSO and FCCO problems. Hu et al. [16] introduced Biased SpiderBoost (BSpiderBoost), which, based on the variance reduction technique SpiderBoost from Wang et al. [38], reduces the variance of $\xi$ to achieve a sample complexity of $\mathcal{O}(\varepsilon^{-5})$ for the CSO problem. Hu et al. [17] proposed multi-level Monte Carlo (MLMC) gradient methods V-MLMC and RT-MLMC to further enhance the sample complexity to $\mathcal{O}(\varepsilon^{-4})$. The SOX [33] and MSVR-V2 [19] algorithms concentrated on the FCCO problem and improved the sample complexity to $\mathcal{O}(n\varepsilon^{-4})$ and $\mathcal{O}(n\varepsilon^{-3})$, respectively.

**Our Contributions.** In this paper, we improve the sample complexities for both the CSO and FCCO problems (see Table 1). To facilitate a clear and concise presentation, we will suppress the dependence on specific problem parameters throughout the ensuing discussion.

**(a)** Our main technical tool in this paper is an extrapolation-based scheme that mitigates. bias in gradient estimations. Considering a suitably differentiable function $q(\cdot)$ and a random variable $\delta \sim \mathcal{D}$, we show that we can approximate the value of $q(\mathbb{E}[\delta])$ via extrapolation from a limited number of evaluations of $q(\delta)$, while maintaining a minimal bias. In the context of CSO and FCCO problems, this scheme is used in gradient estimation, where the function $q$ corresponds to $\nabla f_\xi$ and the random variable $\delta$ corresponds to $g_\eta$.

**(b)** For the CSO problem, we present novel algorithms that integrate the above extrapolation-based scheme with BSGD and BSpiderBoost algorithms of Hu et al. [16]. Our algorithms, referred to as E-BSGD and E-BSpiderBoost, achieve a sample complexity of $\mathcal{O}(\varepsilon^{-4.5})$ and $\mathcal{O}(\varepsilon^{-3.5})$ respectively, in order to attain an $\varepsilon$-stationary point for nonconvex smooth objectives. Notably, the sample complexity of E-BSpiderBoost improves the best-known sample complexity of $\mathcal{O}(n\varepsilon^{-4})$ for the CSO problem from Hu et al. [17].

**(c)** For the FCCO problem[2] we propose a new algorithm that again combines the extrapolation-based scheme with a multi-level variance reduction applied to both inner and outer parts of the problem. Our algorithm, referred to as E-NestedVR, achieves a sample complexity of $\mathcal{O}(n\varepsilon^{-3})$ if $n \leq \varepsilon^{-2/3}$ and $\mathcal{O}(\max\{\sqrt{n}\varepsilon^{-2.5}, \varepsilon^{-4}/\sqrt{n}\})$ if $n > \varepsilon^{-2/3}$ for nonconvex smooth objectives and second-order extrapolation scheme. Our bound is never worse than the $\mathcal{O}(n\varepsilon^{-3})$ bound of MSVR-V2 algorithm of Jiang et al. [19] and is in fact better if $n = \Omega(\varepsilon^{-2/3})$. As an illustration, when $n = \Theta(\varepsilon^{-1.5})$, our bound of $\mathcal{O}(\varepsilon^{-3.25})$ is significantly better than the MSVR-V2 bound of $\mathcal{O}(\varepsilon^{-4.5})$.

---

[2] For the FCCO problem we focus on $n = \mathcal{O}(\varepsilon^{-2})$ case, for $n = \Omega(\varepsilon^{-2})$ we can just treat the FCCO problem as a CSO problem and get an $\mathcal{O}(\varepsilon^{-3.5})$ sample complexity bound via our E-BSpiderBoost algorithm.

| Problem | Old Bounds | | Our Bounds | |
| --- | --- | --- | --- | --- |
| | Algorithm | Bound | Algorithm | Bound |
| CSO | BSGD [16] | $\mathcal{O}(\varepsilon^{-6})$ | E-BSGD | $\mathcal{O}(\varepsilon^{-4.5})$ |
| CSO | BSpiderBoost [16] | $\mathcal{O}(\varepsilon^{-5})$ | E-BSpiderBoost | $\mathcal{O}(\varepsilon^{-3.5})$ |
| CSO | RT-MLMC [17] | $\mathcal{O}(\varepsilon^{-4})$ | | |
| FCCO | MSVR-V2 [19] | $\mathcal{O}(n\varepsilon^{-3})$ | E-NestedVR | $\begin{cases} \mathcal{O}(n\varepsilon^{-3}) & \text{if } n \leq \varepsilon^{-2/3} \\ \mathcal{O}(\max\{\frac{\sqrt{n}}{\varepsilon^{2.5}}, \frac{1}{\sqrt{n}\varepsilon^4}\}), & \text{if } n > \varepsilon^{-2/3} \end{cases}$ |

Table 1: Sample complexities needed to reach $\varepsilon$-stationary point for FCCO and CSO problems with nonconvex smooth objectives. Assumptions are comparable, but our results require an additional mild regularity on $f_\xi$ and $g_\eta$. For FCCO also see Footnote 2. Note that $\Omega(\varepsilon^{-3})$ is a sample complexity lower bound for standard stochastic nonconvex optimization [3], and hence, also for the problems considered in this paper.

In terms of proof techniques, our approach diverges from conventional analyses for the CSO and FCCO problems in that we focus on explicitly bounding the bias and variance terms of the gradient estimator to establish the convergence guarantee. Compared to previous results, our improvements do require an additional mild regularity assumption on $f_\xi$ and $g_\eta$ mainly that $\nabla f_\xi$ is 4th order differentiable. Firstly, as we discuss in Remark 2 most common instantiations of CSO/FCCO framework such as: 1) invariant logistic regression [16], 2) instrumental variable regression [23], 3) first-order MAML for sine-wave few shot regression [11] and other problems, 4) deep average precision maximization [26, 34], tend to satisfy this assumption. Secondly, we highlight that the bounds derived from previous studies do not improve when incorporating this additional regularity assumption. Thirdly, $\Omega(\varepsilon^{-3})$ remains the lower bound for stochastic optimization even under the arbitrary smoothness constraint [2], demonstrating that our improvement is non-trivial. Our results show that, this regularity assumption, which seems to practically valid, can be exploited through a novel extrapolation-based bias reduction technique to provide substantial improvements in sample complexity.[3]

We defer some additional related work to Appendix B and conclude with some preliminaries.

**Notation.** Vectors are denoted by boldface letters. For a vector $\boldsymbol{x}$, $\|\boldsymbol{x}\|_2$ denotes its $\ell_2$-norm. A function with $k$ continuous derivatives is called a $\mathcal{C}^k$ function. We use $a \lesssim b$ to denote that $a \leq Cb$ for some constant $C > 0$. We consider expectation over various randomness: $\mathbb{E}_\xi[\cdot]$ denotes expectation over the random variable $\xi$, $\mathbb{E}_{\eta|\xi}[\cdot]$ denotes expectation over the conditional distribution of $\eta|\xi$. Unless otherwise specified, for a random variable $X$, $\mathbb{E}[X]$ denotes expectation over the randomness in $X$. We focus on nonconvex objectives in this paper and use the following standard convergence criterion for nonconvex optimization [18].

**Definition 1** ($\varepsilon$-stationary point) *For a differentiable function $F(\cdot)$, we say that $\boldsymbol{x}$ is a first-order $\varepsilon$-stationary point if $\|\nabla F(\boldsymbol{x})\|^2 \leq \varepsilon^2$.*

For notational convenience, in the rest of this paper, we omit the dependence on $\xi$ (or $i$ in the FCCO context) in the function $g$ and use $g_\eta(\boldsymbol{x})$ to represent $g_\eta(\boldsymbol{x}; \xi)$.

## 2 Stochastic Extrapolation as a Tool for Bias Correction

In this section, we present an approach for tackling the bias problem as appears in optimization procedures such as BSGD, BSpiderBoost, etc. Importantly, our approach addresses a general problem appearing in optimization settings and could be of independent interest. All missing details from this section are presented in Appendix C.

For ease of presentation, we start by considering the 1-dimensional case and assume a function $q : \mathbb{R} \rightarrow \mathbb{R}$, a constant $s \in \mathbb{R}$. Let $\delta$ be a random variable drawn from an arbitrary distribution $\mathcal{D}$ over $\mathbb{R}$. In Sections 3 and 4, we apply these ideas to the CSO and FCCO problems where the random variable $\delta$ is played by $g_\eta(\cdot)$ and function $q$ is played by $\nabla f_\xi$. Informally stated, our goal in this section will be to

> Efficiently approximate $q(s + \mathbb{E}[\delta])$ with few evaluations of $\{q(s + \delta)\}_{\delta \sim \mathcal{D}}$.

---

[3]Higher-order smoothness conditions have also been exploited in standard stochastic optimization for performance gains [4].

An interesting case is when $s = 0$, where we are approximating $q(\mathbb{E}[\delta])$ with evaluations of $\{q(\delta)\}_{\delta \sim \mathcal{D}}$. Now, if $q$ is an affine function, then $q(s + \mathbb{E}[\delta]) = \mathbb{E}[q(s + \delta)]$. However, the equality does not hold true for general $q$, and there exists a bias, i.e., $|q(s + \mathbb{E}[\delta]) - \mathbb{E}[q(s + \delta)]| > 0$. In this section, we introduce a stochastic *extrapolation*-based method, where we use an affine combination of biased stochastic estimates, to achieve better approximation.

Suppose $q \in \mathcal{C}^{2k}$ is a continuous differentiable up to $2k$-th derivative and let $h = \mathbb{E}[\delta]$. We expand $q(s + \delta)$, the most straightforward approximation of $q(s + \mathbb{E}[\delta])$, using Taylor series at $s + h$, and take expectation,

$$
\begin{aligned}
\mathbb{E}[q(s + \delta)] =\; & q(s + h) + q'(s + h)\,\mathbb{E}[\delta - h] + \tfrac{q''(s+h)}{2}\,\mathbb{E}[(\delta - h)^2] + \tfrac{q^{(3)}(s+h)}{6}\,\mathbb{E}[(\delta - h)^3] \\
& + \ldots + \tfrac{q^{(2k-1)}(s+h)}{(2k-1)!}\,\mathbb{E}[(\delta - h)^{(2k-1)}] + \tfrac{1}{(2k)!}\,\mathbb{E}[q^{(2k)}(\phi_\delta)(\delta - h)^{2k}],
\end{aligned} \tag{2}
$$

where $\phi_\delta$ between $s + \delta$ and $s + h$. While $\mathbb{E}[q(s + \delta)]$ matches $q(s + h)$ in the first 2 terms, the third term is no longer zero. The approximation error (bias) is

$$
|\mathbb{E}[q(s + \delta)] - q(s + h)| = |\tfrac{q''(s+h)}{2}\,\mathbb{E}[(\delta - h)^2] + \ldots + \tfrac{1}{(2k)!}\,\mathbb{E}[q^{(2k)}(\phi_\delta)(\delta - h)^{2k}]|.
$$

In order to analyze the upper bound, we make the following assumption on $\mathcal{D}$ and $q$.

**Assumption 1 (Bounded moments)** *For all $\delta \sim \mathcal{D}$ has bounded higher-order moments:* $\sigma_l := |\mathbb{E}[(\delta - \mathbb{E}[\delta])^l]| < \infty$ *for $l = 2, 3, \ldots 2k$.*

**Assumption 2 (Bounded derivatives)** *The $q \in \mathcal{C}^{2k}$ and has bounded derivatives, i.e., $a_l := \sup_{s \in dom(q)} |q^{(l)}(s)| < \infty$ for $l = 1, 2, \ldots, 2k$.*

In addition, we consider a sample averaged distribution $\mathcal{D}_m$ derived from $\mathcal{D}$ as follows.

**Definition 2** *Given a distribution $\mathcal{D}$ satisfying Assumption 1 and $m \in \mathbb{N}^+$, we define the distribution $\mathcal{D}_m$ that outputs $\delta$ where $\delta = \frac{1}{m} \sum_{i=1}^m \delta_i$ with $\delta_i \overset{i.i.d.}{\sim} \mathcal{D}$.*

The moments of such distribution $\mathcal{D}_m$ decrease with batch size $m$ as $k \geq 2$, $|\mathbb{E}[(\delta - \mathbb{E}[\delta])^k]| = \mathcal{O}(m^{-\lceil k/2 \rceil})$ (see Lemma 1). Our desiderata would be to construct a scheme that uses some samples from the distribution $\mathcal{D}_m$ to construct an approximation of $q(s + \mathbb{E}[\delta])$ that satisfies the following requirement.

**Definition 3 ($k$th-order Extrapolation Operator)** *Given a function $q : \mathbb{R} \to \mathbb{R}$ satisfying Assumption 2 and distribution $\mathcal{D}_m$ satisfying Assumption 1, we define a $k$th-order extrapolation operator $\mathcal{T}_{\mathcal{D}_m}^{(k)}$ as an operator from $\mathcal{C}^{2k} \to \mathcal{C}^{2k}$ that given $N = N(k)$ i.i.d. samples $\delta_1, \ldots, \delta_N$ from $\mathcal{D}_m$ satisfies $\forall s \in \mathbb{R}$: $|\mathbb{E}[\mathcal{T}_m^{(k)} q(s)] - q(s + \mathbb{E}[\delta])| = \mathcal{O}(m^{-k})$.*

We now propose a sequence of operators $\mathcal{L}_{\mathcal{D}_m}^{(1)}, \mathcal{L}_{\mathcal{D}_m}^{(2)}, \mathcal{L}_{\mathcal{D}_m}^{(3)}, \ldots$ that satisfy the above definition. The $\mathcal{L}_{\mathcal{D}_m}^{(k)} q(s)$ is designed to ensure its Taylor expansion at $s + h$ has a form of $q(s + h) + \mathcal{O}(\mathbb{E}[(\delta - h)^{2k}])$. The remainder $\mathcal{O}(\mathbb{E}[(\delta - h)^{2k}])$ is bounded by $\mathcal{O}(m^{-k})$ due to Lemma 1.

**A First-order Extrapolation Operator.** We define the simplest operator

$$
\mathcal{L}_{\mathcal{D}_m}^{(1)} q : s \mapsto [q(s + \delta)] \qquad \text{where } \delta \overset{i.i.d.}{\sim} \mathcal{D}_m.
$$

In Proposition 2 (Appendix C), we show that $\mathcal{L}_{\mathcal{D}_m}^{(1)}$ is a first-order extrapolation operator.[4]

**A Second-order Extrapolation Operator.** We define the following linear operator $\mathcal{L}_{\mathcal{D}_m}^{(2)}$ which transforms $q \in \mathcal{C}^4$ into $\mathcal{L}_{\mathcal{D}_m}^{(2)} q$ which has lesser bias (but similar variance, as shown later).

**Definition 4 ($\mathcal{L}_{\mathcal{D}_m}^{(2)}$ Operator)** *Given $\mathcal{D}_m$ and $q$, define the following operator,*

$$
\mathcal{L}_{\mathcal{D}_m}^{(2)} q : s \mapsto \left[ 2 \cdot q(s + \tfrac{\delta_1 + \delta_2}{2}) - \tfrac{q(s + \delta_1) + q(s + \delta_2)}{2} \right] \qquad \text{where } \delta_1, \delta_2 \overset{i.i.d.}{\sim} \mathcal{D}_m.
$$

---

[4] Note that if the function $q$ is only $L_q$-Lipschitz continuous, then $|\mathbb{E}[q(s + \delta)] - q(s + \mathbb{E}[\delta])| \leq \sqrt{L_q^2 \, \mathbb{E}[|\delta - \mathbb{E}[\delta]|]^2} \leq \tfrac{L_q \sqrt{\sigma_2}}{m^{1/2}}$. Therefore, in this case, $q(s + \delta)$ does not satisfy the first-order guarantee.

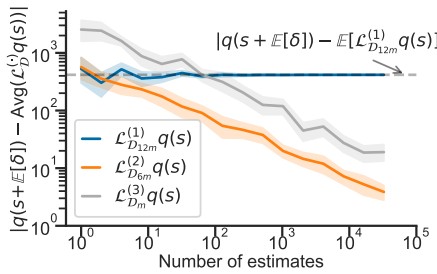 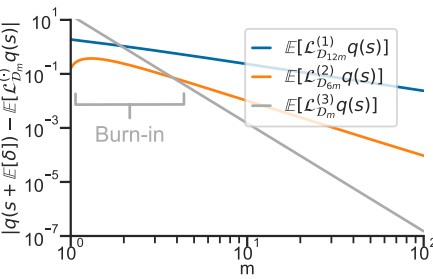

(a) $q(s) = s^2/2$, $\delta \sim \mathcal{N}(10, 100)$, $m = 1$.      (b) $q(s) = s^4$, $p(\delta) = \delta/2$ where $\delta \in [0, 2]$.

Figure 1: The Fig. 1a investigates the estimation errors of $\mathcal{L}^{(\cdot)}q(s)$ with their number of observations. The Fig. 1b compares the biases of $\mathbb{E}[\mathcal{L}^{(\cdot)}q(s)]$ with increasing inner batch size $m$.

Note that $\frac{\delta_1 + \delta_2}{2}$ is same as sampling from $\mathcal{D}_{2m}$. The absolute difference in the Taylor expansion of $\mathcal{L}^{(2)}_{\mathcal{D}_m} q$ at $s + h$ differs from $q(s + h)$ as,

$$\mathcal{O}\left(\left| \mathbb{E}\left[ 2(\tfrac{\delta_1 + \delta_2}{2} - h)^3 - \tfrac{1}{2}((\delta_1 - h)^3 + (\delta_2 - h)^3) \right] \right|\right) = \mathcal{O}(|(\mathbb{E}[(\delta - h)^3]|) \text{ for } \delta \overset{i.i.d.}{\sim} \mathcal{D}_m. \quad (3)$$

The bias error of this scheme can be bounded through the following proposition.

**Proposition 1 (Second-order Guarantee)** *Assume that distribution $\mathcal{D}_m$ and $q(\cdot)$ satisfies Assumption 1 and 2 respectively with $k = 2$. Then, for all $s \in \mathbb{R}$, $\left| \mathbb{E}\left[ \mathcal{L}^{(2)}_{\mathcal{D}_m} q(s) \right] - q(s + \mathbb{E}[\delta]) \right| \leq \frac{4a_3 \sigma_3 + 9a_4 \sigma_2^2}{48m^2} + \frac{5a_4}{96} \frac{\sigma_4 - 3\sigma_2^2}{m^3}$.*

**Remark 1** *While extrapolation is motivated by Taylor expansion which requires smoothness, higher order derivatives are not explicitly computed. Appendix F.3 empirically shows that applying extrapolation to non-smooth functions achieves similar bias correction. Relaxing the smoothness conditions is a direction for future work.*

The above proposition shows that $\mathcal{L}^{(2)}_{\mathcal{D}_m}$ is in fact a second-order extrapolation operator with $k = 2$ under Definition 3. We will use this operator when we consider the CSO and FCCO problems later. Now, focusing on variance, we can relate the variance of $\mathcal{L}^{(2)}_{\mathcal{D}_m} q(s)$ in terms of the variance of $q(s + \delta)$. In particular, a consequence of Lemma 2 is that

$$\mathbb{E}\left[ \left( \mathcal{L}^{(2)}_{\mathcal{D}_m} q(s) - \mathbb{E}[\mathcal{L}^{(2)}_{\mathcal{D}_m} q(s)] \right)^2 \right] = \mathcal{O}(\mathbb{E}[(q(s + \delta) - \mathbb{E}[q(s + \delta)])^2]).$$

**Extension of $\mathcal{L}^{(2)}_{\mathcal{D}_m}$ to Higher-dimensional Case.** If $q : \mathbb{R}^p \to \mathbb{R}^\ell$ is a vector-valued function, then there is a straightforward extension of Definition 4. Now, for distribution $\mathcal{D}$ over $\mathbb{R}^p$ and corresponding sampled averaged distribution $\mathcal{D}_m$, and $\boldsymbol{s} \in \mathbb{R}^p$

$$\mathcal{L}^{(2)}_{\mathcal{D}_m} q : \boldsymbol{s} \mapsto \left[ 2 \cdot q\left(\boldsymbol{s} + \tfrac{\boldsymbol{\delta}_1 + \boldsymbol{\delta}_2}{2}\right) - \tfrac{q(\boldsymbol{s} + \boldsymbol{\delta}_1) + q(\boldsymbol{s} + \boldsymbol{\delta}_2)}{2} \right] \qquad \text{where } \boldsymbol{\delta}_1, \boldsymbol{\delta}_2 \overset{i.i.d.}{\sim} \mathcal{D}_m. \quad (4)$$

**Higher-order Extrapolation Operators.** The idea behind the construction of $\mathcal{L}^{(2)}_{\mathcal{D}_m}$ can be generalized to higher $k$'s. For example, in Proposition 3, we construct a third-order extrapolation operator $\mathcal{L}^{(3)}_{\mathcal{D}_m}$ through higher degree Taylor series approximation

$$\mathcal{L}^{(3)}_{\mathcal{D}_m} q : s \mapsto \left(-\tfrac{1}{36}\mathcal{L}^{(2)}_{\mathcal{D}_m} + \tfrac{5}{9}\mathcal{L}^{(2)}_{\mathcal{D}_{2m}} - \tfrac{3}{4}\mathcal{L}^{(2)}_{\mathcal{D}_{3m}} - \tfrac{16}{9}\mathcal{L}^{(2)}_{\mathcal{D}_{4m}} + 3\mathcal{L}^{(2)}_{\mathcal{D}_{6m}}\right)q(s).$$

While this idea of expressing the $k$-th order operator as an affine combination of lower-order operators works for every $k$, explicit constructions soon become tedious.

In Fig. 1, we empirically demonstrate the effectiveness of extrapolation in stochastic estimation. [5] In Fig. 1a, we choose $q(s) = s^2/2$, $\delta \sim \mathcal{N}(10, 100)$. For both $\mathcal{L}^{(2)}_{\mathcal{D}_{6m}} q(s)$ and $\mathcal{L}^{(3)}_{\mathcal{D}_m} q(s)$, their estimation errors converge to 0 with increasing number of estimates. This coincides with Proposition 1 as $a_3 = 0$

---

[5] We use $\mathcal{L}^{(1)}_{\mathcal{D}_{12m}}$, $\mathcal{L}^{(2)}_{\mathcal{D}_{6m}}$, $\mathcal{L}^{(3)}_{\mathcal{D}_m}$ to ensure that each estimate uses same amount of samples ($12m$).

and $a_4 = 0$ for quadratic $q$. In contrast, biased first order method only converges to a neighborhood. In Fig. 1b, we consider $q(s) = s^4$ and $p(\delta) = \delta/2$ where $\delta \in [0, 2]$. All three methods are biased and their biases decrease with $m$, i.e. $\mathcal{O}(m^{-k})$ for $k$th order method. Depending on the constants (e.g. $a_i$, $\sigma_i$), a higher-order extrapolation method may need decently large $m$ (burn-in phase) to outperform lower-order methods.

## 3 Applying Stochastic Extrapolation in the CSO Problem

In this section, we apply the extrapolation-based scheme from the previous section to reduce the bias in the CSO problem. We focus on variants of BSGD and their accelerated version BSpiderBoost based on our second-order approximation operator (Definition 4). Let $H_\xi$, $\tilde{H}_\xi$, and $H'_\xi$ indicate different sets, each of which contains $m$ i.i.d. random variables/samples drawn from the conditional distribution $\mathbb{P}(\eta|\xi)$. Remember that, as mentioned earlier, we use $g_\eta(\boldsymbol{x})$ to represent $g_\eta(\boldsymbol{x}; \xi)$.

**Extrapolated BSGD.** At time $t$, BSGD constructs a biased estimator of $\nabla F(\boldsymbol{x}^t)$ using one sample $\xi$ and $2m$ i.i.d. samples from the conditional distribution as in (1)

$$G_{\text{BSGD}}^{t+1} = \left(\tfrac{1}{m}\sum_{\tilde{\eta}\in\tilde{H}_\xi}\nabla g_{\tilde{\eta}}(\boldsymbol{x}^t)\right)^\top \nabla f_\xi\left(\tfrac{1}{m}\sum_{\eta\in H_\xi}g_\eta(\boldsymbol{x}^t)\right). \tag{5}$$

To reduce this bias, we apply the second-order extrapolation operator from (4). At time $t$, we define $\mathcal{D}_{\boldsymbol{g},\xi}^{t+1}$ to be the distribution of the random variable $\tfrac{1}{m}\sum_{\eta\in H_\xi}g_\eta(\boldsymbol{x}^t)$. Then we apply $\mathcal{L}_{\mathcal{D}_{\boldsymbol{g},\xi}^{t+1}}^{(2)}$ by setting $q$ to $\nabla f_\xi$ and $\boldsymbol{s} = 0$, i.e.

$$\mathcal{L}_{\mathcal{D}_{\boldsymbol{g},\xi}^{t+1}}^{(2)}\nabla f_\xi(0) := 2\nabla f_\xi\left(\tfrac{1}{2m}\sum_{\eta\in H_\xi}g_\eta(\boldsymbol{x}^t) + \tfrac{1}{2m}\sum_{\eta'\in H'_\xi}g_{\eta'}(\boldsymbol{x}^t)\right)$$
$$- \tfrac{1}{2}\left(\nabla f_\xi(\tfrac{1}{m}\sum_{\eta\in H_\xi}g_\eta(\boldsymbol{x}^t)) + \nabla f_\xi(\tfrac{1}{m}\sum_{\eta'\in H'_\xi}g_{\eta'}(\boldsymbol{x}^t))\right), \tag{6}$$

where $\tfrac{1}{m}\sum_{\eta\in H_\xi}g_\eta(\boldsymbol{x}^t)$ and $\tfrac{1}{m}\sum_{\eta'\in H'_\xi}g_{\eta'}(\boldsymbol{x}^t)$ are i.i.d. drawn from $\mathcal{D}_{\boldsymbol{g},\xi}^{t+1}$. In Algorithm 2 (Appendix A), we present our extrapolated BSGD (E-BSGD) scheme, where we replace $\nabla f_\xi(\tfrac{1}{m}\sum_{\eta\in H_\xi}g_\eta(\boldsymbol{x}^t))$ in (5) by $\mathcal{L}_{\mathcal{D}_{\boldsymbol{g},\xi}^{t+1}}^{(2)}\nabla f_\xi(0)$ resulting in this following gradient estimate:

$$G_{\text{E-BSGD}}^{t+1} = \left(\tfrac{1}{m}\sum_{\tilde{\eta}\in\tilde{H}_\xi}\nabla g_{\tilde{\eta}}(\boldsymbol{x}^t)\right)^\top \mathcal{L}_{\mathcal{D}_{\boldsymbol{g},\xi}^{t+1}}^{(2)}\nabla f_\xi(0). \tag{7}$$

**Extrapolated BSpiderBoost.** BSpiderBoost, proposed by Hu et al. [16], uses the variance reduction methods for nonconvex smooth stochastic optimization developed by Fang et al. [10], Wang et al. [38]. BSpiderBoost builds upon BSGD and has two kinds of updates: a large batch and a small batch update. In each step, it decides which update to apply based on a random coin. With probability $p_{\text{out}}$, it selects a large batch update with $B_1$ outer samples of $\xi$. With remaining probability $1 - p_{\text{out}}$, it selects a small batch update where the gradient estimator will be updated with gradient information in the current iteration generated with $B_2$ outer samples of $\xi$ and the information from the last iteration. Formally, it constructs a gradient estimate as follows,

$$G_{\text{BSB}}^{t+1} = \begin{cases} G_{\text{BSB}}^t + \tfrac{1}{B_2}\sum_{\xi\in\mathcal{B}_2,|\mathcal{B}_2|=B_2}(G_{\text{BSGD}}^{t+1} - G_{\text{BSGD}}^t) & \text{with prob. } 1 - p_{\text{out}} \\ \tfrac{1}{B_1}\sum_{\xi\in\mathcal{B}_1,|\mathcal{B}_1|=B_1}G_{\text{BSGD}}^{t+1} & \text{with prob. } p_{\text{out}}. \end{cases} \tag{8}$$

We propose our extrapolated BSpiderBoost scheme (formally defined in Algorithm 3, Appendix A) by replacing the BSGD gradient estimates in (8) with E-BSGD.

$$G_{\text{E-BSB}}^{t+1} = \begin{cases} G_{\text{E-BSB}}^t + \tfrac{1}{B_2}\sum_{\xi\in\mathcal{B}_2,|\mathcal{B}_2|=B_2}(G_{\text{E-BSGD}}^{t+1} - G_{\text{E-BSGD}}^t) & \text{with prob. } 1 - p_{\text{out}} \\ \tfrac{1}{B_1}\sum_{\xi\in\mathcal{B}_1,|\mathcal{B}_1|=B_1}G_{\text{E-BSGD}}^{t+1} & \text{with prob. } p_{\text{out}}. \end{cases} \tag{9}$$

**Sample Complexity Analyses of E-BSGD and E-BSpiderBoost.** We adopt the standard assumptions used in the literature [27, 35, 33, 41]. All proofs are deferred to Appendix D.

**Assumption 3 (Lower bound)** *$F$ is lower bounded by $F^\star$.*

**Assumption 4 (Bounded variance)** *Assume that $g_\eta$ and $\nabla g_\eta$ have bounded variances, i.e., for all $\xi$ in the support of $\mathbb{P}(\xi)$ and $\boldsymbol{x} \in \mathbb{R}^p$, $\sigma_g^2 := \mathbb{E}_{\eta|\xi}[\|g_\eta(\boldsymbol{x};\xi) - \mathbb{E}_{\eta|\xi}[g_\eta(\boldsymbol{x};\xi)]\|_2^2] < \infty$ and $\zeta_g^2 := \mathbb{E}_{\eta|\xi}[\|\nabla g_\eta(\boldsymbol{x};\xi) - \mathbb{E}_{\eta|\xi}[\nabla g_\eta(\boldsymbol{x};\xi)]\|_2^2] < \infty$.*

**Assumption 5 (Lipschitz continuity/smoothness of $f_\xi$ and $g_\eta$)** *For all $\xi$ in the support of $\mathbb{P}(\xi)$, $f_\xi(\cdot)$ is $C_f$-Lipschitz continuous (i.e., $\|f_\xi(\boldsymbol{x}) - f_\xi(\boldsymbol{x}')\|_2 \leq C_f \|\boldsymbol{x} - \boldsymbol{x}'\|_2 \ \forall \boldsymbol{x}, \boldsymbol{x}' \in \mathbb{R}^p$) and $L_f$-Lipschitz smooth (i.e., $\|\nabla f_\xi(\boldsymbol{x}) - \nabla f_\xi(\boldsymbol{x}')\|_2 \leq L_f \|\boldsymbol{x} - \boldsymbol{x}'\|_2, \forall \boldsymbol{x}, \boldsymbol{x}' \in \mathbb{R}^p$) for any $\xi$. Similarly, for all $\xi$ in the support of $\mathbb{P}(\xi)$ and $\eta$ in the support of $\mathbb{P}(\eta|\xi)$, $g_\eta(\cdot;\xi)$ is $C_g$-Lipschitz continuous and $L_g$-Lipschitz smooth.*

The smoothness of $f_\xi$ and $g_\eta$ naturally implies the smoothness of $F$. Zhang and Xiao [41, Lemma 4.2] show that Assumption 5 ensures $F$ is: 1) $C_F$-Lipschitz continuous with $C_F = C_f C_g$; and 2) $L_F$-Lipschitz smooth with $L_F = L_g C_f + C_g^2 L_f$. We denote $\tilde{L}_F = \zeta_g C_f + \sigma_g C_g L_f$. Moreover, Assumption 5 also guarantees that $f_\xi$ and $g_\eta$ have bounded gradients. In addition, $f_\xi$ and $g_\eta$ are assumed to satisfy the following regularity condition in order to apply our extrapolation-based scheme from Section 2.

**Assumption 6 (Regularity)** *For all $\xi$ in the support of $\mathbb{P}(\xi)$, $\nabla f_\xi$ is 4th-order differentiable with bounded derivatives (i.e., $a_l := \sup_{\boldsymbol{g} \in \mathbb{R}^p} \|\nabla^{(l)} f_\xi(\boldsymbol{g})\|_2 < \infty$ for $l = 1, 2, 3, 4, \forall \boldsymbol{x} \in \mathbb{R}^p$) and $g_\eta$ has bounded moments upto 4th-order (i.e., $\sigma_k = \sup_{\boldsymbol{x} \in \mathbb{R}^d} \sup_\xi \mathbb{E}_{\eta|\xi} \left[ \sum_{i=1}^p \left[ g_\eta(\boldsymbol{x}) - \mathbb{E}_{\eta|\xi}[g_\eta(\boldsymbol{x})] \right]_i^k \right] < \infty, k = 1, 2, 3, 4$).*

**Remark 2** *The core piece of Assumption 6 is the 4th order differentiability of $\nabla f_\xi$ as other parts can be easily satisfied through appropriate boundedness assumptions. This condition though is satisfied by common instantiations of CSO/FCCO. We discuss some examples including invariant logistic regression, instrumental variable regression, first-order MAML for sine-wave few-shot regression task, deep average precision maximization in Section 5. Therefore, our improvements in sample complexity apply to all these problems.*

Consider some time $t > 0$. Let $G^{t+1}$ be a stochastic estimate of $\nabla F(\boldsymbol{x}^t)$ where $\boldsymbol{x}^t$ is the current iterate. The next iterate $\boldsymbol{x}^{t+1} := \boldsymbol{x}^t - \gamma G^t$. Let $\mathbb{E}[\cdot]$ denote the conditional expectation, where we condition on all the randomness until time $t$. We consider the bias and variance terms coming from our gradient estimate. Formally, we define the following two quantities

$$\mathcal{E}_{\text{bias}}^{t+1} = \left\| \nabla F(\boldsymbol{x}^t) - \mathbb{E}[G^{t+1}] \right\|_2^2, \quad \mathcal{E}_{\text{var}}^{t+1} = \mathbb{E}[\| G^{t+1} - \mathbb{E}[G^{t+1}] \|_2^2].$$

Our idea of getting to an $\varepsilon$-stationary point (Definition 1) will be to ensure that $\mathcal{E}_{\text{bias}}^{t+1}$ and $\mathcal{E}_{\text{var}}^{t+1}$ are bounded. The main technical component of our analyses is in fact analyzing these bias and variance terms for the various gradient estimates considered. For this purpose, we first analyze the bias and variance terms for the (original) BSGD (Lemma 5) and BSpiderBoost (Lemma 7) algorithms, which are then used to get the corresponding bounds for our E-BSGD (Lemma 6) and E-BSpiderBoost (Lemma 8) algorithms. Through these bias and variance bounds, we establish the following main results of this section.

**Theorem 3** *[E-BSGD Convergence] Consider the* (CSO) *problem. Suppose Assumptions 3, 4, 5, 6 hold true and $L_F, C_F, \tilde{L}_F, C_g, F^\star$ are constants and $C_e(f; g) := \frac{8a_3\sigma_3 + 18a_4\sigma_2^2 + 5a_4\sigma_4}{96}$ defined in Corollary 1 are associated with second order extrapolation in the CSO problem. Let step size $\gamma \leq 1/(2L_F)$. Then the output $\boldsymbol{x}^s$ of E-BSGD (Algorithm 2) satisfies: $\mathbb{E}[\|\nabla F(\boldsymbol{x}^s)\|_2^2] \leq \varepsilon^2$, for nonconvex $F$, if the inner batch size $m = \Omega(C_e C_g \varepsilon^{-1/2})$, and the number of iterations*

$$T = \Omega(L_F(F(\boldsymbol{x}^0) - F^\star)(\tilde{L}_F^2/m + C_F^2)\varepsilon^{-4}).$$

The E-BSGD takes $\mathcal{O}(\varepsilon^{-4})$ iterations to converge and compute $\mathcal{O}(\varepsilon^{-0.5})$ gradients per iteration. Therefore, its resulting sample complexity is $\mathcal{O}(\varepsilon^{-4.5})$ which is more efficient than $\mathcal{O}(\varepsilon^{-6})$ of BSGD. Similar improvements can be observed for E-BSpiderBoost in Theorem 4.

**Theorem 4** *[E-BSpiderBoost Convergence] Consider the* (CSO) *problem under the same assumptions as Theorem 3. Let step size $\gamma \leq 1/(13L_F)$. Then the output $\boldsymbol{x}^s$ of E-BSpiderBoost (Algorithm 3)*

*satisfies:* $\mathbb{E}[\|\nabla F(\boldsymbol{x}^s)\|_2^2] \le \varepsilon^2$, *for nonconvex F, if the inner batch size* $m = \mathcal{O}(C_e C_g \varepsilon^{-0.5})$, *the hyperparameters of the outer loop of E-BSpiderBoost* $B_1 = (\tilde{L}_F^2/m + C_F^2)\varepsilon^{-2}$, $B_2 = \sqrt{B_1}$, $p_{out} = 1/B_2$, *and the number of iterations*

$$T = \Omega(L_F(F(\boldsymbol{x}^0) - F^\star)\varepsilon^{-2}).$$

The resulting sample complexity of E-BSpiderBoost is $\mathcal{O}(\varepsilon^{-3.5})$, which improves $\mathcal{O}(\varepsilon^{-5})$ bound of BSpiderBoost [16] and $\mathcal{O}(\varepsilon^{-4})$ bound of V-MLMC/RT-MLMC [17].

## 4 Applying Stochastic Extrapolation in the FCCO Problem

In this section, we apply the extrapolation-based scheme from Section 2 to the FCCO problem. We focus on case where $n = O(\varepsilon^{-2})$. For larger $n$, we can treat the FCCO problem as a CSO problem and get an $\mathcal{O}(\varepsilon^{-3.5})$ bound from Theorem 4. All missing details are presented in Appendix E.

Now, a straightforward algorithm for FCCO is to use the finite-sum variant of SpiderBoost (or SPIDER) [10, 38] in Algorithm 3. In this case, if we choose the outer batch sizes to be $B_1 = n$, $B_2 = \sqrt{n}$ and the inner batch size to be $m = \max\{\varepsilon^{-2}/n, \varepsilon^{-1/2}\}$. The resulting sample complexity of E-BSpiderBoost now becomes, $\mathcal{O}(\max\{\sqrt{n}/\varepsilon^{2.5}, 1/\sqrt{n}\varepsilon^4\})$, which recovers $\mathcal{O}(\varepsilon^{-3.5})$ bound as in Theorem 4 for $n = \Theta(\varepsilon^{-2})$. However, when $n$ is small, such as $n = \mathcal{O}(1)$, the sample complexity degenerates to $\mathcal{O}(\varepsilon^{-4})$ which is worse than the $\Omega(\varepsilon^{-3})$ lower bound of stochastic optimization [3]. We leave the details to Theorem 8. We still use Assumptions 3, 4, 5, 6 for the analysis of FCCO problem, replacing the role of $\xi$ with $i$.

---

**Algorithm 1** E-NestedVR

---

1: **Input:** $\boldsymbol{x}^0 \in \mathbb{R}^d$, step-size $\gamma$, batch sizes $S_1, S_2, B_1, B_2$, Probability $p_{\text{in}}, p_{\text{out}}$
2: **for** $t = 0, 1, \ldots, T-1$ **do**
3:      **if** $(t = 0)$ or (with prob. $p_{\text{out}}$) **then**                           ▷ Large outer batch
4:          **for** $i \in \mathcal{B}_1 \sim [n]$ with $|\mathcal{B}_1| = B_1$ **do**
5:              draw $\boldsymbol{y}_i^{t+1}$ from distribution $\mathcal{D}_{\boldsymbol{y},i}^{t+1}$ defined in (10)
6:              compute $\boldsymbol{z}_i^{t+1}$ using (11) and define $\phi_i^t = \boldsymbol{x}^t$
7:          **end for**
8:          $G_{\text{E-NVR}}^{t+1} = \frac{1}{B_1} \sum_{i \in \mathcal{B}_1} (\boldsymbol{z}_i^{t+1})^\top \mathcal{L}_{\mathcal{D}_{\boldsymbol{y},i}^{t+1}}^{(2)} \nabla f_i(0)$
9:      **else**                                                     ▷ Small outer batch
10:          **for** $i \in \mathcal{B}_2$ with $|\mathcal{B}_2| = B_2$ **do**
11:              draw $\boldsymbol{y}_i^{t+1}$ and $\boldsymbol{y}_i^t$ from distribution $\mathcal{D}_{\boldsymbol{y},i}^{t+1}$ and $\mathcal{D}_{\boldsymbol{y},i}^t$ defined in (10)
12:              compute $\boldsymbol{z}_i^{t+1}$ using (11) and define $\phi_i^t = \boldsymbol{x}^t$
13:          **end for**
14:          $G_{\text{E-NVR}}^{t+1} = G_{\text{E-NVR}}^t + \frac{1}{B_2} \sum_{i \in \mathcal{B}_2} (\boldsymbol{z}_i^{t+1})^\top (\mathcal{L}_{\mathcal{D}_{\boldsymbol{y},i}^{t+1}}^{(2)} \nabla f_i(0) - \mathcal{L}_{\mathcal{D}_{\boldsymbol{y},i}^t}^{(2)} \nabla f_i(0))$
15:      **end if**
16:      $\boldsymbol{x}^{t+1} = \boldsymbol{x}^t - \gamma G_{\text{E-NVR}}^{t+1}$
17: **end for**
18: **Output:** $\boldsymbol{x}^s$ picked uniformly at random from $\{\boldsymbol{x}^t\}_{t=0}^{T-1}$

---

**Extrapolated NestedVR.** We now introduce a nested variance reduction algorithm E-NestedVR which reaches low sample complexity for all choices of $n$. Missing proofs from this section are presented in Appendix E. For the stochasticities in the FCCO problem, our idea is to use two nested SpiderBoost variance reduction components: one for the outer random variable $i$ and the other for the inner random variable $\eta|i$. In each outer (resp. inner) SpiderBoost step, we choose large batch $B_1$ (resp. $S_1$) with probability $p_{\text{out}}$ (resp. $p_{\text{in}}$); otherwise we choose small batch. Let $H_i$ denote a set of $m$ i.i.d. samples drawn from the conditional distribution $\mathbb{P}(\eta|i)$. Similarly, let $\tilde{H}_i$ denote another set of $m$ i.i.d. samples drawn from the same conditional distribution. For each given $i$, we approximate $\mathbb{E}_{\eta|i}[g_\eta(\boldsymbol{x}^t)]$ with $\boldsymbol{y}_i^{t+1}$ from distribution $\mathcal{D}_{\boldsymbol{y},i}^{t+1}$ where,

$$\boldsymbol{y}_i^{t+1} = \begin{cases} \frac{1}{S_1} \sum_{\eta \in H_i} g_\eta(\boldsymbol{x}^t) & \text{with prob. } p_{\text{in}} \text{ or } t = 0 \\ \boldsymbol{y}_i^t + \frac{1}{S_2} \sum_{\eta \in H_i} (g_\eta(\boldsymbol{x}^t) - g_\eta(\phi_i^t)) & \text{with prob. } 1 - p_{\text{in}}. \end{cases} \quad (10)$$

Similarly, we approximate $\mathbb{E}_{\tilde{\eta}|i}[\nabla g_{\tilde{\eta}}(\boldsymbol{x}^t)]$ with $\boldsymbol{z}_i^{t+1}$ defined as follows

$$\boldsymbol{z}_i^{t+1} = \begin{cases} \frac{1}{S_1} \sum_{\tilde{\eta} \in \tilde{H}_i} \nabla g_{\tilde{\eta}}(\boldsymbol{x}^t) & \text{with prob. } p_{\text{in}} \text{ or } t = 0 \\ \boldsymbol{z}_i^t + \frac{1}{S_2} \sum_{\tilde{\eta} \in \tilde{H}_i} (\nabla g_{\tilde{\eta}}(\boldsymbol{x}^t) - \nabla g_{\tilde{\eta}}(\phi_i^t)) & \text{with prob. } 1 - p_{\text{in}}, \end{cases} \quad (11)$$

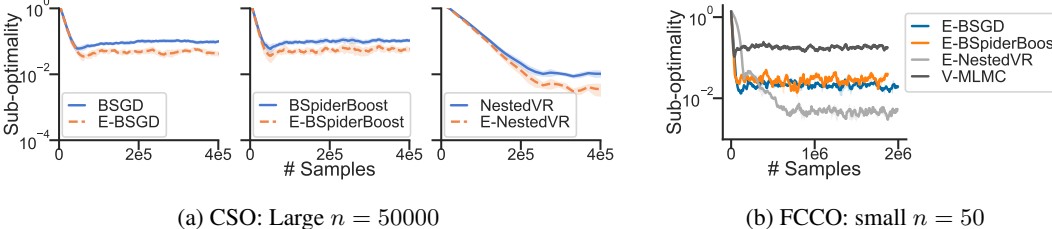

(a) CSO: Large $n = 50000$          (b) FCCO: small $n = 50$

Figure 2: Performances of algorithms and their extrapolated versions on the invariant logistic regression task. Algorithms in each subplot use the same amount of inner batch size in each iteration. The shaded region represents the 95%-confidence interval computed over 10 runs.

where $\phi_i^t$ is the last time $i$ is visited before time $t$. If $i$ is not selected at time $t$, then $\boldsymbol{y}_i^{t+1} = \boldsymbol{y}_i^t$ and $\boldsymbol{z}_i^{t+1} = \boldsymbol{z}_i^t$. Note that we use independent samples for $\boldsymbol{y}_i^{t+1}$ and $\boldsymbol{z}_i^{t+1}$.

Finally, we present E-NestedVR in Algorithm 1 where second-order extrapolation operator $\mathcal{L}_\cdot^{(2)}$ is applied to each occurrence of $\nabla f_i$. We now analyze its convergence guarantee. Our analysis works by first looking at the effect of multi-level variance reduction without the extrapolation (that we refer to as NestedVR, Theorem 10, Appendix E.2), and then showing how extrapolation could further help to drive down the sample complexity.

**Theorem 5** *[E-NestedVR Convergence] Consider the* (FCCO) *problem. Under the same assumptions as Theorem 3.*

- *If $n = \mathcal{O}(\varepsilon^{-2/3})$, then we choose the hyperaparameters of E-NestedVR (Algorithm 1) as* $B_1 = B_2 = n, p_{out} = 1, S_1 = \tilde{L}_F^2 \varepsilon^{-2}, S_2 = \tilde{L}_F \varepsilon^{-1}, p_{in} = \tilde{L}_F^{-1} \varepsilon, \gamma = \mathcal{O}(\frac{1}{L_F})$.

- *If $n = \Omega(\varepsilon^{-2/3})$, then we choose the hyperaparameters of E-NestedVR as* $B_1 = n, B_2 = \sqrt{n}, p_{out} = 1/\sqrt{n}, S_1 = S_2 = \max\left\{C_e C_g \varepsilon^{-1/2}, \tilde{L}_F^2/(n\varepsilon^2)\right\}, p_{in} = 1, \gamma = \mathcal{O}(\frac{1}{L_F})$.

*Then the output $\boldsymbol{x}^s$ of E-NestedVR satisfies: $\mathbb{E}[\|\nabla F(\boldsymbol{x}^s)\|_2^2] \leq \varepsilon^2$, for nonconvex $F$ with iterations*
$$T = \Omega\left(L_F(F(\boldsymbol{x}^0) - F^\star)\varepsilon^{-2}\right).$$

From Theorem 5, E-NestedVR has a sample complexity of $\mathcal{O}(n\varepsilon^{-3})$ in the small $n$ regime ($n = \mathcal{O}(\varepsilon^{-2/3})$) and $\mathcal{O}(\max\{\sqrt{n}/\varepsilon^{2.5}, 1/\sqrt{n}\varepsilon^4\})$ in the large $n$ regime ($n = \Omega(\varepsilon^{-2/3})$). Therefore, in the large $n$ regime, this improves the $\mathcal{O}(n\varepsilon^{-3})$ sample complexity of MSVR-V2 [19].

# 5 Applications

In this section, we demonstrate the numerical performance of our proposed algorithms. We focus on the application of invariant logistic regression here. In Appendix F, we discuss performance of our proposed algorithms on other common CSO/FCCO applications.

## 5.1 Application of Invariant Risk Minimization

Invariant learning has wide applications in machine learning and related areas [22, 1]. Invariant logistic regression [16] is formulated as follows:
$$\min_{\boldsymbol{x}} \mathbb{E}_{\xi=(\boldsymbol{a},b)}[\log(1 + \exp(-b\mathbb{E}_{\eta|\xi}[\eta]^\top \boldsymbol{x})],$$

where $\boldsymbol{a}$ and $b$ represent a sample and its corresponding label, and $\eta$ is a noisy observation of the sample $\boldsymbol{a}$. This first part can be considered as a CSO objective, with $f_\xi(y) := \log(1 + \exp(-by))$ and $g_\eta(\boldsymbol{x}; \xi) := \eta^\top \boldsymbol{x}$. As the loss $f_\xi \in \mathcal{C}^\infty$ is smooth, our results from Sections 3 and 4 are applicable.

An $\ell_2$-regularizer is added to ensure the existence of an unique minimizer. Since the gradient of the penalization term is unbiased, we only have to consider the bias of the data-dependent term. We generate a synthetic dataset with $d = 10$ dimensions. The minimizer is drawn from Gaussian distribution $\boldsymbol{x}^\star \sim \mathcal{N}(0, 1) \in \mathbb{R}^d$. We draw invariant samples $\{(\boldsymbol{a}_i, b_i)\}_i$ where $\boldsymbol{a}_i \sim \mathcal{N}(0, 1) \in \mathbb{R}^d$ and compute $b_i = \text{sgn}(\boldsymbol{a}_i^\top \boldsymbol{x}^\star)$ and its perturbed observation $\eta \sim \mathcal{N}(\boldsymbol{a}_i, 100) \in \mathbb{R}^d$.

We consider drawing $\xi$ from a large set ($n = 50000$) and a small set ($n = 50$) as CSO and FCCO problems respectively. As baselines, we implemented the BSGD and BSpiderBoost methods

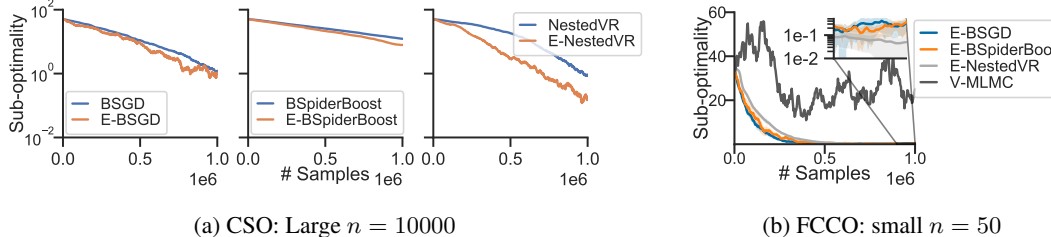

(a) CSO: Large $n = 10000$             (b) FCCO: small $n = 50$

Figure 3: Performances of algorithms and their extrapolated versions on the instrumental variable regression task. The shaded region represents the 95%-confidence interval.

from [16], V-MLMC approach from [17], and NestedVR approach from Appendix E.2 which achieves the same complexity as MSVR-V2 [19] for the FCCO problem. RT-MLMC shares the same sample complexity as V-MLMC, and is thus omitted from the experiment [17]. The results are shown in Fig. 2. In the CSO setting, we compare biased gradient methods with their extrapolated variants (BSGD vs. E-BSGD, BSpiderBoost vs. E-BSpiderBoost, and NestedVR vs. E-NestedVR). The extrapolated versions of BSGD, BSpiderBoost, and NestedVR consistently reach lower error than their non-extrapolated counterparts, as is evident in Figure 2a. In this case, the performance of BSpiderBoost is similar to BSGD as also noted by the authors of these techniques [16], and a drawback of BSpiderBoost seems to be that it is much harder to tune in practice. However, it is clear that E-BSGD outperforms BSGD, and E-BSpiderBoost outperforms BSpiderBoost, respectively. In the FCCO setting, we compare extrapolation based methods and MLMC based methods. Figure 2a, shows that E-NestedVR outperforms all other extrapolated algorithms, including the V-MLMC approach of [17], matching our theoretical findings.

## 5.2 Application of Instrumental Variable Regression

Instrumental variable regression is a popular technique in econometrics that aims to estimate the causal effect of input $X$ on a confounded outcome $Y$ with an instrument variable $Z$, which, for a given $X$, is conditionally independent of $Y$. As noted by [23], the instrumental variable regression is a special case of the CSO problem. The instrumental variable regression problem is phrased as:

$$\min_w \mathbb{E}_{\xi=(Y,Z)} \left[ \ell(Y, \mathbb{E}_{X|Z}[g_X(w)]) \right]$$

where $\xi = (Y, Z), \eta = X$. This can be viewed in the CSO framework with $f_\xi(\cdot) = \ell(Y, \cdot)$. We choose $\ell(y, \hat{y}) = \log \cosh(y - \hat{y})$ as regression loss function and $g_X(w) = w^\top X$ to be a linear regressor. In this case, $f_\xi \in \mathcal{C}^\infty$ with $\nabla f_\xi(\hat{y}) = \tanh(\hat{y} - Y)$, and our results from Sections 3 and 4 apply. We generate the data similar to [31]

$$Z \sim \text{Uniform}([-3,3]^2), \quad e \sim \mathcal{N}(0,1), \quad \delta \sim \mathcal{N}(0,0.1), \quad \gamma \sim \text{Exponential}(10)$$
$$X = \tfrac{1}{2}z_1 + \tfrac{1}{2}e + \gamma, \quad Y = X + e + \delta$$

where $z_1$ is the first dimension of $Z$, $e$ is the confounding variable and $\delta, \gamma$ are noises. In this experiment, we solve the instrumental variable regression using BSGD, BSpiderBoost, NestedVR and their extrapolated variants described in Sections 3 and 4. In each pair of experiments, the samples used per iteration are fixed same, i.e.: 1) BSGD uses $m = 2$ and E-BSGD uses $m = 1$; 2) For BSpiderBoost and E-BSpiderBoost, we use cycle length of 10, small batch and large batch in Spider to be 10 and 100 respectively, and we choose inner batch sizes $m = 2$ for BSpiderBoost and $m = 1$ for E-BSpiderBoost; 3) For NestedVR and E-NestedVR, we fix the outer batch size to 10 and 5 respectively, and choose fix the inner Spider Cycle to be 10 with large batch 100 and small batch 10. The results are presented in Figure 3. As is quite evident, the extrapolation variants achieve faster convergence in all 3 cases, confirming our theoretical findings.

## 6 Concluding Remarks

In this paper, we consider the conditional stochastic optimization CSO problem and its finite-sum variant FCCO. Due to the interplay between nested structure and stochasticity, most of the existing gradient estimates suffer from large biases and have large sample complexity of $\mathcal{O}(\varepsilon^{-5})$. We propose stochastic extrapolation-based algorithms that tackle this bias problem and improve the sample complexities for both these problems. While we focus on nonconvex objectives, our proposed algorithms can also be beneficial when used with strongly convex, convex objectives. We also believe that similar ideas could also prove helpful for multi-level stochastic optimization problems [41] with nested dependency.

**Acknowledgements**

We would like to thank Caner Turkmen, Sai Praneeth Karimireddy, and Martin Jaggi for helpful initial discussions surrounding this project.

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

# Appendix

## A  Missing Pseudocodes

We present pseudocodes of E-BSGD and E-BSpiderBoost scheme in Algorithms 2 and 3 respectively.

---
**Algorithm 2** E-BSGD
---
1: **Input:** $\boldsymbol{x}^0 \in \mathbb{R}^d$, step-size $\gamma$, batch sizes $m$
2: **for** $t = 0, 1, \ldots, T - 1$ **do**
3:      Draw one sample $\xi$ and compute extrapolated gradient $G_{\text{E-BSGD}}^{t+1}$ from (7)
4:      $\boldsymbol{x}^{t+1} \leftarrow \boldsymbol{x}^t - \gamma G_{\text{E-BSGD}}^{t+1}$
5: **end for**
6: **Output:** $\boldsymbol{x}^s$ picked uniformly at random from $\{\boldsymbol{x}^t\}_{t=0}^{T-1}$

---

## B  Missing Details from Section 1

### B.1  Other Related Work

**CSO.** Dai et al. [5] proposed a primal-dual stochastic approximation algorithm to solve a min-max reformulation of CSO, employing the kernel embedding techniques. However, this method requires convexity of $f_\xi$ and linearity of $g_\eta$, which are not satisfied by general applications when neural networks are involved. Goda and Kitade [13] showed that a special class of CSO problems can be

---

**Algorithm 3** E-BSpiderBoost

---

1: **Input:** $\boldsymbol{x}^0 \in \mathbb{R}^d$, step-size $\gamma$, batch sizes $B_1$, $B_2$, Probability $p_{\text{out}}$
2: **for** $t = 0, 1, \ldots, T - 1$ **do**
3:     Draw $\chi_{\text{out}}$ from **Bernoulli**$(p_{\text{out}})$
4:     **if** $(t = 0)$ or $(\chi_{\text{out}} = 1)$ **then**                                            ▷ Large batch
5:         Draw $\mathcal{B}_1$ outer samples $\{\xi_1, \ldots, \xi_{B_1}\}$
6:         Compute extrapolated gradient $G_{\text{E-BSGD}}^{t+1}$ with (7)

$$G_{\text{E-BSB}}^{t+1} = \tfrac{1}{B_1} \sum_{\xi \in \mathcal{B}_1} G_{\text{E-BSGD}}^{t+1}$$

7:     **else**                                                          ▷ Small batch
8:         Draw $\mathcal{B}_2$ outer samples $\{\xi_1, \ldots, \xi_{B_2}\}$
9:         Compute extrapolated gradient $G_{\text{E-BSGD}}^{t+1}$ with (7)

$$G_{\text{E-BSB}}^{t+1} = G_{\text{E-BSB}}^{t} + \tfrac{1}{B_2} \sum_{\xi \in \mathcal{B}_2} (G_{\text{E-BSGD}}^{t+1} - G_{\text{E-BSGD}}^{t})$$

10:     **end if**
11:     $\boldsymbol{x}^{t+1} = \boldsymbol{x}^t - \gamma G_{\text{E-BSB}}^{t+1}$
12: **end for**
13: **Output:** $\boldsymbol{x}^s$ picked uniformly at random from $\{\boldsymbol{x}^t\}_{t=0}^{T-1}$

---

unbiased, e.g., when $f_\xi$ measures the squared error between some $u(\xi)$ and $\mathbb{E}_{\eta|\xi}[g_\eta(\boldsymbol{x}; \xi)]$, giving rise to this objective function $\mathbb{E}_\xi[(u(\xi) - \mathbb{E}_{\eta|\xi}[g_\eta(\boldsymbol{x}; \xi)])^2]$. However, they did not show any improvement over the sample complexity of BSGD (i.e., $\mathcal{O}(\varepsilon^{-6})$). Hu et al. [16] also analyzed lower bounds on the minimax error for the CSO problem and showed that for a specific class of biased gradients with $\mathcal{O}(\varepsilon)$ bias (same bias as BSGD) and variance $\mathcal{O}(1)$ the bound achieved by BSpiderBoost is tight. However, these lower bounds are not applicable in settings such as ours (and also to [17]) where the bias is smaller than the BSGD bias.

**Variance Reduction.** The reduction of variance in stochastic optimization is a crucial approach to decrease sample complexity, particularly when dealing with finite-sum formulations of the form $\min_{\boldsymbol{x}} \frac{1}{n} \sum_{i=1}^n f_i(\boldsymbol{x})$. Pioneering works such as Stochastic Average Gradient (SAG) [30], Stochastic Variance Reduced Gradient (SVRG) [21, 28], and SAGA [7, 29] improved the iteration complexity from $\mathcal{O}(\varepsilon^{-4})$ in Stochastic Gradient Descent (SGD) to $\mathcal{O}(\varepsilon^{-2})$. Subsequent research, including Stochastic Path-Integrated Differential Estimator (SPIDER) [10] and Stochastic Recursive Gradient Algorithm (SARAH) [25], expanded the application of these techniques to both finite-sum and online scenarios, where $n$ is large or possibly infinite. These methods boast an improved sample complexity of $\min(\sqrt{n}\varepsilon^{-2}, \varepsilon^{-3})$. SpiderBoost [38], achieves the same near-optimal complexity performance as SPIDER, but allows a much larger step size and hence runs faster in practice than SPIDER. In this paper, we use a probabilistic variant of SpiderBoost as the variance reduction module for CSO and FCCO problems. We highlight that alternative techniques, such as SARAH, can also be applied and offer similar guarantees.

**Bias Correction.** One of the classic problems in statistics is to design procedures to reduce the bias of estimators. Well-established general bias correction techniques, such as the jackknife [32], bootstrap [8], Taylor series [39, 14], have been extensively studied and applied in various contexts [20]. However, these methods are predominantly examined in relation to standard statistical distributions, with limited emphasis on their adaptability to optimization problems. Our proposed extrapolation-based approach is derived from sample-splitting methods [14], specifically tailored and analyzed for optimization problems involving unknown distributions.

**Stochastic Composition Optimization.** Finally, a closely related class of problems, called stochastic composition optimization, has been extensively studied (e.g., [40, 9, 36, 37]) in the literature where the goal is:

$$\min_{\boldsymbol{x} \in \mathbb{R}^d} \mathbb{E}_\xi[f_\xi(\mathbb{E}_\eta[g_\eta(\boldsymbol{x})])]. \tag{12}$$

Despite having nested expectations in their formulations (CSO) and (12) are fundamentally different: a) in stochastic composite optimization the inner randomness $\eta$ is conditionally dependent on the outer randomness $\xi$ and b) in CSO the inner random function $g_\eta(\boldsymbol{x}, \xi)$ depends on both $\xi$ and $\eta$. These differences lead to quite different sample complexity bounds for these problems, as explored in Hu et al. [15]. In fact, Zhang and Xiao [41] presented a near optimal complexity of $\mathcal{O}(\min(\varepsilon^{-3}, \sqrt{n}\varepsilon^{-2}))$

for stochastic composite optimization problems using nested variance reduction. While Wang et al. [36] also use the "extrapolation" technique, their motivation and formula are significantly different from ours and cannot reduce the bias in the CSO problem.

## C Missing Details from Section 2

**Lemma 1 (Moments of $\mathcal{D}_m$)** *The moments of $\delta \in \mathcal{D}_m$ are bounded as follows*

$$\mathbb{E}[(\delta - \mathbb{E}[\delta])^2] = \frac{\sigma_2}{m}, \quad |\mathbb{E}[(\delta - \mathbb{E}[\delta])^3]| = \frac{\sigma_3}{m^2}, \quad \mathbb{E}[(\delta - \mathbb{E}[\delta])^4] = \frac{\sigma_4}{m^3} + \frac{3(m-1)\sigma_2^2}{m^3}.$$

*More generally, for $k \geq 2$, $|\mathbb{E}[(\delta - \mathbb{E}[\delta])^k]| = \mathcal{O}(m^{-\lceil k/2 \rceil})$.*

**Proof:** Define $\hat{\delta} = \delta - \mathbb{E}[\delta]$ as the centered random variable. Now

$$\mathbb{E}[(\delta - \mathbb{E}[\delta])^k] = \mathbb{E}[\hat{\delta}^k].$$

So we focus on $\mathbb{E}[\hat{\delta}^k]$ in the remainder of the proof. For $k = 2$,

$$|\mathbb{E}[\hat{\delta}^2]| = \frac{1}{m^2}|\mathbb{E}[\sum_{i=1}^m \hat{\delta}_i]^2| = \frac{1}{m^2}\left|\mathbb{E}\left[\sum_i \hat{\delta}_i^2 + 2\sum_{i<j} \hat{\delta}_i \hat{\delta}_j\right]\right| = \frac{\sigma_2}{m}.$$

For $k = 3$,

$$|\mathbb{E}[\hat{\delta}^3]| = \frac{1}{m^3}|\mathbb{E}[\sum_{i=1}^m \hat{\delta}_i]^3|$$
$$= \frac{1}{m^3}\left|\mathbb{E}\left[\sum_i \hat{\delta}_i^3 + 3\sum_{i\neq j} \hat{\delta}_i^2 \hat{\delta}_j + 6\sum_{i<j<k} \hat{\delta}_i \hat{\delta}_j \hat{\delta}_k\right]\right|$$
$$= \frac{\sigma_3}{m^2}.$$

For $k = 4$,

$$|\mathbb{E}[\hat{\delta}^4]| = \frac{1}{m^4}|\mathbb{E}[\sum_{i=1}^m \hat{\delta}_i]^4|$$
$$= \frac{1}{m^4}\left|\mathbb{E}\left[\sum_i \hat{\delta}_i^4 + 4\sum_{i\neq j} \hat{\delta}_i^3 \hat{\delta}_j + 6\sum_{i<j} \hat{\delta}_i^2 \hat{\delta}_j^2 + 24\sum_{i<j<k<l} \hat{\delta}_i \hat{\delta}_j \hat{\delta}_k \hat{\delta}_l\right]\right|$$
$$= \frac{1}{m^4}\left|m \mathbb{E}[\hat{\delta}_i^4] + 6\frac{m(m-1)}{2} \mathbb{E}[\hat{\delta}_i^2] \mathbb{E}[\hat{\delta}_j^2]\right|$$
$$= \frac{\sigma_4}{m^3} + \frac{3(m-1)\sigma_2^2}{m^3}.$$

For $k = 5$,

$$|\mathbb{E}[\hat{\delta}^5]| = \frac{1}{m^5}|\mathbb{E}[\sum_{i=1}^m \hat{\delta}_i]^5|$$
$$= \frac{1}{m^5}\left|\mathbb{E}\left[\sum_i \hat{\delta}_i^5 + 10\sum_{i\neq j} \hat{\delta}_i^3 \hat{\delta}_j^2\right]\right|$$
$$= \frac{1}{m^5}\left|m \mathbb{E}[\hat{\delta}_i^5] + 10m(m-1) \mathbb{E}[\hat{\delta}_i^3] \mathbb{E}[\hat{\delta}_j^2]\right|$$
$$= \frac{\sigma_5}{m^4} + \frac{10(m-1)\sigma_3\sigma_2}{m^4}.$$

For general $k > 0$, we expand the following term as a function of $m$

$$|\mathbb{E}[\hat{\delta}^k]| = \frac{1}{m^k}|\mathbb{E}[\sum_{i=1}^m \hat{\delta}_i]^k|.$$

As $\mathbb{E}[\hat{\delta}_i] = 0$ and $\hat{\delta}_i$ and $\hat{\delta}_j$ are independent for different $i$ and $j$, the outcome has the following form

$$|\mathbb{E}[\hat{\delta}^k]| = \frac{1}{m^k}\mathcal{O}\left(\sum_{\substack{2a_2+3a_3+\cdots+ka_k=k \\ a_i\geq 0 \; \forall i}} m^{\sum_{i=2}^k a_i}\sigma_2^{a_2}\sigma_3^{a_3}\cdots\sigma_k^{a_k}\right) \tag{13}$$

where $\sum_{i=2}^k a_i$ is the count of independent $\{\hat{\delta}_i\}$ used in $\sigma_2^{a_2}\sigma_3^{a_3}\cdots\sigma_4^{a_4}$. Among the terms in (13), the dominating one in terms of $m$ is one with largest $\sum_{i=2}^k a_i$, i.e.

$$|\mathbb{E}[\hat{\delta}^k]| = \begin{cases} \frac{1}{m^k}\mathcal{O}(m^{k/2})\sigma_2^{k/2} & \text{if k even,} \\ \frac{1}{m^k}\mathcal{O}(m^{\lfloor k/2 \rfloor})\sigma_2^{\lfloor k/2 \rfloor-1}\sigma_3 & \text{if k odd.} \end{cases}$$

Then, we can simplify the upper right-hand side with

$$|\mathbb{E}[\hat{\delta}^k]| = \mathcal{O}(m^{-k+\lfloor k/2 \rfloor}),$$

which gives all the desired results. $\square$

**Proposition 2 (First-order Guarantee)** *Assume that $\mathcal{D}_m$ and $q(\cdot)$ satisfy Assumption 1 and 2 respectively with $k = 1$. Then, $\forall s \in \mathbb{R}$, $\left| \mathbb{E}\left[ \mathcal{L}^{(1)}_{\mathcal{D}_m} q(s) \right] - q(s + \mathbb{E}[\delta]) \right| \leq a_2 \sigma_2 / (2m)$.*

**Proof:** Let $h = \mathbb{E}[\delta]$. If the function $q \in \mathcal{C}^2$, then the Taylor expansion at $s + h$ with remainders leads to

$$\mathbb{E}[q(s + \delta)] = q(s + h) + q'(s + h)\,\mathbb{E}[\delta - h] + \tfrac{1}{2}\,\mathbb{E}[q''(\phi_1)(\delta - h)^2]$$

where $\phi_1$ between $s + h$ and $s + \delta$. Then the error of extrapolation becomes

$$\left| \mathbb{E}[q(s + \delta)] - q(s + h) \right| = \left| \tfrac{1}{2}\,\mathbb{E}[q''(\phi_1)(\delta - h)^2] \right| \leq \tfrac{a_2}{2}\,\mathbb{E}[(\delta - h)^2].$$

By Assumption 2 and Lemma 1, we have that

$$\left| \mathbb{E}[q(s + \delta)] - q(s + h) \right| \leq \tfrac{a_2}{2}\,\mathbb{E}[(\delta - h)^2] = \tfrac{a_2}{2}\,\mathbb{E}[(\delta - h)^2] = \tfrac{a_2 \sigma_2}{2m}.$$

This completes the proof. $\square$

**Proposition 1 (Second-order Guarantee)** *Assume that distribution $\mathcal{D}_m$ and $q(\cdot)$ satisfies Assumption 1 and 2 respectively with $k = 2$. Then, for all $s \in \mathbb{R}$, $\left| \mathbb{E}\left[ \mathcal{L}^{(2)}_{\mathcal{D}_m} q(s) \right] - q(s + \mathbb{E}[\delta]) \right| \leq$ $\frac{4a_3\sigma_3 + 9a_4\sigma_2^2}{48m^2} + \frac{5a_4}{96}\frac{\sigma_4 - 3\sigma_2^2}{m^3}$.*

**Proof:** Let $h = \mathbb{E}[\delta]$. If the function $q \in \mathcal{C}^4$, then the Taylor expansion at $s + h$ with remainders leads to

$$\mathbb{E}[q(s + \delta_1)] = q(s + h) + q'(s + h)\,\mathbb{E}[\delta_1 - h] + \tfrac{q''(s+h)}{2}\,\mathbb{E}[(\delta_1 - h)^2] + \tfrac{q^{(3)}(s+h)}{6}\,\mathbb{E}[(\delta_1 - h)^3]$$
$$+ \tfrac{1}{24}\,\mathbb{E}[q^{(4)}(\phi_1)(\delta_1 - h)^4]$$

$$\mathbb{E}[q(s + \delta_2)] = q(s + h) + q'(s + h)\,\mathbb{E}[\delta_2 - h] + \tfrac{q''(s+h)}{2}\,\mathbb{E}[(\delta_2 - h)^2] + \tfrac{q^{(3)}(s+h)}{6}\,\mathbb{E}[(\delta_2 - h)^3]$$
$$+ \tfrac{1}{24}\,\mathbb{E}[q^{(4)}(\phi_2)(\delta_2 - h)^4]$$

$$\mathbb{E}[q(s + \tfrac{\delta_1+\delta_2}{2})] = q(s + h) + q'(s + h)\,\mathbb{E}[\tfrac{\delta_1+\delta_2}{2} - h] + \tfrac{q''(s+h)}{2}\,\mathbb{E}[(\tfrac{\delta_1+\delta_2}{2} - h)^2]$$
$$+ \tfrac{q^{(3)}(s+h)}{6}\,\mathbb{E}[(\tfrac{\delta_1+\delta_2}{2} - h)^3] + \tfrac{1}{24}\,\mathbb{E}[q^{(4)}(\phi_3)(\tfrac{\delta_1+\delta_2}{2} - h)^4]$$

where $\phi_1, \phi_2, \phi_3$ between $s + h$ and $s + \delta_1$, $s + \delta_2$, $s + \delta_3$ respectively.

As $\mathbb{E}[\delta - h] = 0$, the error of extrapolation becomes

$$\left| \mathbb{E}[\mathcal{L}^2_{\mathcal{D}_m} q(s)] - q(s + h) \right|$$
$$\leq \left| 2\,\mathbb{E}\left[ \tfrac{q^{(3)}(s+h)}{6}\left(\tfrac{\delta_1+\delta_2}{2} - h\right)^3 \right] - \tfrac{1}{2}\left( \mathbb{E}[\tfrac{q^{(3)}(s+h)}{6}(\delta_1 - h)^3] + \mathbb{E}[\tfrac{q^{(3)}(s+h)}{6}(\delta_2 - h)^3] \right) \right|$$
$$+ \left| 2\,\mathbb{E}\left[ \tfrac{q^{(4)}(\phi_3)}{24}\left(\tfrac{\delta_1+\delta_2}{2} - h\right)^4 \right] - \tfrac{1}{2}\left( \mathbb{E}[\tfrac{q^{(4)}(\phi_1)}{24}(\delta_1 - h)^4] + \mathbb{E}[\tfrac{q^{(4)}(\phi_2)}{24}(\delta_2 - h)^4] \right) \right|$$
$$\leq \tfrac{a_3}{6}\left| 2\,\mathbb{E}\left[ \left(\tfrac{\delta_1+\delta_2}{2} - h\right)^3 \right] - \tfrac{1}{2}\left( \mathbb{E}[(\delta_1 - h)^3] + \mathbb{E}[(\delta_2 - h)^3] \right) \right|$$
$$+ \left| 2\,\mathbb{E}\left[ \tfrac{q^{(4)}(\phi_3)}{24}\left(\tfrac{\delta_1+\delta_2}{2} - h\right)^4 \right] - \tfrac{1}{2}\left( \mathbb{E}[\tfrac{q^{(4)}(\phi_1)}{24}(\delta_1 - h)^4] + \mathbb{E}[\tfrac{q^{(4)}(\phi_2)}{24}(\delta_2 - h)^4] \right) \right|$$
$$\leq \tfrac{a_3}{6}\left| 2\,\mathbb{E}\left[ \left(\tfrac{\delta_1+\delta_2}{2} - h\right)^3 \right] - \tfrac{1}{2}\left( \mathbb{E}[(\delta_1 - h)^3] + \mathbb{E}[(\delta_2 - h)^3] \right) \right|$$
$$+ \left| 2\,\mathbb{E}\left[ \tfrac{|q^{(4)}(\phi_3)|}{24}\left(\tfrac{\delta_1+\delta_2}{2} - h\right)^4 \right] + \tfrac{1}{2}\left( \mathbb{E}[\tfrac{|q^{(4)}(\phi_1)|}{24}(\delta_1 - h)^4] + \mathbb{E}[\tfrac{|q^{(4)}(\phi_2)|}{24}(\delta_2 - h)^4] \right) \right|$$
$$\leq \tfrac{a_3}{6}\left| 2\,\mathbb{E}\left[ \left(\tfrac{\delta_1+\delta_2}{2} - h\right)^3 \right] - \tfrac{1}{2}\left( \mathbb{E}[(\delta_1 - h)^3] + \mathbb{E}[(\delta_2 - h)^3] \right) \right|$$
$$+ \tfrac{a_4}{24}\left| 2\,\mathbb{E}\left[ \left(\tfrac{\delta_1+\delta_2}{2} - h\right)^4 \right] + \tfrac{1}{2}\left( \mathbb{E}[(\delta_1 - h)^4] + \mathbb{E}[(\delta_2 - h)^4] \right) \right|.$$

where the second inequality uses the upper bound on $q^{(3)}(\cdot)$ (Assumption 2) and the third inequality uses $(\delta - h)^4$ is non-negative and the last inequality uses the uniform bound on $q^{(4)}(\cdot)$ (Assumption 2).

Then

$$|\mathbb{E}[\mathcal{L}^2_{\mathcal{D}_m} q(s)] - q(s+h)|$$

$$\leq \tfrac{a_3}{12} |\mathbb{E}(\delta_1 - h)^3| + \tfrac{a_4}{24}\left(2\,\mathbb{E}\left(\tfrac{\delta_1+\delta_2}{2} - h\right)^4 + \mathbb{E}(\delta_1 - h)^4\right)$$

$$\leq \tfrac{a_3\sigma_3}{12m^2} + \tfrac{a_4}{24}\left(\tfrac{\sigma_4}{4m^3} + \tfrac{3(2m-1)\sigma_2^2}{4m^3} + \tfrac{\sigma_4}{m^3} + \tfrac{3(m-1)\sigma_2^2}{m^3}\right)$$

$$\leq \tfrac{a_3\sigma_3}{12m^2} + \tfrac{a_4}{24}\left(\tfrac{9\sigma_2^2}{2m^2} + \tfrac{5(\sigma_4 - 3\sigma_2^2)}{4m^3}\right)$$

$$\leq \tfrac{4a_3\sigma_3 + 9a_4\sigma_2^2}{48m^2} + \tfrac{5a_4}{96}\tfrac{\sigma_4 - 3\sigma_2^2}{m^3}.$$

we first use that $\mathbb{E}[(\delta_1 - h)^3] = \mathbb{E}[(\delta_2 - h)^3] = 4\,\mathbb{E}[(\tfrac{\delta_1+\delta_2}{2} - h)^3]$ and the uses the bound on moments in Lemma 1. Note that $\mathbb{E}\left(\tfrac{\delta_1+\delta_2}{2} - h\right)^4$ can be seen as the 4th order moments of a batch size of $2m$.

$\square$

**Proposition 3** *Assume $q \in \mathcal{C}^6$. Then $\mathcal{L}^{(3)}_{\mathcal{D}_m}$ as defined below is a third-order extrapolation operator.*

$$\mathcal{L}^{(3)}_{\mathcal{D}_m} q : s \mapsto (-\tfrac{1}{36}\mathcal{L}^{(2)}_{\mathcal{D}_m} + \tfrac{5}{9}\mathcal{L}^{(2)}_{\mathcal{D}_{2m}} - \tfrac{3}{4}\mathcal{L}^{(2)}_{\mathcal{D}_{3m}} - \tfrac{16}{9}\mathcal{L}^{(2)}_{\mathcal{D}_{4m}} + 3\mathcal{L}^{(2)}_{\mathcal{D}_{6m}})q(s).$$

**Proof:** Let $h = \mathbb{E}[\delta]$. If $q \in \mathcal{C}^{2k}$, then $q$ has the following Taylor expansion

$$\mathbb{E}[q(s+\delta)] = \underbrace{q(s+h)}_{\text{zero order term}} + q'(s+h)\,\mathbb{E}[\delta - h] + \underbrace{\tfrac{q''(s+h)}{2}\,\mathbb{E}[(\delta - h)^2]}_{\text{second order term}} + \dots$$

$$+ \tfrac{q^{(2k-1)}(s+h)}{(2k-1)!}\,\mathbb{E}[(\delta - h)^{2k-1}] + \tfrac{1}{2k!}\,\mathbb{E}[q^{(2k)}(\phi)(\delta - h)^{2k}].$$

**Eliminate the third order term in the Taylor expansion.** Consider the following affine combination which

$$\mathcal{F}^{(3)}_{\mathcal{D}_m} q : s \mapsto \alpha_1 \mathcal{L}^{(2)}_{\mathcal{D}_m} q(s) + \alpha_2 \mathcal{L}^{(2)}_{\mathcal{D}_{2m}} q(s).$$

We determine $\alpha_1$ and $\alpha_2$ by expanding $\mathcal{L}^{(2)}_{\mathcal{D}_m} q(s)$ and $\mathcal{L}^{(2)}_{\mathcal{D}_{2m}} q(s)$ and analyze the coefficients of terms:

- **(Affine).** Taylor expansion of $\mathcal{F}^{(3)}_{\mathcal{D}_m} q(s)$ at $s + h$ should have zero order term $q(s+h)$, i.e.

$$\alpha_1 q(s+h) + \alpha_2 q(s+h) = q(s+h).$$

- **(Eliminate third term).** Taylor expansion of $\mathcal{F}^{(3)}_{\mathcal{D}_m} q(s)$ at $s + h$ should have third order term $\mathbb{E}[(\delta - h)^3]$. That is,

$$\alpha_1 \mathbb{E}[(\delta_1 - h)^3] + \alpha_2 \mathbb{E}\left[\left(\tfrac{\delta_1+\delta_2}{2} - h\right)^3\right] = 0.$$

    This is equivalent to

$$\alpha_1 \mathbb{E}[(\delta_1 - h)^3] + \tfrac{\alpha_2}{4}\mathbb{E}\left[(\delta_1 - h)^3\right] = 0.$$

Therefore, $\alpha_1$ and $\alpha_2$ can be determined through the following linear system

$$\alpha_1 + \alpha_2 = 1$$
$$\alpha_1 + \tfrac{1}{4}\alpha_2 = 0.$$

The solution is $\alpha_1 = -\tfrac{1}{3}$ and $\alpha_2 = \tfrac{4}{3}$.

**For $k = 3$ order extrapolation,** consider the following

$$\mathcal{L}^{(3)}_{\mathcal{D}_m} q : s \mapsto \alpha'_1 \mathcal{F}^{(3)}_{\mathcal{D}_m} q(s) + \alpha'_2 \mathcal{F}^{(3)}_{\mathcal{D}_{2m}} q(s) + \alpha'_3 \mathcal{F}^{(3)}_{\mathcal{D}_{3m}} q(s).$$

We determine $\alpha'_1$, $\alpha'_2$ and $\alpha'_3$ by satisfying the following two conditions

- **(Affine).** Taylor expansion of $\mathcal{L}^{(3)}_{\mathcal{D}_m} q(s)$ at $s + h$ should have zero order term $q(x+h)$, i.e.

$$(\alpha'_1 + \alpha'_2 + \alpha'_3)q(x+h) = q(x+h).$$

- Taylor expansion of $\mathcal{L}_{\mathcal{D}_m}^{(3)} q(s)$ at $s + h$ should have 4th order term $\mathbb{E}[(\delta - h)^4]$. That is

$$\alpha_1' \, \mathbb{E}[(\delta_1 - h)^4] + \alpha_2' \, \mathbb{E}\left[\left(\tfrac{\delta_1 + \delta_2}{2} - h\right)^4\right] + \alpha_3' \, \mathbb{E}\left[\left(\tfrac{\delta_1 + \delta_2 + \delta_3}{3} - h\right)^4\right] = 0.$$

This is equivalent to

$$\left(\alpha_1' + \tfrac{\alpha_2'}{8} + \tfrac{\alpha_3'}{27}\right) \mathbb{E}[(\delta_1 - h)^4] = 0$$
$$\left(\tfrac{3}{8}\alpha_2' + \tfrac{2}{9}\alpha_2'\right)\left(\mathbb{E}[(\delta_1 - h)^2]\right)^2 = 0.$$

Therefore, $\alpha_1'$, $\alpha_2'$ and $\alpha_3'$ can be determined through the following linear system

$$\alpha_1' + \alpha_2' + \alpha_3' = 1$$
$$\alpha_1' + \tfrac{1}{8}\alpha_2' + \tfrac{1}{27}\alpha_3' = 0$$
$$\alpha_1' + \tfrac{3}{8}\alpha_2' + \tfrac{2}{9}\alpha_3' = 0.$$

The solution is $\alpha_1' = \tfrac{1}{12}$, $\alpha_2' = -\tfrac{4}{3}$ and $\alpha_3' = \tfrac{9}{4}$. Then consider the Taylor expansion of $\mathcal{L}_{\mathcal{D}_m}^{(3)} q(s)$ at $s + h$ with (2), we can

$$|\mathbb{E}[\mathcal{L}_{\mathcal{D}_m}^{(3)} q(s)] - q(s+h)| \lesssim \left|q^{(5)}(s+h)\,\mathbb{E}[(\delta - h)^5]\right| + \left|\mathbb{E}[q^{(6)}(\phi_\delta)(\delta - h)^6]\right| \lesssim \mathcal{O}((a_5 + a_6)m^{-3})$$

where the first inequality uses the fact that $\mathcal{L}_{\mathcal{D}_m}^{(3)}$ is an affine mapping and the last inequality uses Lemma 1. Therefore, $\mathcal{L}_{\mathcal{D}_m}^{(3)}$ is a 3rd-order extrapolation operator. We can expand it into

$$\mathcal{L}_{\mathcal{D}_m}^{(3)} q : s \mapsto \tfrac{1}{12}\left(-\tfrac{1}{3}\mathcal{L}_{\mathcal{D}_m}^{(2)} q(s) + \tfrac{4}{3}\mathcal{L}_{\mathcal{D}_{2m}}^{(2)} q(s)\right) - \tfrac{4}{3}\left(-\tfrac{1}{3}\mathcal{L}_{\mathcal{D}_{2m}}^{(2)} q(s) + \tfrac{4}{3}\mathcal{L}_{\mathcal{D}_{4m}}^{(2)} q(s)\right)$$
$$+ \tfrac{9}{4}\left(-\tfrac{1}{3}\mathcal{L}_{\mathcal{D}_{3m}}^{(2)} q(s) + \tfrac{4}{3}\mathcal{L}_{\mathcal{D}_{6m}}^{(2)} q(s)\right)$$
$$= \left(-\tfrac{1}{36}\mathcal{L}_{\mathcal{D}_m}^{(2)} + \tfrac{5}{9}\mathcal{L}_{\mathcal{D}_{2m}}^{(2)} - \tfrac{3}{4}\mathcal{L}_{\mathcal{D}_{3m}}^{(2)} - \tfrac{16}{9}\mathcal{L}_{\mathcal{D}_{4m}}^{(2)} + 3\mathcal{L}_{\mathcal{D}_{6m}}^{(2)}\right) q(s).$$

$\square$

**Lemma 2 (Variance Bound)** *Assume that* $q : \mathbb{R}^p \to \mathbb{R}^\ell$ *is in* $\mathcal{C}^4$ *and* $\mathcal{D}_m$ *is the distribution in Assumption 1. Suppose that the variance of* $q(s + \delta)$ *is bounded as*

$$\mathbb{E}[\|q(s + \delta) - \mathbb{E}[q(s + \delta)]\|_2^2] \leq \tfrac{V^2}{m} + C.$$

*Then the variance of extrapolation* $\mathcal{L}_{\mathcal{D}_m}^{(2)} q(s)$ *is upper bounded by*

$$\mathbb{E}\left[\left\|\mathcal{L}_{\mathcal{D}_m}^{(2)} q(s) - \mathbb{E}[\mathcal{L}_{\mathcal{D}_m}^{(2)} q(s)]\right\|_2^2\right] \leq 14(\tfrac{V^2}{m} + C).$$

**Proof:** Let us use the definition of $\mathcal{L}_{\mathcal{D}_m}^{(2)} q(s)$:

$$\mathbb{E}\left[\left\|\mathcal{L}_{\mathcal{D}_m}^{(2)} q(s) - \mathbb{E}[\mathcal{L}_{\mathcal{D}_m}^{(2)} q(s)]\right\|_2^2\right]$$
$$\leq \mathbb{E}\left[\left\|2q(s + \tfrac{\delta_1 + \delta_2}{2}) - \tfrac{q(s+\delta_1) + q(s+\delta_2)}{2} - \mathbb{E}\left[2q(s + \tfrac{\delta_1 + \delta_2}{2}) - \tfrac{q(s+\delta_1) + q(s+\delta_2)}{2}\right]\right\|_2^2\right]$$
$$\leq 3\,\mathbb{E}\left[\left\|2q(s + \tfrac{\delta_1 + \delta_2}{2}) - \mathbb{E}\left[2q(s + \tfrac{\delta_1 + \delta_2}{2})\right]\right\|_2^2\right] + 3\,\mathbb{E}\left[\left\|\tfrac{q(s+\delta_1)}{2} - \mathbb{E}\left[\tfrac{q(s+\delta_1)}{2}\right]\right\|_2^2\right]$$
$$+ 3\,\mathbb{E}\left[\left\|\tfrac{q(s+\delta_2)}{2} - \mathbb{E}\left[\tfrac{q(s+\delta_2)}{2}\right]\right\|_2^2\right]$$
$$\leq 12(\tfrac{V^2}{2m} + C) + \tfrac{3}{4}(\tfrac{V^2}{m} + C) + \tfrac{3}{4}(\tfrac{V^2}{m} + C)$$
$$= \tfrac{15V^2}{2m} + \tfrac{27C}{2}.$$

This completes the proof. $\square$

# D    Stationary Point Convergence Proofs from Section 3 (CSO)

In this section, we provide the convergence proofs for the CSO problem. We start by establishing some helpful lemmas in Appendix D.1. In Appendix D.2, we reanalyze the BSGD algorithm to obtain explicit bias and variance bounds, which are then useful when we analyze E-BSGD in Appendix D.3. Similarly, we reanalyze BSpiderBoost in Appendix D.4 and use the resulting bias and variance bounds for the analysis of E-BSpiderBoost in Appendix D.5.

Note that throughout our analyses, we define $\mathbb{E}^{t+1}[\cdot|t]$ as the expectation of randomness at time $t+1$ conditioning on the randomness until time $t$. When there is no ambiguity, we use $\mathbb{E}[\cdot]$ instead of $\mathbb{E}^{t+1}[\cdot|t]$.

## D.1    Helpful Lemmas

**Lemma 3 (Sufficient Decrease)** *Suppose Assumption 5 holds true and $\gamma \leq \frac{1}{2L_F}$ then*

$$\left\|\nabla F(\boldsymbol{x}^t)\right\|_2^2 \leq \frac{2(\mathbb{E}[F(\boldsymbol{x}^{t+1})]-F(\boldsymbol{x}^t))}{\gamma} + L_F\gamma\mathcal{E}_{var}^{t+1} + \mathcal{E}_{bias}^{t+1},$$

*where $\mathbb{E}[\cdot]$ denote conditional expectation over the randomness at time $t$ conditioned on all of the past randomness until time $t$.*

**Proof:** In this proof, we use $\mathbb{E}[\cdot]$ to denote conditional expectation over the randomness at time $t$ conditioned on all the past randomness until time $t$.

Let us expand $F(\boldsymbol{x}^{t+1})$ and apply the $L_F$-smoothness of $F$

$$\mathbb{E}[F(\boldsymbol{x}^{t+1})] \leq F(\boldsymbol{x}^t) - \gamma \, \mathbb{E}[\langle \nabla F(\boldsymbol{x}^t), G^{t+1}\rangle] + \frac{L_F\gamma^2}{2} \mathbb{E}[\|G^{t+1}\|_2^2].$$

Since $\mathbb{E}[\|G^{t+1}\|_2^2] = \mathbb{E}[\|G^{t+1} - \mathbb{E}[G^{t+1}]\|_2^2] + \|\mathbb{E}[G^{t+1}]\|_2^2 = \mathcal{E}_{var}^{t+1} + \|\mathbb{E}[G^{t+1}]\|_2^2$, then

$$\mathbb{E}[F(\boldsymbol{x}^{t+1})] \leq F(\boldsymbol{x}^t) - \gamma \, \mathbb{E}[\langle \nabla F(\boldsymbol{x}^t), G^{t+1}\rangle] + \frac{L_F\gamma^2}{2}(\mathcal{E}_{var}^{t+1} + \|\mathbb{E}[G^{t+1}]\|_2^2).$$

Expand the middle term with

$$-\gamma \, \mathbb{E}[\langle \nabla F(\boldsymbol{x}^t), G^{t+1}\rangle] = -\frac{\gamma}{2}\left\|\nabla F(\boldsymbol{x}^t)\right\|_2^2 - \frac{\gamma}{2}\left\|\mathbb{E}[G^{t+1}]\right\|_2^2 + \frac{\gamma}{2}\left\|\nabla F(\boldsymbol{x}^t) - \mathbb{E}[G^{t+1}]\right\|_2^2$$

$$= -\frac{\gamma}{2}\left\|\nabla F(\boldsymbol{x}^t)\right\|_2^2 - \frac{\gamma}{2}\left\|\mathbb{E}[G^{t+1}]\right\|_2^2 + \frac{\gamma}{2}\mathcal{E}_{bias}^{t+1}.$$

Combine with the inequality

$$\mathbb{E}[F(\boldsymbol{x}^{t+1})] \leq F(\boldsymbol{x}^t) - \frac{\gamma}{2}\left\|\nabla F(\boldsymbol{x}^t)\right\|_2^2 - \frac{\gamma}{2}(1 - L_F\gamma)\left\|\mathbb{E}[G^{t+1}]\right\|_2^2 + \frac{\gamma}{2}\mathcal{E}_{bias}^{t+1} + \frac{L_F\gamma^2}{2}\mathcal{E}_{var}^{t+1}.$$

By taking $\gamma \leq \frac{1}{2L_F}$, we have that

$$\mathbb{E}[F(\boldsymbol{x}^{t+1})] \leq F(\boldsymbol{x}^t) - \frac{\gamma}{2}\left\|\nabla F(\boldsymbol{x}^t)\right\|_2^2 - \frac{\gamma}{4}\left\|\mathbb{E}[G^{t+1}]\right\|_2^2 + \frac{\gamma}{2}\mathcal{E}_{bias}^{t+1} + \frac{L_F\gamma^2}{2}\mathcal{E}_{var}^{t+1}.$$

Re-arranging the terms we get the desired inequality.  $\square$

A consequence of Lemma 3 is the following result.

**Lemma 4 (Descent Lemma)** *Suppose Assumption 5 holds true. By taking $\gamma \leq \frac{1}{2L_F}$, we have,*

$$\frac{1}{T}\sum_{t=0}^{T-1} \mathbb{E}\left[\|\nabla F(\boldsymbol{x}^t)\|_2^2\right] + \frac{1}{2}\frac{1}{T}\sum_{t=0}^{T-1} \mathbb{E}\left[\|\mathbb{E}^t[G^{t+1}|t]\|_2^2\right]$$

$$\leq \frac{2(F(\boldsymbol{x}^0)-F^\star)}{\gamma T} + \frac{1}{T}\sum_{t=0}^{T-1}\mathbb{E}[\mathcal{E}_{bias}^{t+1}] + \frac{L_F\gamma}{T}\sum_{t=0}^{T-1}\mathbb{E}[\mathcal{E}_{var}^{t+1}]$$

*where the expectation is taken over all randomness from $t=0$ to $T$.*

**Proof:** We denote the conditional expectation at time $t$ in the descent lemma (Lemma 3) as $\mathbb{E}^{t+1}[\cdot|t]$ which conditions on all past randomness until time $t$. Then the descent lemma can be written as

$$\mathbb{E}^{t+1}[F(\boldsymbol{x}^{t+1})|t] \leq F(\boldsymbol{x}^t) - \frac{\gamma}{2}\left\|\nabla F(\boldsymbol{x}^t)\right\|_2^2 - \frac{\gamma}{4}\left\|\mathbb{E}^{t+1}[G^{t+1}|t]\right\|_2^2 + \frac{\gamma}{2}\mathbb{E}^{t+1}[\mathcal{E}_{bias}^{t+1}|t] + \frac{L_F\gamma^2}{2}\mathbb{E}^{t+1}[\mathcal{E}_{var}^{t+1}|t].$$

If we additionally consider the randomness at time $t-1$, and apply $\mathbb{E}^t[\cdot|t-1]$ to both sides

$$\mathbb{E}^t\left[\mathbb{E}^{t+1}[F(\boldsymbol{x}^{t+1})|t]|t-1\right] \leq \mathbb{E}^t[F(\boldsymbol{x}^t)|t-1] - \frac{\gamma}{2}\mathbb{E}[\|\nabla F(\boldsymbol{x}^t)\|_2^2|t-1]$$

$$- \mathbb{E}^t\left[\frac{\gamma}{4}\left\|\mathbb{E}^{t+1}[G^{t+1}|t]\right\|_2^2|t-1\right] + \frac{\gamma}{2}\mathbb{E}^t\left[\mathbb{E}^{t+1}[\mathcal{E}_{\text{bias}}^{t+1}|t]|t-1\right]$$

$$+ \frac{L_F\gamma^2}{2}\mathbb{E}^{t-1]}\left[\mathbb{E}^{t+1}[\mathcal{E}_{\text{var}}^{t+1}|t]|t-1\right].$$

By the law of iterative expectations, we have $\mathbb{E}^t\left[\mathbb{E}^{t+1}[\cdot|t]|t-1\right] = \mathbb{E}^t\,\mathbb{E}^{t+1}\,[\cdot|t-1]$

$$\mathbb{E}^t[\mathbb{E}^{t+1}\left[F(\boldsymbol{x}^{t+1})|t-1\right]] \leq \mathbb{E}^t[F(\boldsymbol{x}^t)|t-1] - \frac{\gamma}{2}\mathbb{E}[\|\nabla F(\boldsymbol{x}^t)\|_2^2|t-1]$$

$$- \mathbb{E}^t\left[\frac{\gamma}{4}\left\|\mathbb{E}^{t+1}[G^{t+1}|t]\right\|_2^2|t-1\right] + \frac{\gamma}{2}\mathbb{E}^t\left[\mathbb{E}^{t+1}\left[\mathcal{E}_{\text{bias}}^{t+1}|t-1\right]\right]$$

$$+ \frac{L_F\gamma^2}{2}\mathbb{E}^t[\mathbb{E}^t\left[\mathcal{E}_{\text{var}}^{t+1}|t-1\right]].$$

Similarly, we can apply $\mathbb{E}^{t-1}[\cdot|t-2]$, $\mathbb{E}^{t-2}[\cdot|t-3]$, …, $\mathbb{E}^2[\cdot|1]$ and finally $\mathbb{E}^1[\cdot]$

$$\mathbb{E}^1\ldots[\mathbb{E}^{t+1}\left[F(\boldsymbol{x}^{t+1})\right]] \leq \mathbb{E}^1\ldots[\mathbb{E}^t[F(\boldsymbol{x}^t)]] - \frac{\gamma}{2}\mathbb{E}^1\ldots[\mathbb{E}^t[\|\nabla F(\boldsymbol{x}^t)\|_2^2]]$$

$$- \mathbb{E}^1\ldots[\mathbb{E}^t[\frac{\gamma}{4}\left\|\mathbb{E}^{t+1}[G^{t+1}|t]\right\|_2^2]] + \frac{\gamma}{2}\mathbb{E}^1\ldots[\mathbb{E}^{t+1}\left[\mathcal{E}_{\text{bias}}^{t+1}\right]]$$

$$+ \frac{L_F\gamma^2}{2}\mathbb{E}^1\ldots[\mathbb{E}^t\left[\mathcal{E}_{\text{var}}^{t+1}\right]].$$

Now that both sides of the inequality have no randomness, we can simplify the notation by applying $\mathbb{E}^{t+1}\ldots[\mathbb{E}^t[\cdot]]$ to both sides and by denoting

$$\mathbb{E}[\cdot] = \mathbb{E}^1\ldots[\mathbb{E}^{+1}[\cdot]].$$

Then the descent lemma becomes

$$\mathbb{E}[F(\boldsymbol{x}^{t+1})] \leq \mathbb{E}[F(\boldsymbol{x}^t)] - \frac{\gamma}{2}\mathbb{E}[\|\nabla F(\boldsymbol{x}^t)\|_2^2] - \frac{\gamma}{4}\mathbb{E}[\|\mathbb{E}^{t+1}[G^{t+1}|t]\|_2^2] + \frac{\gamma}{2}\mathbb{E}[\mathcal{E}_{\text{bias}}^{t+1}] + \frac{L_F\gamma^2}{2}\mathbb{E}[\mathcal{E}_{\text{var}}^{t+1}].$$

Now we can sum the descent lemmas from $t=0$ to $T-1$

$$\sum_{t=0}^{T-1}\mathbb{E}[F(\boldsymbol{x}^{t+1})] \leq \sum_{t=0}^{T-1}\mathbb{E}[F(\boldsymbol{x}^t)] - \frac{\gamma}{2}\sum_{t=0}^{T-1}\mathbb{E}\left[\|\nabla F(\boldsymbol{x}^t)\|_2^2\right]$$

$$- \frac{\gamma}{4}\sum_{t=0}^{T-1}\mathbb{E}\left[\|\mathbb{E}^{t+1}[G^{t+1}|t]\|_2^2\right] + \frac{\gamma}{2}\sum_{t=0}^{T-1}\mathbb{E}[\mathcal{E}_{\text{bias}}^{t+1}] + \frac{L_F\gamma^2}{2}\sum_{t=0}^{T-1}\mathbb{E}[\mathcal{E}_{\text{var}}^{t+1}].$$

After simplification and division by $T$, we get

$$\frac{1}{T}\sum_{t=0}^{T-1}\mathbb{E}\left[\|\nabla F(\boldsymbol{x}^t)\|_2^2\right] + \frac{1}{2}\frac{1}{T}\sum_{t=0}^{T-1}\mathbb{E}\left[\|\mathbb{E}^{t+1}[G^{t+1}|t]\|_2^2\right]$$

$$\leq \frac{2(\mathbb{E}[F(\boldsymbol{x}^T)] - \mathbb{E}[F(\boldsymbol{x}^0)])}{\gamma T} + \frac{\gamma}{2}\frac{1}{T}\sum_{t=0}^{T-1}\mathbb{E}[\mathcal{E}_{\text{bias}}^{t+1}] + \frac{L_F\gamma^2}{2}\frac{1}{T}\sum_{t=0}^{T-1}\mathbb{E}[\mathcal{E}_{\text{var}}^{t+1}]$$

$$\leq \frac{2(\mathbb{E}[F(\boldsymbol{x}^T)] - F^\star)}{\gamma T} + \frac{1}{T}\sum_{t=0}^{T-1}\mathbb{E}[\mathcal{E}_{\text{bias}}^{t+1}] + \frac{L_F\gamma}{T}\sum_{t=0}^{T-1}\mathbb{E}[\mathcal{E}_{\text{var}}^{t+1}].$$

$\square$

The following corollary is a consequence of Proposition 1.

**Corollary 1** *Assume $\nabla f_\xi$ in CSO satisfies*

$$a_l := \sup_{\boldsymbol{x}}\sup_{\xi}\left\|\nabla^{l+1}f_\xi(\boldsymbol{x})\right\|_2 < \infty, \qquad l = 1, 2, 3, 4.$$

*Let's further assume that the higher order moments of $g_\eta(\cdot)$ are bounded,*

$$\sigma_k = \sup_{\boldsymbol{x}}\sup_{\xi}\mathbb{E}_{\eta|\xi}\left[\sum_{i=1}^p\left[g_\eta(\boldsymbol{x}) - \mathbb{E}_{\eta|\xi}[g_\eta(\boldsymbol{x})]\right]_i^k\right] < \infty, \qquad k = 1, 2, 3, 4$$

*where $[\cdot]_i$ refers to the $i$-th coordinate of a vector. Consider the $\mathcal{L}_{\mathcal{D}_{g,\xi}^{t+1}}^{(2)}\nabla f_\xi(0)$ defined in (6), then*

$$\left\|\mathbb{E}\left[\mathcal{L}_{\mathcal{D}_{g,\xi}^{t+1}}^{(2)}\nabla f_\xi(0)\right] - \nabla f_\xi(\mathbb{E}_{\eta|\xi}[g_\eta(\boldsymbol{x}^t)])\right\|_2^2 \leq \frac{C_e^2}{m^4} \qquad \forall \xi,$$

*where $C_e^2(f;g) := \left(\frac{8a_3\sigma_3 + 18a_4\sigma_2^2 + 5a_4\sigma_4}{96}\right)^2$.*

**Proof:** The Proposition 1 gives the following upper bound

$$\left\|\mathbb{E}\left[\mathcal{L}^{(2)}_{\mathcal{D}^{t+1}_{g,\xi}}\nabla f_\xi(0)\right] - \nabla f_\xi(\mathbb{E}_{\eta|\xi}[g_\eta(\boldsymbol{x}^t)])\right\|^2_2 \leq \left(\frac{4a_3\sigma_3+9a_4\sigma_2^2}{48m^2} + \frac{5a_4}{96}\frac{\sigma_4-3\sigma_2^2}{m^3}\right)^2.$$

For simplicity, we can relax the upper bound to

$$\left\|\mathbb{E}\left[\mathcal{L}^{(2)}_{\mathcal{D}^{t+1}_{g,\xi}}\nabla f_\xi(0)\right] - \nabla f_\xi(\mathbb{E}_{\eta|\xi}[g_\eta(\boldsymbol{x}^t)])\right\|^2_2 \leq \frac{1}{m^4}\left(\frac{8a_3\sigma_3+18a_4\sigma_2^2+5a_4\sigma_4}{96}\right)^2.$$

$\square$

## D.2 Convergence of BSGD

In this section, we reanalyze the BSGD algorithm of [16] to obtain bounds on bias and variance of its gradient estimates. Theorem 6 shows that BSGD achieves an $\mathcal{O}(\varepsilon^{-6})$ sample complexity.

**Lemma 5 (Bias and Variance of BSGD)** *The bias and variance of BSGD are*

$$\mathcal{E}^{t+1}_{bias} \leq \frac{\sigma^2_{bias}}{m}, \quad \mathcal{E}^{t+1}_{var} \leq \frac{\sigma^2_{in}}{m} + \sigma^2_{out}$$

*where $\sigma^2_{in} = \zeta_g^2 C_f^2 + \sigma_g^2 C_g^2 L_f^2$, $\sigma^2_{out} = C_F^2$, and $\sigma^2_{bias} = \sigma_g^2 C_g^2 L_f^2$.*

**Proof:** Denote $G^{t+1} = G^{t+1}_{\text{BSGD}}$ (5) and denote $\mathbb{E}[\cdot]$ as the conditional expectation $\mathbb{E}^{t+1}[\cdot|t]$ which conditions on all past randomness until time $t$. Note that the $\nabla g_{\tilde{\eta}}$ can be estimated without bias, i.e.

$$\mathbb{E}_{\tilde{\eta}|\xi}\left[\frac{1}{m}\sum_{\tilde{\eta}\in\tilde{H}_\xi}\nabla g_{\tilde{\eta}}(\boldsymbol{x})\right] = \mathbb{E}_{\tilde{\eta}|\xi}\left[\nabla g_{\tilde{\eta}}(\boldsymbol{x})\right],$$

Then let's first look at the bias of BSGD

$$\begin{aligned}
\mathcal{E}^{t+1}_{\text{bias}} &= \left\|\nabla F(\boldsymbol{x}^{t+1}) - \mathbb{E}[G^{t+1}]\right\|^2_2 \\
&= \left\|\mathbb{E}_\xi\left[(\mathbb{E}_{\tilde{\eta}|\xi}[\nabla g_{\tilde{\eta}}(\boldsymbol{x}^t)])^\top\left(\nabla f_\xi(\mathbb{E}_{\eta|\xi}[g_\eta(\boldsymbol{x}^t)]) - \mathbb{E}_{\eta|\xi}[\nabla f_\xi(\frac{1}{m}\sum_{\eta\in H_\xi}g_\eta(\boldsymbol{x}^t))]\right)\right]\right\|^2_2 \\
&\leq C_g^2\,\mathbb{E}_\xi\left[\left\|\nabla f_\xi(\mathbb{E}_{\eta|\xi}[g_\eta(\boldsymbol{x}^t)]) - \mathbb{E}_{\eta|\xi}[\nabla f_\xi(\frac{1}{m}\sum_{\eta\in H_\xi}g_\eta(\boldsymbol{x}^t))]\right\|^2_2\right] \\
&\leq C_g^2 L_f^2\,\mathbb{E}_\xi\left[\mathbb{E}_{\eta|\xi}\left[\left\|\mathbb{E}_{\eta|\xi}[g_\eta(\boldsymbol{x}^t)] - \frac{1}{m}\sum_{\eta\in H_\xi}g_\eta(\boldsymbol{x}^t)\right\|^2_2\right]\right] \\
&\leq \frac{C_g^2 L_f^2}{m}\,\mathbb{E}_\xi\left[\mathbb{E}_{\eta|\xi}\left[\left\|\mathbb{E}_{\eta|\xi}[g_\eta(\boldsymbol{x}^t)] - g_\eta(\boldsymbol{x}^t)\right\|^2_2\right]\right] \\
&= \frac{\sigma_g^2 C_g^2 L_f^2}{m} = \frac{\sigma^2_{\text{bias}}}{m}.
\end{aligned}$$

For the first inequality, we take the expectation outside the norm and bound $\nabla g_{\tilde{\eta}}$ with $C_g$.

On the other hand, the variance of BSGD can be decomposed into inner variance and outer variance

$$\begin{aligned}
\mathcal{E}^{t+1}_{\text{var}} &= \mathbb{E}_\xi[\mathbb{E}_{\eta|\xi,\tilde{\eta}|\xi}[\|G^{t+1} - \mathbb{E}_\xi[\mathbb{E}_{\eta|\xi,\tilde{\eta}|\xi}[G^{t+1}]]\|^2_2]] \\
&= \mathbb{E}_\xi[\mathbb{E}_{\eta|\xi,\tilde{\eta}|\xi}[\|(G^{t+1} - \mathbb{E}_{\eta|\xi,\tilde{\eta}|\xi}[G^{t+1}]) + (\mathbb{E}_{\eta|\xi,\tilde{\eta}|\xi}[G^{t+1}] - \mathbb{E}_\xi[\mathbb{E}_{\eta|\xi,\tilde{\eta}|\xi}[G^{t+1}]])\|^2_2]] \\
&= \underbrace{\mathbb{E}_\xi[\mathbb{E}_{\eta|\xi,\tilde{\eta}|\xi}[\|G^{t+1} - \mathbb{E}_{\eta|\xi,\tilde{\eta}|\xi}[G^{t+1}]\|^2_2]]}_{\text{Inner variance}} + \underbrace{\mathbb{E}_\xi[\|\mathbb{E}_{\eta|\xi,\tilde{\eta}|\xi}[G^{t+1}] - \mathbb{E}_\xi[\mathbb{E}_{\eta|\xi,\tilde{\eta}|\xi}[G^{t+1}]]\|^2_2]}_{\text{Outer variance}}.
\end{aligned}$$

The inner variance is bounded as follows

$$\mathbb{E}_\xi[\mathbb{E}_{\eta|\xi,\tilde{\eta}|\xi}[\|G^{t+1} - \mathbb{E}_\xi[\mathbb{E}_{\eta|\xi,\tilde{\eta}|\xi}[G^{t+1}]\|_2^2]]$$

$$= \mathbb{E}_\xi\left[\mathbb{E}_{\eta|\xi,\tilde{\eta}|\xi}\left[\left\|(\tfrac{1}{m}\sum_{\tilde{\eta}\in\tilde{H}_\xi}\nabla g_{\tilde{\eta}}(\boldsymbol{x}^t) - \mathbb{E}_{\tilde{\eta}|\xi}[\nabla g_{\tilde{\eta}}(\boldsymbol{x}^t)])^\top \nabla f_\xi(\tfrac{1}{m}\sum_{\eta\in H_\xi}g_\eta(\boldsymbol{x}^t))\right\|_2^2\right]\right]$$

$$\quad + \mathbb{E}_\xi\left[\mathbb{E}_{\eta|\xi}\left[\left\|(\mathbb{E}_{\tilde{\eta}|\xi}[\nabla g_{\tilde{\eta}}(\boldsymbol{x}^t)])^\top(\nabla f_\xi(\tfrac{1}{m}\sum_{\eta\in H_\xi}g_\eta(\boldsymbol{x}^t)) - \mathbb{E}_{\eta|\xi}[\nabla f_\xi(\tfrac{1}{m}\sum_{\eta\in H_\xi}g_\eta(\boldsymbol{x}^t))])\right\|_2^2\right]\right]$$

$$\leq C_f^2\,\mathbb{E}_\xi\left[\mathbb{E}_{\tilde{\eta}|\xi}\left[\left\|\tfrac{1}{m}\sum_{\tilde{\eta}\in\tilde{H}_\xi}\nabla g_{\tilde{\eta}}(\boldsymbol{x}^t) - \mathbb{E}_{\tilde{\eta}|\xi}[\nabla g_{\tilde{\eta}}(\boldsymbol{x}^t)]\right\|_2^2\right]\right]$$

$$\quad + C_g^2\,\mathbb{E}_\xi\left[\mathbb{E}_{\eta|\xi}\left[\left\|\nabla f_\xi(\tfrac{1}{m}\sum_{\eta\in H_\xi}g_\eta(\boldsymbol{x}^t)) - \mathbb{E}_{\eta|\xi}[\nabla f_\xi(\tfrac{1}{m}\sum_{\eta\in H_\xi}g_\eta(\boldsymbol{x}^t))]\right\|_2^2\right]\right]$$

$$= C_f^2\,\mathbb{E}_\xi\left[\mathbb{E}_{\tilde{\eta}|\xi}\left[\left\|\tfrac{1}{m}\sum_{\tilde{\eta}\in\tilde{H}_\xi}\nabla g_{\tilde{\eta}}(\boldsymbol{x}^t) - \mathbb{E}_{\tilde{\eta}|\xi}[\nabla g_{\tilde{\eta}}(\boldsymbol{x}^t)]\right\|_2^2\right]\right]$$

$$\quad + C_g^2\,\mathbb{E}_\xi\left[\mathbb{E}_{\eta|\xi}\left[\left\|\nabla f_\xi(\tfrac{1}{m}\sum_{\eta\in H_\xi}g_\eta(\boldsymbol{x}^t)) - \nabla f_\xi(\mathbb{E}_\eta[g_\eta(\boldsymbol{x}^t)])\right\|_2^2\right]\right]$$

$$\quad - C_g^2\,\mathbb{E}_\xi\left[\left\|\nabla f_\xi(\mathbb{E}_\eta[g_\eta(\boldsymbol{x}^t)]) - \mathbb{E}_{\eta|\xi}[\nabla f_\xi(\tfrac{1}{m}\sum_{\eta\in H_\xi}g_\eta(\boldsymbol{x}^t))]\right\|_2^2\right]$$

$$\leq \tfrac{\zeta_g^2 C_f^2}{m} + C_g^2 L_f\,\mathbb{E}_\xi\left[\mathbb{E}_{\eta|\xi}\left[\left\|\tfrac{1}{m}\sum_{\eta\in H_\xi}g_\eta(\boldsymbol{x}^t) - \mathbb{E}_{\eta|\xi}[g_\eta(\boldsymbol{x}^t)]\right\|_2^2\right]\right]$$

$$\leq \tfrac{C_f^2\zeta_g^2}{m} + \tfrac{C_g^2 L_f}{m}\,\mathbb{E}_\xi\left[\mathbb{E}_{\eta|\xi}\left[\left\|g_\eta(\boldsymbol{x}^t) - \mathbb{E}_{\eta|\xi}[g_\eta(\boldsymbol{x}^t)]\right\|_2^2\right]\right]$$

$$\leq \tfrac{\zeta_g^2 C_f^2 + \sigma_g^2 C_g^2 L_f^2}{m} = \tfrac{\sigma_{in}^2}{m}.$$

The outer variance is independent of the inner batch size and can be bounded by

$$\mathbb{E}_\xi[\|\mathbb{E}_{\eta|\xi,\tilde{\eta}|\xi}[G^{t+1}] - \mathbb{E}_\xi[\mathbb{E}_{\eta|\xi,\tilde{\eta}|\xi}[G^{t+1}]]\|_2^2] \leq \mathbb{E}_\xi[\|\mathbb{E}_{\eta|\xi,\tilde{\eta}|\xi}[G^{t+1}]\|_2^2] \leq C_f^2 C_g^2 = C_F^2 = \sigma_{out}^2$$

Therefore, the variance is bounded as follows

$$\mathcal{E}_{var}^{t+1} \leq \tfrac{\sigma_{in}^2}{m} + \sigma_{out}^2.$$

This completes the proof. $\qquad\square$

**Theorem 6 (BSGD Convergence)** *Consider the* (CSO) *problem. Suppose Assumptions 3, 4, 5 holds true. Let step size $\gamma \leq 1/(2L_F)$. Then for BSGD, $\boldsymbol{x}^s$ picked uniformly at random among $\{\boldsymbol{x}^t\}_{t=0}^{T-1}$ satisfies: $\mathbb{E}[\|\nabla F(\boldsymbol{x}^s)\|_2^2] \leq \varepsilon^2$, for nonconvex $F$, if the inner batch size $m = \Omega\left(\sigma_{bias}^2\varepsilon^{-2}\right)$ and the number of iterations $T = \Omega\left((F(\boldsymbol{x}^0) - F^\star)L_F(\sigma_{in}^2/m + \sigma_{out}^2)\varepsilon^{-4}\right)$, where $\sigma_{in}^2 = \zeta_g^2 C_f^2 + \sigma_g^2 C_g^2 L_f^2$, $\sigma_{out}^2 = C_F^2$, and $\sigma_{bias}^2 = \sigma_g^2 C_g^2 L_f^2$.*

**Proof:** Denote $G^{t+1} = G_{BSGD}^{t+1}$ (1). Using descent lemma (Lemma 4) and bias-variance bounds of BSGD (Lemma 5)

$$\tfrac{1}{T}\sum_{t=0}^{T-1}\mathbb{E}\left[\|\nabla F(\boldsymbol{x}^t)\|_2^2\right] + \tfrac{1}{2}\tfrac{1}{T}\sum_{t=0}^{T-1}\mathbb{E}\left[\|\mathbb{E}^{t+1}[G^{t+1}|t]\|_2^2\right]$$
$$\leq \tfrac{2(\mathbb{E}[F(\boldsymbol{x}^T)] - F^\star)}{\gamma T} + L_F\gamma(\tfrac{\sigma_{in}^2}{m} + \sigma_{out}^2) + \tfrac{\sigma_{bias}^2}{m}$$

Then we can minimize the right-hand size by optimizing $\gamma$ to

$$\gamma = \sqrt{\tfrac{2(F(\boldsymbol{x}^0) - F^\star)}{L_F(\sigma_{in}^2/m + \sigma_{out}^2)T}}$$

which is smaller than the bound of step size $\gamma \leq \tfrac{1}{2L_F}$ if $T$ is greater than the following constant which does not rely on the target precision $\varepsilon$

$$T \geq \tfrac{8L_F(F(\boldsymbol{x}^0) - F^\star)}{\sigma_{in}^2/m + \sigma_{out}^2}.$$

Then the upper bound of gradient becomes

$$\frac{1}{T}\sum_{t=0}^{T-1}\mathbb{E}\left[\|\nabla F(\boldsymbol{x}^t)\|_2^2\right] \leq \sqrt{\frac{2(F(\boldsymbol{x}^0)-F^\star)L_F(\sigma_{\text{in}}^2/m+\sigma_{\text{out}}^2)}{T}} + \frac{\sigma_{\text{bias}}^2}{m}.$$

By taking inner batch size of at least

$$m \geq \frac{\sigma_{\text{bias}}^2}{\varepsilon^2},$$

and iteration $T$ greater than

$$T \geq \frac{2(F(\boldsymbol{x}^0)-F^\star)L_F(\sigma_{\text{in}}^2/m+\sigma_{\text{out}}^2)}{\varepsilon^4},$$

we have that

$$\frac{1}{T}\sum_{t=0}^{T-1}\mathbb{E}\left[\|\nabla F(\boldsymbol{x}^t)\|_2^2\right] \leq 2\varepsilon^2.$$

By picking $\boldsymbol{x}^s$ uniformly at random among $\{\boldsymbol{x}^t\}_{t=0}^{T-1}$, we get the desired guarantee. □

The resulting sample complexity of BSGD to get to an $\varepsilon$-stationary point is $\mathcal{O}(\varepsilon^{-6})$.

### D.3  Convergence of E-BSGD

In this section, we analyze the sample complexity of Algorithm 2 (E-BSGD) for the CSO problem.

**Lemma 6 (Bias and Variance of E-BSGD)** *The bias and variance of E-BSGD are*

$$\mathcal{E}_{bias}^{t+1} \leq \frac{\tilde{\sigma}_{bias}^2}{m^4}, \quad \mathcal{E}_{var}^{t+1} \leq 14(\frac{\sigma_{in}^2}{m}+\sigma_{out}^2)$$

*where $\sigma_{in}^2 = \zeta_g^2 C_f^2 + \sigma_g^2 C_g^2 L_f^2$, $\sigma_{out}^2 = C_F^2$, and $\tilde{\sigma}_{bias}^2 = C_g^2 C_e^2$ with $C_e^2$ defined in Corollary 1.*

**Proof:** Denote $G^{t+1} = G_{\text{E-BSGD}}^{t+1}$ (7). Like previously (Lemma 5), let $\mathbb{E}[\cdot]$ denote the conditional expectation $\mathbb{E}^{t+1}[\cdot|t]$ which conditions on all past randomness until time $t$. In E-BSGD, we apply extrapolation to $\nabla f_\xi(\cdot)$. The bias can be estimated with the help of Corollary 1 as

$$\begin{aligned}
\mathcal{E}_{\text{bias}}^{t+1} &= \left\|\nabla F(\boldsymbol{x}^{t+1}) - \mathbb{E}[G^{t+1}]\right\|_2^2 \\
&\leq C_g^2 \, \mathbb{E}_\xi\left[\left\|\nabla f_\xi(\mathbb{E}_{\eta|\xi}[g_\eta(\boldsymbol{x}^t)]) - \mathbb{E}\left[\mathcal{L}_{\mathcal{D}_{\boldsymbol{g},\xi}^{t+1}}^{(2)}\nabla f_\xi(0)\right]\right\|_2^2\right] \\
&\leq \frac{C_g^2 C_e^2}{m^4}.
\end{aligned}$$

Since the variance of BSGD in Lemma 5 is upper bounded by $\frac{\sigma_{\text{in}}^2}{m}+\sigma_{\text{out}}^2$, then Lemma 2 gives

$$\mathcal{E}_{\text{var}}^{t+1} \leq 14(\sigma_{\text{in}}^2/m+\sigma_{\text{out}}^2).$$

This proves the claimed bounds. □

**Theorem 3** *[E-BSGD Convergence] Consider the* (CSO) *problem. Suppose Assumptions 3, 4, 5, 6 hold true and $L_F, C_F, \tilde{L}_F, C_g, F^\star$ are constants and $C_e(f;g) := \frac{8a_3\sigma_3+18a_4\sigma_2^2+5a_4\sigma_4}{96}$ defined in Corollary 1 are associated with second order extrapolation in the CSO problem. Let step size $\gamma \leq 1/(2L_F)$. Then the output $\boldsymbol{x}^s$ of E-BSGD (Algorithm 2) satisfies: $\mathbb{E}[\|\nabla F(\boldsymbol{x}^s)\|_2^2] \leq \varepsilon^2$, for nonconvex $F$, if the inner batch size $m = \Omega(C_e C_g \varepsilon^{-1/2})$, and the number of iterations*

$$T = \Omega(L_F(F(\boldsymbol{x}^0)-F^\star)(\tilde{L}_F^2/m+C_F^2)\varepsilon^{-4}).$$

**Proof:** The proof is very similar to Theorem 6. Denote $G^{t+1} = G_{\text{E-BSGD}}^{t+1}$ (7). Using descent lemma (Lemma 4) and bias-variance bounds of E-BSGD (Lemma 6)

$$\begin{aligned}
\frac{1}{T}\sum_{t=0}^{T-1}\mathbb{E}\left[\|\nabla F(\boldsymbol{x}^t)\|_2^2\right] + \frac{1}{2}\frac{1}{T}\sum_{t=0}^{T-1}\mathbb{E}\left[\left\|\mathbb{E}^{t+1}[G^{t+1}|t]\right\|_2^2\right] & \\
&\hspace{-6em}\leq \frac{2(\mathbb{E}[F(\boldsymbol{x}^T)]-F^\star)}{\gamma T} + 14L_F\gamma(\frac{\sigma_{\text{in}}^2}{m}+\sigma_{\text{out}}^2) + \frac{C_g^2 C_e^2}{m^4}.
\end{aligned}$$

Then we optimize $\gamma$ to

$$\gamma = \sqrt{\frac{(F(\boldsymbol{x}^0)-F^\star)}{7L_F(\sigma_{\text{in}}^2/m+\sigma_{\text{out}}^2)T}}$$

which is smaller than the bound of step size $\gamma \le \frac{1}{2L_F}$ if $T$ is greater than the following constant which does not rely on the target precision $\varepsilon$

$$T \ge \frac{4L_F(F(\boldsymbol{x}^0)-F^\star)}{7(\sigma_{\text{in}}^2/m+\sigma_{\text{out}}^2)}.$$

Then the gradient norm has the following upper bound.

$$\frac{1}{T}\sum_{t=0}^{T-1}\|\nabla F(\boldsymbol{x}^t)\|_2^2 \le 4\sqrt{\frac{7(F(\boldsymbol{x}^0)-F^\star)L_F(\sigma_{\text{in}}^2+\sigma_{\text{out}}^2)}{T}} + \frac{\tilde\sigma_{\text{bias}}^2}{m^4}.$$

In order to reach $\varepsilon$-stationary point, i.e.

$$\frac{1}{T}\sum_{t=0}^{T-1}\|\nabla F(\boldsymbol{x}^t)\|_2^2 \le \varepsilon^2,$$

we can enforce

$$4\sqrt{\frac{7(F(\boldsymbol{x}^0)-F^\star)L_F(\sigma_{\text{in}}^2/m+\sigma_{\text{out}}^2)}{T}} \le \varepsilon^2, \quad \frac{C_g^2 C_e^2}{m^4} \le \varepsilon^2.$$

By taking inner batch size of at least

$$m = \Omega(\tilde\sigma_{\text{bias}}^{1/2}\varepsilon^{-1/2}),$$

and iteration $T$ greater than

$$T \ge \frac{112(F(\boldsymbol{x}^0)-F^\star)L_F(\frac{\sigma_{\text{in}}^2}{m}+\sigma_{\text{out}}^2)}{\varepsilon^4},$$

we have that

$$\frac{1}{T}\sum_{t=0}^{T-1}\|\nabla F(\boldsymbol{x}^t)\|_2^2 \le 3\varepsilon^2.$$

By picking $\boldsymbol{x}^s$ uniformly at random among $\{\boldsymbol{x}^t\}_{t=0}^{T-1}$, we get the desired guarantee. $\qquad\square$

### D.4 Convergence of BSpiderBoost

In this section, we reanalyze the BSpiderBoost algorithm of [16] to obtain bounds on bias and variance of its gradient estimates. Theorem 6 shows that BSpiderBoost achieves an $\mathcal{O}(\varepsilon^{-5})$ sample complexity.

Let $G_{\text{BSB}}^{t+1}$ as the BSpiderBoost gradient estimate

$$G_{\text{BSB}}^{t+1} = \begin{cases} G_{\text{BSB}}^t + \frac{1}{B_2}\sum_{\xi\in\mathcal{B}_2}(G_{\text{BSGD}}^{t+1}-G_{\text{BSGD}}^t) & \text{with prob. } 1-p_{\text{out}} \\ \frac{1}{B_1}\sum_{\xi\in\mathcal{B}_1}G_{\text{BSGD}}^{t+1} & \text{with prob. } p_{\text{out}}. \end{cases}$$

**Lemma 7 (Bias and Variance of BSpiderBoost)** *If $\gamma \le \min\{\frac{1}{2L_F},\frac{\sqrt{B_2}}{6L_F}\}$, then the bias and variance of BSpiderBoost are*

$$\frac{1}{T}\sum_{t=0}^{T-1}\mathbb{E}[\mathcal{E}_{bias}^{t+1}] \le \frac{2\sigma_{bias}^2}{m} + \frac{(1-p_{out})^3}{p_{out}B_2}\frac{56L_F^2\gamma^2}{T}\sum_{t=0}^{T-1}\mathbb{E}[\|\mathbb{E}^{t+1}[G^{t+1}|t]\|_2^2] + (\frac{1}{Tp_{out}}+1)\frac{4(1-p_{out})^2}{B_1}(\frac{\sigma_{in}^2}{m}+\sigma_{out}^2)$$

$$\frac{1}{T}\sum_{t=0}^{T-1}\mathbb{E}[\mathcal{E}_{var}^{t+1}] \le \frac{28(1-p_{out})L_F^2\gamma^2}{B_2}\frac{1}{T}\sum_{t=0}^{T-1}\mathbb{E}[\|\mathbb{E}^{t+1}[G^{t+1}|t]\|_2^2] + (\frac{1}{T}+p_{out})\frac{2}{B_1}(\frac{\sigma_{in}^2}{m}+\sigma_{out}^2),$$

*where $\sigma_{in}^2 = \zeta_g^2 C_f^2 + \sigma_g^2 C_g^2 L_f^2$, $\sigma_{out} = C_F^2$, and $\sigma_{bias}^2 = \sigma_g^2 C_g^2 L_f^2$.*

**Proof:** Denote $G^{t+1} = G_{\text{BSB}}^{t+1}$ (8). Like previously (Lemma 5), let $\mathbb{E}[\cdot]$ denote the conditional expectation $\mathbb{E}^{t+1}[\cdot|t]$ which conditions on all past randomness until time $t$. Denote $G_L^{t+1}$ and $G_S^{t+1}$ as the large batch and small batch in BSpiderBoost separately, i.e.,

$$\begin{cases} G_L^{t+1} = \frac{1}{B_1}\sum_{\xi\in\mathcal{B}_1}G_{\text{BSGD}}^{t+1} & \text{with prob. } p_{\text{out}} \\ G_S^{t+1} = G^t + \frac{1}{B_2}\sum_{\xi\in\mathcal{B}_2}(G_{\text{BSGD}}^{t+1}-G_{\text{BSGD}}^t) & \text{with prob. } 1-p_{\text{out}}. \end{cases}$$

The bias of BSpiderBoost can be decomposed to its distance to BSGD and the distance from BSGD to the full gradient, i.e.,

$$
\begin{aligned}
\mathcal{E}_{\text{bias}}^{t+1} &= \left\| \nabla F(\boldsymbol{x}^{t+1}) - \mathbb{E}[G^{t+1}] \right\|_2^2 \\
&\leq 2 \left\| \nabla F(\boldsymbol{x}^{t+1}) - \mathbb{E}[G_{\text{BSGD}}^{t+1}] \right\|_2^2 + 2 \left\| \mathbb{E}[G_{\text{BSGD}}^{t+1}] - \mathbb{E}[G^{t+1}] \right\|_2^2 \\
&\leq \frac{2\sigma_{\text{bias}}^2}{m} + 2 \left\| \mathbb{E}[G_{\text{BSGD}}^{t+1}] - \mathbb{E}[G^{t+1}] \right\|_2^2.
\end{aligned}
\tag{14}
$$

where the last inequality uses the bias of BSGD from Lemma 5. Then the second term can be bounded as follows

$$
\begin{aligned}
\left\| \mathbb{E}[G_{\text{BSGD}}^{t+1}] - \mathbb{E}[G^{t+1}] \right\|_2^2 &= (1 - p_{\text{out}})^2 \left\| \mathbb{E}[G_{\text{BSGD}}^{t+1}] - \mathbb{E}[G_S^{t+1}] \right\|_2^2 \\
&= (1 - p_{\text{out}})^2 \left\| \mathbb{E}[G_{\text{BSGD}}^t] - G^t \right\|_2^2.
\end{aligned}
$$

By taking the expectation of randomness of $G^t$

$$
\begin{aligned}
\left\| \mathbb{E}[G_{\text{BSGD}}^{t+1}] - \mathbb{E}[G^{t+1}] \right\|_2^2 &= (1 - p_{\text{out}})^2 \left( \left\| \mathbb{E}[G_{\text{BSGD}}^t] - \mathbb{E}[G^t] \right\|_2^2 + \mathbb{E} \left\| G^t - \mathbb{E}[G^t] \right\|_2^2 \right) \\
&= (1 - p_{\text{out}})^2 \left( \left\| \mathbb{E}[G_{\text{BSGD}}^t] - \mathbb{E}[G^t] \right\|_2^2 + \mathcal{E}_{\text{var}}^t \right)
\end{aligned}
$$

Note that $\left\| \mathbb{E}[G_{\text{BSGD}}^1] - \mathbb{E}[G^1] \right\|_2^2 = 0$ as the first iteration always chooses the large batch. Then as we always use large batch at $t = 0$ we know that

$$
\frac{1}{T} \sum_{t=0}^{T-1} \left\| \mathbb{E}[G_{\text{BSGD}}^{t+1}] - \mathbb{E}[G^{t+1}] \right\|_2^2 \leq \frac{(1 - p_{\text{out}})^2}{p_{\text{out}}} \frac{1}{T} \sum_{t=0}^{T-1} \mathcal{E}_{\text{var}}^{t+1}.
\tag{15}
$$

Therefore combine (14) and (15) we can upper bound the bias

$$
\frac{1}{T} \sum_{t=0}^{T-1} \mathcal{E}_{\text{bias}}^{t+1} \leq \frac{2\sigma_{\text{bias}}^2}{m} + \frac{2(1 - p_{\text{out}})^2}{p_{\text{out}}} \frac{1}{T} \sum_{t=0}^{T-1} \mathcal{E}_{\text{var}}^{t+1}.
\tag{16}
$$

**Variance.** Now we consider the variance,

$$
\begin{aligned}
\mathcal{E}_{\text{var}}^{t+1} &= \mathbb{E}\left[ \left\| G^{t+1} - \mathbb{E}[G^{t+1}] \right\|_2^2 \right] \\
&\leq (1 - p_{\text{out}}) \mathbb{E}\left[ \left\| G_S^{t+1} - \mathbb{E}[G_S^{t+1}] \right\|_2^2 \right] + p_{\text{out}} \mathbb{E}\left[ \left\| G_L^{t+1} - \mathbb{E}[G_L^{t+1}] \right\|_2^2 \right] \\
&= \frac{(1 - p_{\text{out}})}{B_2} \mathbb{E}\left[ \left\| G_{\text{BSGD}}^{t+1} - G_{\text{BSGD}}^t - \mathbb{E}[G_{\text{BSGD}}^{t+1} - G_{\text{BSGD}}^t] \right\|_2^2 \right] + \frac{p_{\text{out}}}{B_1} \mathbb{E}\left[ \left\| G_{\text{BSGD}}^{t+1} - \mathbb{E}[G_{\text{BSGD}}^{t+1}] \right\|_2^2 \right] \\
&\leq \frac{1 - p_{\text{out}}}{B_2} \mathbb{E}\left[ \left\| G_{\text{BSGD}}^{t+1} - G_{\text{BSGD}}^t - \mathbb{E}[G_{\text{BSGD}}^{t+1} - G_{\text{BSGD}}^t] \right\|_2^2 \right] + \frac{p_{\text{out}}}{B_1} \left( \frac{\sigma_{\text{in}}^2}{m} + \sigma_{\text{out}}^2 \right)
\end{aligned}
\tag{17}
$$

where the last equality is because the large batch in BSpiderBoost is similar to BSGD.

$$
\mathcal{E}_{\text{var}}^1 = \mathbb{E}\left[ \left\| G^1 - \mathbb{E}[G^1] \right\|_2^2 \right] = \mathbb{E}\left[ \left\| G_L^1 - \mathbb{E}[G_L^1] \right\|_2^2 \right] = \frac{1}{B_1} \mathbb{E}\left[ \left\| G_{\text{BSGD}}^1 - \mathbb{E}[G_{\text{BSGD}}^1] \right\|_2^2 \right] \leq \frac{1}{B_1} \left( \frac{\sigma_{\text{in}}^2}{m} + \sigma_{\text{out}}^2 \right).
\tag{18}
$$

Finally, we expand the variance at small batch size epoch

$$
\begin{aligned}
&\mathbb{E}\left[ \left\| G_{\text{BSGD}}^{t+1} - G_{\text{BSGD}}^t - \mathbb{E}[G_{\text{BSGD}}^{t+1} - G_{\text{BSGD}}^t] \right\|_2^2 \right] \\
&= \underbrace{\mathbb{E}\left[ \left\| G_{\text{BSGD}}^{t+1} - G_{\text{BSGD}}^t - \mathbb{E}_{\eta|\xi, \tilde{\eta}|\xi}[G_{\text{BSGD}}^{t+1} - G_{\text{BSGD}}^t] \right\|_2^2 \right]}_{\text{Inner variance } \mathcal{T}_{\text{in}}} \\
&\quad + \underbrace{\mathbb{E}_{\xi}\left[ \left\| \mathbb{E}_{\eta|\xi, \tilde{\eta}|\xi}[G_{\text{BSGD}}^{t+1} - G_{\text{BSGD}}^t] - \mathbb{E}[G_{\text{BSGD}}^{t+1} - G_{\text{BSGD}}^t] \right\|_2^2 \right]}_{\text{Outer variance } \mathcal{T}_{\text{out}}}.
\end{aligned}
$$

The outer variance $\mathcal{T}_{\text{out}}$ can be upper bounded as

$$\mathcal{T}_{\text{out}} \le \mathbb{E}_\xi \left[ \left\| \mathbb{E}_{\eta|\xi, \tilde{\eta}|\xi}[G_{\text{BSGD}}^{t+1} - G_{\text{BSGD}}^t] \right\|_2^2 \right]$$

$$= \mathbb{E}_\xi \left[ \left\| (\mathbb{E}_{\tilde{\eta}|\xi}[\nabla g_{\tilde{\eta}}(\boldsymbol{x}^t)])^\top \mathbb{E}_{\eta|\xi}[\nabla f_\xi(\tfrac{1}{m} \sum_{\eta \in H_\xi} g_\eta(\boldsymbol{x}^t))] - (\mathbb{E}_{\tilde{\eta}|\xi}[\nabla g_{\tilde{\eta}}(\boldsymbol{x}^{t-1})])^\top \mathbb{E}_{\eta|\xi}[\nabla f_\xi(\tfrac{1}{m} \sum_{\eta \in H_\xi} g_\eta(\boldsymbol{x}^{t-1}))] \right\|_2^2 \right]$$

$$\le 2\,\mathbb{E}_\xi \left[ \left\| (\mathbb{E}_{\tilde{\eta}|\xi}[\nabla g_{\tilde{\eta}}(\boldsymbol{x}^t)] - \mathbb{E}_{\tilde{\eta}|\xi}[\nabla g_{\tilde{\eta}}(\boldsymbol{x}^{t-1})])^\top \mathbb{E}_{\eta|\xi}[\nabla f_\xi(\tfrac{1}{m} \sum_{\eta \in H_\xi} g_\eta(\boldsymbol{x}^t))] \right\|_2^2 \right]$$

$$+ 2\,\mathbb{E}_\xi \left[ \left\| (\mathbb{E}_{\tilde{\eta}|\xi}[\nabla g_{\tilde{\eta}}(\boldsymbol{x}^{t-1})])^\top \mathbb{E}_{\eta|\xi}[\nabla f_\xi(\tfrac{1}{m} \sum_{\eta \in H_\xi} g_\eta(\boldsymbol{x}^t)) - \nabla f_\xi(\tfrac{1}{m} \sum_{\eta \in H_\xi} g_\eta(\boldsymbol{x}^{t-1}))] \right\|_2^2 \right]$$

$$\le 2L_g^2 C_f^2 \left\| \boldsymbol{x}^t - \boldsymbol{x}^{t-1} \right\|_2^2 + 2C_g^4 L_f^2 \left\| \boldsymbol{x}^t - \boldsymbol{x}^{t-1} \right\|_2^2$$

$$= 2L_F^2 \left\| \boldsymbol{x}^t - \boldsymbol{x}^{t-1} \right\|_2^2$$

$$= 2L_F^2 \gamma^2 \left\| G^t \right\|_2^2.$$

The inner variance can be bounded by

$$\mathcal{T}_{\text{in}} \le 4\,\mathbb{E} \left[ \left\| (\tfrac{1}{m} \sum_{\tilde{\eta} \in \tilde{H}_\xi} (\nabla g_{\tilde{\eta}}(\boldsymbol{x}^t) - \nabla g_{\tilde{\eta}}(\boldsymbol{x}^{t-1})) - \mathbb{E}_{\tilde{\eta}|\xi}[\nabla g_{\tilde{\eta}}(\boldsymbol{x}^t) - \nabla g_{\tilde{\eta}}(\boldsymbol{x}^{t-1})])^\top \nabla f_\xi(\tfrac{1}{m} \sum_{\eta \in H_\xi} g_\eta(\boldsymbol{x}^t)) \right\|_2^2 \right]$$

$$+ 4\,\mathbb{E} \left[ \left\| (\mathbb{E}_{\tilde{\eta}|\xi}[\nabla g_{\tilde{\eta}}(\boldsymbol{x}^t) - \nabla g_{\tilde{\eta}}(\boldsymbol{x}^{t-1})])^\top (\nabla f_\xi(\tfrac{1}{m} \sum_{\eta \in H_\xi} g_\eta(\boldsymbol{x}^t)) - \mathbb{E}_{\eta|\xi}[\nabla f_\xi(\tfrac{1}{m} \sum_{\eta \in H_\xi} g_\eta(\boldsymbol{x}^t))]) \right\|_2^2 \right]$$

$$+ 4\,\mathbb{E} \left[ \left\| (\tfrac{1}{m} \sum_{\tilde{\eta} \in \tilde{H}_\xi} \nabla g_{\tilde{\eta}}(\boldsymbol{x}^{t-1}))^\top \left( \nabla f_\xi(\tfrac{1}{m} \sum_{\eta \in H_\xi} g_\eta(\boldsymbol{x}^t)) - \nabla f_\xi(\tfrac{1}{m} \sum_{\eta \in H_\xi} g_\eta(\boldsymbol{x}^{t-1})) \right. \right. \right.$$

$$\left. \left. \left. - \mathbb{E}_{\eta|\xi} \left[ \nabla f_\xi(\tfrac{1}{m} \sum_{\eta \in H_\xi} g_\eta(\boldsymbol{x}^t)) - \nabla f_\xi(\tfrac{1}{m} \sum_{\eta \in H_\xi} g_\eta(\boldsymbol{x}^{t-1})) \right] \right) \right\|_2^2 \right]$$

$$+ 4\,\mathbb{E} \left[ \left\| (\tfrac{1}{m} \sum_{\tilde{\eta} \in \tilde{H}_\xi} \nabla g_{\tilde{\eta}}(\boldsymbol{x}^{t-1}) - \mathbb{E}_{\tilde{\eta}|\xi}[\nabla g_{\tilde{\eta}}(\boldsymbol{x}^{t-1})])^\top (\nabla f_\xi(\tfrac{1}{m} \sum_{\eta \in H_\xi} g_\eta(\boldsymbol{x}^t)) - \nabla f_\xi(\tfrac{1}{m} \sum_{\eta \in H_\xi} g_\eta(\boldsymbol{x}^{t-1}))) \right\|_2^2 \right]$$

$$\le \frac{4C_f^2}{m} \mathbb{E} \left[ \left\| \nabla g_{\tilde{\eta}}(\boldsymbol{x}^t) - \nabla g_{\tilde{\eta}}(\boldsymbol{x}^{t-1}) - \mathbb{E}_{\tilde{\eta}|\xi}[\nabla g_{\tilde{\eta}}(\boldsymbol{x}^t) - \nabla g_{\tilde{\eta}}(\boldsymbol{x}^{t-1})] \right\|_2^2 \right] + \frac{4L_g^2 C_f^2}{m} \left\| \boldsymbol{x}^t - \boldsymbol{x}^{t-1} \right\|_2^2$$

$$+ 4C_g^2 \mathbb{E} \left[ \left\| \nabla f_\xi(\tfrac{1}{m} \sum_{\eta \in H_\xi} g_\eta(\boldsymbol{x}^t)) - \nabla f_\xi(\tfrac{1}{m} \sum_{\eta \in H_\xi} g_\eta(\boldsymbol{x}^{t-1})) \right. \right.$$

$$\left. \left. - \mathbb{E}_{\eta|\xi}[\nabla f_\xi(\tfrac{1}{m} \sum_{\eta \in H_\xi} g_\eta(\boldsymbol{x}^t)) - \nabla f_\xi(\tfrac{1}{m} \sum_{\eta \in H_\xi} g_\eta(\boldsymbol{x}^{t-1}))] \right\|_2^2 \right]$$

$$+ \frac{4C_g^4 L_f^2}{m} \left\| \boldsymbol{x}^t - \boldsymbol{x}^{t-1} \right\|_2^2.$$

Then we have that

$$\mathcal{T}_{\text{in}} \le \frac{4L_g^2 C_f^2}{m} \left\| \boldsymbol{x}^t - \boldsymbol{x}^{t-1} \right\|_2^2 + \frac{4L_g^2 C_f^2}{m} \left\| \boldsymbol{x}^t - \boldsymbol{x}^{t-1} \right\|_2^2$$

$$+ 4C_g^2 \mathbb{E} \left[ \left\| \nabla f_\xi(\tfrac{1}{m} \sum_{\eta \in H_\xi} g_\eta(\boldsymbol{x}^t)) - \nabla f_\xi(\tfrac{1}{m} \sum_{\eta \in H_\xi} g_\eta(\boldsymbol{x}^{t-1})) - (\nabla f_\xi(\mathbb{E}_{\eta|\xi}[g_\eta(\boldsymbol{x}^t)]) - \nabla f_\xi(\mathbb{E}_{\eta|\xi}[g_\eta(\boldsymbol{x}^{t-1})])) \right\|_2^2 \right]$$

$$+ 4C_g^2 \mathbb{E} \left[ \left\| (\nabla f_\xi(\mathbb{E}_{\eta|\xi}[g_\eta(\boldsymbol{x}^t)]) - \nabla f_\xi(\mathbb{E}_{\eta|\xi}[g_\eta(\boldsymbol{x}^{t-1})])) \right. \right.$$

$$\left. \left. - \mathbb{E}_{\eta|\xi}[\nabla f_\xi(\tfrac{1}{m} \sum_{\eta \in H_\xi} g_\eta(\boldsymbol{x}^t)) - \nabla f_\xi(\tfrac{1}{m} \sum_{\eta \in H_\xi} g_\eta(\boldsymbol{x}^{t-1}))] \right\|_2^2 \right]$$

$$+ \frac{4C_g^4 L_f^2}{m} \left\| \boldsymbol{x}^t - \boldsymbol{x}^{t-1} \right\|_2^2$$

$$\le \frac{8L_g^2 C_f^2}{m} \left\| \boldsymbol{x}^t - \boldsymbol{x}^{t-1} \right\|_2^2 + \frac{8C_g^4 L_f^2}{m} \left\| \boldsymbol{x}^t - \boldsymbol{x}^{t-1} \right\|_2^2 + \frac{4C_g^4 L_f^2}{m} \left\| \boldsymbol{x}^t - \boldsymbol{x}^{t-1} \right\|_2^2$$

$$\le \frac{12L_F^2}{m} \left\| \boldsymbol{x}^t - \boldsymbol{x}^{t-1} \right\|_2^2$$

$$= \frac{12L_F^2 \gamma^2}{m} \left\| G^t \right\|_2^2.$$

To sum up, the variance is bounded by

$$\mathcal{E}_{\text{var}}^{t+1} \leq \frac{2(1-p_{\text{out}})L_F^2\gamma^2}{B_2}(1+\frac{6}{m})\left\|G^t\right\|_2^2 + \frac{p_{\text{out}}}{B_1}(\frac{\sigma_{\text{in}}^2}{m}+\sigma_{\text{out}}^2)$$

$$\leq \frac{14(1-p_{\text{out}})L_F^2\gamma^2}{B_2}\left\|G^t\right\|_2^2 + \frac{p_{\text{out}}}{B_1}(\frac{\sigma_{\text{in}}^2}{m}+\sigma_{\text{out}}^2)$$

$$= \frac{14(1-p_{\text{out}})L_F^2\gamma^2}{B_2}\mathbb{E}^t[\left\|G^t\right\|_2^2|t-1] + \frac{p_{\text{out}}}{B_1}(\frac{\sigma_{\text{in}}^2}{m}+\sigma_{\text{out}}^2).$$

Then averaging over time and now we redefine $\mathbb{E}[\cdot] = \mathbb{E}^T\ldots[\mathbb{E}^0[\cdot]]$

$$\frac{1}{T}\sum_{t=0}^{T-1}\mathbb{E}[\mathcal{E}_{\text{var}}^{t+1}] \leq \frac{14(1-p_{\text{out}})L_F^2\gamma^2}{B_2}\frac{1}{T}\sum_{t=0}^{T-1}\mathbb{E}[\left\|G^{t+1}\right\|_2^2] + \frac{\mathcal{E}_{\text{var}}^1}{T} + \frac{p_{\text{out}}}{B_1}(\frac{\sigma_{\text{in}}^2}{m}+\sigma_{\text{out}}^2)$$

$$= \frac{14(1-p_{\text{out}})L_F^2\gamma^2}{B_2}\frac{1}{T}\sum_{t=0}^{T-1}(\mathbb{E}[\mathcal{E}_{\text{var}}^{t+1}] + \mathbb{E}[\left\|\mathbb{E}^t[G^{t+1}|t]\right\|_2^2]) + \frac{\mathcal{E}_{\text{var}}^1}{T} + \frac{p_{\text{out}}}{B_1}(\frac{\sigma_{\text{in}}^2}{m}+\sigma_{\text{out}}^2).$$

If we take $\gamma \leq \frac{\sqrt{B_2}}{6L_F}$, then $\frac{14(1-p_{\text{out}})L_F^2\gamma^2}{B_2} \leq \frac{1}{2}$, therefore

$$\frac{1}{T}\sum_{t=0}^{T-1}\mathbb{E}[\mathcal{E}_{\text{var}}^{t+1}] \leq \frac{28(1-p_{\text{out}})L_F^2\gamma^2}{B_2}\frac{1}{T}\sum_{t=0}^{T-1}\mathbb{E}[\left\|\mathbb{E}^t[G^{t+1}|t]\right\|_2^2] + \frac{\mathcal{E}_{\text{var}}^1}{T} + \frac{2p_{\text{out}}}{B_1}(\frac{\sigma_{\text{in}}^2}{m}+\sigma_{\text{out}}^2)$$

$$\overset{(18)}{\leq} \frac{28(1-p_{\text{out}})L_F^2\gamma^2}{B_2}\frac{1}{T}\sum_{t=0}^{T-1}\mathbb{E}[\left\|\mathbb{E}^t[G^{t+1}|t]\right\|_2^2] + (\frac{1}{T}+p_{\text{out}})\frac{2}{B_1}(\frac{\sigma_{\text{in}}^2}{m}+\sigma_{\text{out}}^2)$$

Then with (16), we can bound the bias by

$$\frac{1}{T}\sum_{t=0}^{T-1}\mathbb{E}[\mathcal{E}_{\text{bias}}^{t+1}] \leq \frac{2\sigma_{\text{bias}}^2}{m} + \frac{(1-p_{\text{out}})^3}{p_{\text{out}}B_2}\frac{56L_F^2\gamma^2}{T}\sum_{t=0}^{T-1}\mathbb{E}[\left\|\mathbb{E}^t[G^{t+1}|t]\right\|_2^2] + (\frac{1}{Tp_{\text{out}}}+1)\frac{4(1-p_{\text{out}})^2}{B_1}(\frac{\sigma_{\text{in}}^2}{m}+\sigma_{\text{out}}^2)$$

$\square$

**Theorem 7 (BSpiderBoost Convergence)** *Consider the* (CSO) *problem. Suppose Assumptions 3, 4, 5 holds true. Let step size $\gamma \leq 1/(13L_F)$. Then for BSpiderBoost, $\boldsymbol{x}^s$ picked uniformly at random among $\{\boldsymbol{x}^t\}_{t=0}^{T-1}$ satisfies: $\mathbb{E}[\|\nabla F(\boldsymbol{x}^s)\|_2^2] \leq \varepsilon^2$, for nonconvex $F$, if the inner batch size $m = \Omega(\sigma_{bias}^2\varepsilon^{-2})$, the hyperparameters of the outer loop of BSpiderBoost are $B_1 = (\sigma_{in}^2/m + \sigma_{out}^2)\varepsilon^{-2}$, $B_2 = \mathcal{O}(\varepsilon^{-1})$, $p_{out} = 1/B_2$, and the number of iterations $T = \Omega\left(L_F(F(\boldsymbol{x}^0) - F^\star)\varepsilon^{-2}\right)$, where $\sigma_{in}^2 = \zeta_g^2 C_f^2 + \sigma_g^2 C_g^2 L_f^2$, $\sigma_{out}^2 = C_F^2$, and $\sigma_{bias}^2 = \sigma_g^2 C_g^2 L_f^2$.*

**Proof:** Denote $G^{t+1} = G_{\text{BSB}}^{t+1}$ (8). Using descent lemma (Lemma 4) and bias-variance bounds of BSpiderBoost (Lemma 7)

$$\frac{1}{T}\sum_{t=0}^{T-1}\mathbb{E}[\|\nabla F(\boldsymbol{x}^t)\|_2^2] + \frac{1}{2T}\sum_{t=0}^{T-1}\mathbb{E}[\left\|\mathbb{E}^t[G^{t+1}|t]\right\|_2^2]$$

$$\leq \frac{2(F(\boldsymbol{x}^0)-F^\star)}{\gamma T} + L_F\gamma\frac{1}{T}\sum_{t=0}^{T-1}\mathbb{E}[\mathcal{E}_{\text{var}}^{t+1}] + \frac{1}{T}\sum_{t=0}^{T-1}\mathbb{E}[\mathcal{E}_{\text{bias}}^{t+1}]$$

$$\leq \frac{2(F(\boldsymbol{x}^0)-F^\star)}{\gamma T} + L_F\gamma\frac{1}{T}\sum_{t=0}^{T-1}\mathbb{E}[\mathcal{E}_{\text{var}}^{t+1}] + \frac{2\sigma_{\text{bias}}^2}{m} + \frac{2}{p_{\text{out}}}\frac{1}{T}\sum_{t=0}^{T-1}\mathbb{E}[\mathcal{E}_{\text{var}}^{t+1}]$$

$$\leq \frac{2(F(\boldsymbol{x}^0)-F^\star)}{\gamma T} + \frac{2\sigma_{\text{bias}}^2}{m} + \frac{3}{p_{\text{out}}}\frac{1}{T}\sum_{t=0}^{T-1}\mathbb{E}[\mathcal{E}_{\text{var}}^{t+1}]$$

where the last inequality use $\gamma \leq \frac{1}{2L_F}$. Use the variance estimation of $G^{t+1}$ and choose $B_2p_{\text{out}} = 1$

$$\frac{1}{T}\sum_{t=0}^{T-1}\mathbb{E}[\|\nabla F(\boldsymbol{x}^t)\|_2^2] + \frac{1}{2T}\sum_{t=0}^{T-1}\mathbb{E}[\left\|\mathbb{E}^t[G^{t+1}|t]\right\|_2^2]$$

$$\leq \frac{2(F(\boldsymbol{x}^0)-F^\star)}{\gamma T} + \frac{2\sigma_{\text{bias}}^2}{m} + 84L_F^2\gamma^2\frac{1}{T}\sum_{t=0}^{T-1}\mathbb{E}[\left\|\mathbb{E}^t[G^{t+1}|t]\right\|_2^2] + (\frac{1}{Tp_{\text{out}}}+1)\frac{6}{B_1}(\frac{\sigma_{\text{in}}^2}{m}+\sigma_{\text{out}}^2).$$

Now we can let $\gamma \leq \frac{1}{13L_F}$ such that $84L_F^2\gamma^2 \leq \frac{1}{2}$

$$\frac{1}{T}\sum_{t=0}^{T-1}\mathbb{E}[\|\nabla F(\boldsymbol{x}^t)\|_2^2] \leq \frac{2(F(\boldsymbol{x}^0)-F^\star)}{\gamma T} + \frac{2\sigma_{\text{bias}}^2}{m} + (\frac{1}{Tp_{\text{out}}}+1)\frac{6}{B_1}(\frac{\sigma_{\text{in}}^2}{m}+\sigma_{\text{out}}^2).$$

In order for the right-hand side to be $\varepsilon^2$, the inner batch size

$$m \geq \frac{2\sigma_{\text{bias}}^2}{\varepsilon^2},$$

and the outer batch size

$$B_1 = \frac{\sigma_{\text{in}}^2/m+\sigma_{\text{out}}^2}{\varepsilon^2}, \quad B_2 = \sqrt{B_1}, \quad p_{\text{out}} = \frac{1}{B_2}.$$

The step size $\gamma$ is upper bounded by $\min\{\frac{1}{2L_F}, \frac{\sqrt{B_2}}{6L_F}, \frac{1}{13L_F}\}$. As $B_2 \geq 1$, we can take $\gamma = \frac{1}{13L_F}$. So we need iteration $T$ greater than

$$T \geq \frac{26L_F(F(\boldsymbol{x}^0)-F^\star)}{\varepsilon^2}.$$

By picking $\boldsymbol{x}^s$ uniformly at random among $\{\boldsymbol{x}^t\}_{t=0}^{T-1}$, we get the desired guarantee. $\square$

The resulting sample complexity of BSpiderBoost to get to an $\varepsilon$-stationary point is $\mathcal{O}(\varepsilon^{-5})$.

## D.5 Convergence of E-BSpiderBoost

In this section, we analyze the sample complexity of Algorithm 3 (E-BSpiderBoost) for the CSO problem.

**Lemma 8 (Bias and Variance of E-BSpiderBoost)** *The bias and variance of E-BSpiderBoost are*

$$\frac{1}{T}\sum_{t=0}^{T-1}\mathbb{E}[\mathcal{E}_{var}^{t+1}] \leq \frac{28(1-p_{out})L_F^2\gamma^2}{B_2}\frac{1}{T}\sum_{t=0}^{T-1}\mathbb{E}[\|\mathbb{E}^t[G^{t+1}|t]\|_2^2] + (\frac{1}{Tp_{out}}+1)\frac{28p_{out}}{B_1}(\frac{\sigma_{in}^2}{m}+\sigma_{out}^2)$$

$$\frac{1}{T}\sum_{t=0}^{T-1}\mathbb{E}[\mathcal{E}_{bias}^{t+1}] \leq \frac{2\tilde{\sigma}_{bias}^2}{m^4} + \frac{2}{p_{out}}\frac{(1-p_{out})^2}{T}\sum_{t=0}^{T-1}\mathbb{E}[\mathcal{E}_{var}^{t+1}],$$

*where $\sigma_{in}^2 := \zeta_g^2 C_f^2 + \sigma_g^2 C_g^2 L_f^2$, $\sigma_{out} = C_F^2$, and $\tilde{\sigma}_{bias}^2 = C_g^2 C_e^2$ with $C_e^2$ defined in Corollary 1.*

**Proof:** Denote $G^{t+1} = G_{E\text{-BSB}}^{t+1}$ (9). Like previously (Lemma 5), let $\mathbb{E}[\cdot]$ denote the conditional expectation $\mathbb{E}^t[\cdot|t]$ which conditions on all past randomness until time $t$. Let $G^{t+1} = G_{E\text{-BSB}}^{t+1}$ be the E-BSpiderBoost update. We expand the bias as follows

$$\mathcal{E}_{bias}^{t+1} = \|\nabla F(\boldsymbol{x}^{t+1}) - \mathbb{E}[G^{t+1}]\|_2^2$$
$$\leq 2\|\nabla F(\boldsymbol{x}^{t+1}) - \mathbb{E}[G_{E\text{-BSGD}}^{t+1}\|_2^2] + 2\|\mathbb{E}[G_{E\text{-BSGD}}^{t+1}] - \mathbb{E}[G^{t+1}]\|_2^2.$$

From Lemma 6, we know that

$$\|\nabla F(\boldsymbol{x}^{t+1}) - \mathbb{E}[G_{E\text{-BSGD}}^{t+1}]\|_2^2 \leq \frac{\tilde{\sigma}_{bias}^2}{m^4}.$$

The distance between $\mathbb{E}[G_{E\text{-BSGD}}^{t+1}]$ and $\mathbb{E}[G^{t+1}]$ can be bounded as follows.

$$\|\mathbb{E}[G_{E\text{-BSGD}}^{t+1}] - \mathbb{E}[G^{t+1}]\|_2^2 = (1-p_{out})^2\|\mathbb{E}[G_{E\text{-BSGD}}^{t+1}] - (G^t + \mathbb{E}[G_{E\text{-BSGD}}^{t+1} - G_{E\text{-BSGD}}^t])\|_2^2$$
$$= (1-p_{out})^2\|\mathbb{E}[G_{E\text{-BSGD}}^t] - G^t\|_2^2$$

Taking expectation with respect to $G^t$

$$\|\mathbb{E}[G_{E\text{-BSGD}}^{t+1}] - \mathbb{E}[G^{t+1}]\|_2^2 \leq (1-p_{out})^2(\|\mathbb{E}[G_{E\text{-BSGD}}^t] - \mathbb{E}[G^t]\|_2^2 + \|G^t - \mathbb{E}[G^t]\|_2^2).$$

where $\|\mathbb{E}[G_{E\text{-BSGD}}^1] - \mathbb{E}[G^1]\|_2^2 = 0$. By averaging over time we have

$$\frac{1}{T}\sum_{t=0}^{T-1}\|\mathbb{E}[G_{E\text{-BSGD}}^{t+1}] - \mathbb{E}[G^{t+1}]\|_2^2 \leq \frac{1}{p_{out}}\frac{(1-p_{out})^2}{T}\sum_{t=0}^{T-1}\mathbb{E}[\mathcal{E}_{var}^{t+1}].$$

Then the bias is bounded by

$$\frac{1}{T}\sum_{t=0}^{T-1}\mathbb{E}[\mathcal{E}_{bias}^{t+1}] \leq \frac{2\tilde{\sigma}_{bias}^2}{m^4} + \frac{2}{p_{out}}\frac{(1-p_{out})^2}{T}\sum_{t=0}^{T-1}\mathbb{E}[\mathcal{E}_{var}^{t+1}].$$

**Variance.** Since the extrapolation only gives a constant overhead given Lemma 2

$$\frac{1}{T}\sum_{t=0}^{T-1}\mathbb{E}\left[\|G_{BSB}^{t+1} - \mathbb{E}^t[G_{BSB}^{t+1}|t]\|_2^2\right] \leq \frac{28(1-p_{out})L_F^2\gamma^2}{B_2}\frac{1}{T}\sum_{t=0}^{T-1}\mathbb{E}[\|\mathbb{E}^t[G_{BSB}^{t+1}|t]\|_2^2] + (\frac{1}{T}+p_{out})\frac{28}{B_1}(\frac{\sigma_{in}^2}{m}+\sigma_{out}^2).$$

Then the variance is bounded by

$$\frac{1}{T}\sum_{t=0}^{T-1}\mathbb{E}[\mathcal{E}_{var}^{t+1}] \leq \frac{28(1-p_{out})L_F^2\gamma^2}{B_2}\frac{1}{T}\sum_{t=0}^{T-1}\mathbb{E}[\|\mathbb{E}^t[G^{t+1}|t]\|_2^2] + (\frac{1}{Tp_{out}}+1)\frac{28p_{out}}{B_1}(\frac{\sigma_{in}^2}{m}+\sigma_{out}^2).$$

$\square$

**Theorem 4** *[E-BSpiderBoost Convergence] Consider the* (CSO) *problem under the same assumptions as Theorem 3. Let step size $\gamma \leq 1/(13L_F)$. Then the output $\boldsymbol{x}^s$ of E-BSpiderBoost (Algorithm 3) satisfies: $\mathbb{E}[\|\nabla F(\boldsymbol{x}^s)\|_2^2] \leq \varepsilon^2$, for nonconvex $F$, if the inner batch size $m = \mathcal{O}(C_e C_g \varepsilon^{-0.5})$, the hyperparameters of the outer loop of E-BSpiderBoost $B_1 = (\tilde{L}_F^2/m + C_F^2)\varepsilon^{-2}$, $B_2 = \sqrt{B_1}$, $p_{out} = 1/B_2$, and the number of iterations*

$$T = \Omega(L_F(F(\boldsymbol{x}^0) - F^\star)\varepsilon^{-2}).$$

**Proof:** Denote $G^{t+1} = G_{\text{E-BSB}}^{t+1}$ (9). Using descent lemma (Lemma 4) and bias-variance bounds of E-BSpiderBoost (Lemma 8)

$$\frac{1}{T}\sum_{t=0}^{T-1}\mathbb{E}[\|\nabla F(\boldsymbol{x}^t)\|_2^2] + \frac{1}{2T}\sum_{t=0}^{T-1}\mathbb{E}[\|\mathbb{E}^t[G^{t+1}|t]\|_2^2]$$

$$\leq \frac{2(F(\boldsymbol{x}^0)-F^\star)}{\gamma T} + L_F\gamma\frac{1}{T}\sum_{t=0}^{T-1}\mathbb{E}[\mathcal{E}_{\text{var}}^{t+1}] + \frac{1}{T}\sum_{t=0}^{T-1}\mathbb{E}[\mathcal{E}_{\text{bias}}^{t+1}]$$

$$\leq \frac{2(F(\boldsymbol{x}^0)-F^\star)}{\gamma T} + L_F\gamma\frac{1}{T}\sum_{t=0}^{T-1}\mathbb{E}[\mathcal{E}_{\text{var}}^{t+1}] + \frac{2\tilde{\sigma}_{\text{bias}}^2}{m^4} + \frac{2}{p_{\text{out}}}\frac{1}{T}\sum_{t=0}^{T-1}\mathbb{E}[\mathcal{E}_{\text{var}}^{t+1}]$$

$$\leq \frac{2(F(\boldsymbol{x}^0)-F^\star)}{\gamma T} + \frac{2\tilde{\sigma}_{\text{bias}}^2}{m^4} + \frac{3}{p_{\text{out}}}\frac{1}{T}\sum_{t=0}^{T-1}\mathbb{E}[\mathcal{E}_{\text{var}}^{t+1}]$$

where the last inequality use $\gamma \leq \frac{1}{2L_F}$. Use the variance estimation of $G^{t+1}$ and choose $B_2 p_{\text{out}} = 1$

$$\frac{1}{T}\sum_{t=0}^{T-1}\mathbb{E}[\|\nabla F(\boldsymbol{x}^t)\|_2^2] + \frac{1}{2T}\sum_{t=0}^{T-1}\mathbb{E}[\|\mathbb{E}^t[G^{t+1}|t]\|_2^2]$$

$$\leq \frac{2(F(\boldsymbol{x}^0)-F^\star)}{\gamma T} + \frac{2\tilde{\sigma}_{\text{bias}}^2}{m^4} + 84L_F^2\gamma^2\frac{1}{T}\sum_{t=0}^{T-1}\mathbb{E}[\|\mathbb{E}^t[G^{t+1}|t]\|_2^2] + \left(\frac{1}{Tp_{\text{out}}}+1\right)\frac{84}{B_1}\left(\frac{\sigma_{\text{in}}^2}{m}+\sigma_{\text{out}}^2\right).$$

Now we can let $\gamma \leq \frac{1}{13L_F}$ such that $84L_F^2\gamma^2 \leq \frac{1}{2}$

$$\frac{1}{T}\sum_{t=0}^{T-1}\mathbb{E}[\|\nabla F(\boldsymbol{x}^t)\|_2^2] \leq \frac{2(F(\boldsymbol{x}^0)-F^\star)}{\gamma T} + \frac{2\tilde{\sigma}_{\text{bias}}^2}{m^4} + \left(\frac{1}{Tp_{\text{out}}}+1\right)\frac{84}{B_1}\left(\frac{\sigma_{\text{in}}^2}{m}+\sigma_{\text{out}}^2\right). \tag{19}$$

In order to make the right-hand side $\varepsilon^2$, the inner batch size

$$m = \Omega(\tilde{\sigma}_{\text{bias}}^2 \varepsilon^{-0.5}),$$

and the outer batch size

$$B_1 = \frac{(\sigma_{\text{in}}^2/m + \sigma_{\text{out}}^2)}{\varepsilon^2}, \quad B_2 = \sqrt{B_1}, \quad p_{\text{out}} = \frac{1}{B_2}.$$

The step size $\gamma$ is upper bounded by $\min\{\frac{1}{2L_F}, \frac{\sqrt{B_2}}{6L_F}, \frac{1}{13L_F}\}$. As $B_2 \geq 1$, we can take $\gamma = \frac{1}{13L_F}$. So we need iteration $T$ greater than

$$T \geq \frac{26L_F(F(\boldsymbol{x}^0)-F^\star)}{\varepsilon^2}.$$

By picking $\boldsymbol{x}^s$ uniformly at random among $\{\boldsymbol{x}^t\}_{t=0}^{T-1}$, we get the desired guarantee. $\qquad\square$

# E  Stationary Point Convergence Proofs from Section 4 (FCCO)

In this section, we provide the convergence proofs for the FCCO problem. We start by analyzing a variant of BSpiderBoost (Algorithm 3) for this case in Appendix E.1. In Appendix E.2, we present a multi-level variance reduction approach (called NestedVR) that applies variance reduction in both outer (over the random variable $i$) and inner (over the random variable $\eta|i$) loops. In Appendix E.3, we analyze E-NestedVR. As in the case of CSO analyses, our proofs go via bounds on bias and variance terms of these algorithms.

## E.1  E-BSpiderBoost for FCCO problem

**Theorem 8** *Consider the* (FCCO) *problem. Suppose Assumptions 3, 4, 5, 6 holds true. Let step size $\gamma = \mathcal{O}(1/L_F)$. Then the output of E-BSpiderBoost (Algorithm 3) satisfies: $\mathbb{E}[\|\nabla F(\boldsymbol{x}^s)\|_2^2] \leq \varepsilon^2$, for nonconvex F, if the inner batch size $m = \Omega(\max\{C_e C_g \varepsilon^{-1/2}, \sigma_{in}^2 n^{-1}\varepsilon^{-2}\})$, the hyperparameters of the outer loop of E-BSpiderBoost $B_1 = n, B_2 = \sqrt{n}, p_{out} = 1/B_2$, and the number of iterations $T = \Omega\left(L_F(F(\boldsymbol{x}^0) - F^\star)\varepsilon^{-2}\right)$. The resulting sample complexity is*

$$\mathcal{O}\left(L_F(F(\boldsymbol{x}^0) - F^\star)\max\left\{\frac{\sqrt{n}C_e C_g}{\varepsilon^{2.5}}, \frac{\sigma_{in}^2}{\sqrt{n}\varepsilon^4}\right\}\right).$$

**Remark 9** *The sample complexity depends on the relation between $n$ and $\varepsilon$*

- *When $n = \mathcal{O}(1)$, we have a complexity of $\mathcal{O}(\varepsilon^{-4})$. This happens because we did not apply variance reduction for the inner loop.*

- *When $n = \Theta(\varepsilon^{-2/3})$, E-BSpiderBoost has same performance as MSVR-V2 [19] of $\mathcal{O}(n\varepsilon^{-3}) = \mathcal{O}(\varepsilon^{-11/3})$.*

- *When $n = \Theta(\varepsilon^{-1.5})$, E-BSpiderBoost achieves a better sample complexity of $\mathcal{O}(\varepsilon^{-3.25})$ than $\mathcal{O}(\varepsilon^{-4.5})$ from MSVR-V2 [19].*

- *When $n = \Theta(\varepsilon^{-2})$, we recover $\mathcal{O}(\varepsilon^{-3.5})$ sample complexity as in Theorem 4.*

**Proof:** Denote $G^{t+1} = G_{\text{E-BSB}}^{t+1}$ (9). As we are using the finite-sum variant of SpiderBoost for the outer loop of the CSO problem, we only need to change the (17) and (18) to reflect that the outer variance is 0 now instead of $\frac{\sigma_{\text{out}}^2}{B_1}$ in the general CSO case. More concretely, we update (17) to

$$
\begin{aligned}
\mathcal{E}_{\text{var}}^{t+1} &= \mathbb{E}[\|G^{t+1} - \mathbb{E}[G^{t+1}]\|_2^2] \\
&\leq (1 - p_{\text{out}}) \, \mathbb{E}[\|G_S^{t+1} - \mathbb{E}[G_S^{t+1}]\|_2^2] + p_{\text{out}} \, \mathbb{E}[\|G_L^{t+1} - \mathbb{E}[G_L^{t+1}]\|_2^2] \\
&= \tfrac{(1 - p_{\text{out}})}{B_2} \, \mathbb{E}[\|G_{\text{E-BSGD}}^{t+1} - G_{\text{E-BSGD}}^t - \mathbb{E}[G_{\text{E-BSGD}}^{t+1} - G_{\text{E-BSGD}}^t]\|_2^2] + \tfrac{p_{\text{out}}}{B_1} \, \mathbb{E}[\|G_{\text{E-BSGD}}^{t+1} - \mathbb{E}[G_{\text{E-BSGD}}^{t+1}]\|_2^2] \\
&\leq \tfrac{1 - p_{\text{out}}}{B_2} \, \mathbb{E}[\|G_{\text{E-BSGD}}^{t+1} - G_{\text{E-BSGD}}^t - \mathbb{E}[G_{\text{E-BSGD}}^{t+1} - G_{\text{E-BSGD}}^t]\|_2^2] + \tfrac{p_{\text{out}}}{B_1} \tfrac{\sigma_{\text{in}}^2}{m}.
\end{aligned}
\tag{20}
$$

and change (18) to

$$
\mathcal{E}_{\text{var}}^1 = \mathbb{E}[\|G^1 - \mathbb{E}[G^1]\|_2^2] = \mathbb{E}[\|G_L^1 - \mathbb{E}[G_L^1]\|_2^2] = \tfrac{1}{B_1} \, \mathbb{E}[\|G_{\text{E-BSGD}}^1 - \mathbb{E}[G_{\text{E-BSGD}}^1]\|_2^2] \leq \tfrac{1}{B_1} \tfrac{\sigma_{\text{in}}^2}{m}.
\tag{21}
$$

Then our analysis only has to start from the updated version of (19)

$$
\tfrac{1}{T} \sum_{t=0}^{T-1} \mathbb{E}[\|\nabla F(\boldsymbol{x}^t)\|_2^2] \leq \tfrac{2(F(\boldsymbol{x}^0) - F^\star)}{\gamma T} + \tfrac{2\tilde{\sigma}_{\text{bias}}^2}{m^4} + \left(\tfrac{1}{T p_{\text{out}}} + 1\right) \tfrac{84}{B_1} \tfrac{\sigma_{\text{in}}^2}{m}.
$$

We would like all terms on the right-hand side to be bounded by $\varepsilon^2$. From $\frac{2\tilde{\sigma}_{\text{bias}}^2}{m^4} \leq \varepsilon^2$ we know that

$$
m = \Omega\left(\tfrac{\tilde{\sigma}_{\text{bias}}^{1/2}}{\varepsilon^{1/2}}\right).
$$

From $\left(\frac{1}{T p_{\text{out}}} + 1\right) \frac{84}{B_1} \frac{\sigma_{\text{in}}^2}{m} \leq \varepsilon^2$, we know that

$$
m = \Omega\left(\tfrac{\sigma_{\text{in}}^2}{n\varepsilon^2}\right).
$$

From $\frac{2(F(\boldsymbol{x}^0) - F^\star)}{\gamma T} \leq \varepsilon^2$, we can choose that

$$
\gamma = \mathcal{O}(\tfrac{1}{L_F}), \quad T = \Omega\left(\tfrac{L_F (F(\boldsymbol{x}^0) - F^\star)}{\varepsilon^2}\right).
$$

Now the total sample complexity for E-BSpiderBoost for the FCCO problem becomes

$$
B_2 m T = \mathcal{O}\left(L_F^2 (F(\boldsymbol{x}^0) - F^\star) \max\left\{\tfrac{\sqrt{n} \tilde{\sigma}_{\text{bias}}^{1/2}}{\varepsilon^{2.5}}, \tfrac{\sigma_{\text{in}}^2}{\sqrt{n} \varepsilon^4}\right\}\right).
$$

By picking $\boldsymbol{x}^s$ uniformly at random among $\{\boldsymbol{x}^t\}_{t=0}^{T-1}$, we get the desired guarantee. $\qquad\square$

### E.2 Convergence of NestedVR

**NestedVR Algorithm.** We start by describing the NestedVR construction. We maintain states $\boldsymbol{y}_i^{t+1}$ and $\boldsymbol{z}_i^{t+1}$ to approximate

$$
\boldsymbol{y}_i^{t+1} \approx \mathbb{E}_{\eta|i}[g_\eta(\boldsymbol{x}^t)], \quad \boldsymbol{z}_i^{t+1} \approx \mathbb{E}_{\tilde{\eta}|i}[\nabla g_{\tilde{\eta}}(\boldsymbol{x}^t)].
$$

In iteration $t + 1$, if $i$ is selected, then the state $\boldsymbol{y}_i^{t+1}$ is updated as follows

$$
\boldsymbol{y}_i^{t+1} = \begin{cases} \tfrac{1}{S_1} \sum_{\eta \in H_i} g_\eta(\boldsymbol{x}^t) & \text{with prob. } p_{\text{in}} \\ \boldsymbol{y}_i^t + \tfrac{1}{S_2} \sum_{\eta \in H_i} (g_\eta(\boldsymbol{x}^t) - g_\eta(\boldsymbol{\phi}_i^t)) & \text{with prob. } 1 - p_{\text{in}}, \end{cases}
$$

where $\phi_i^t$ is the last time node $i$ is visited. If $i$ is not selected, then
$$\boldsymbol{y}_i^{t+1} = \boldsymbol{y}_i^t.$$
In this case, $\boldsymbol{y}_i^{t+1}$ was never used to compute $\nabla f_i(\boldsymbol{y}_i^{t+1})$ because $i$ is not selected at the time $t+1$. We use the following quantities
$$\hat{\boldsymbol{z}}_i^{t+1} = \mathbb{E}_{\tilde{\eta}|i}[\nabla g_{\tilde{\eta}}(\boldsymbol{x}^t)], \qquad \boldsymbol{z}_i^{t+1} = \tfrac{1}{m} \sum_{\tilde{\eta}\in\tilde{H}_i} \nabla g_{\tilde{\eta}}(\boldsymbol{x}^t). \tag{22}$$

We use $G_{\text{NVR}}^{t+1}$ as the actual updates,
$$G_{\text{NVR}}^{t+1} = \begin{cases} \frac{1}{B_1} \sum_{i\in\mathcal{B}_1} (\boldsymbol{z}_i^{t+1})^\top \nabla f_i(\boldsymbol{y}_i^{t+1}) & \text{with prob. } p_{\text{out}} \\ G_{\text{NVR}}^t + \frac{1}{B_2} \sum_{i\in\mathcal{B}_2} ((\boldsymbol{z}_i^{t+1})^\top \nabla f_i(\boldsymbol{y}_i^{t+1}) - (\boldsymbol{z}_i^t)^\top \nabla f_i(\tilde{\boldsymbol{y}}_i^t)) & \text{with prob. } 1 - p_{\text{out}}. \end{cases}$$

We can also use the following quantity $\hat{G}_{\text{NVR}}^{t+1}$ as an auxiliary
$$\hat{G}_{\text{NVR}}^{t+1} = \begin{cases} \frac{1}{B_1} \sum_{i\in\mathcal{B}_1} (\hat{\boldsymbol{z}}_i^{t+1})^\top \nabla f_i(\boldsymbol{y}_i^{t+1}) & \text{with prob. } p_{\text{out}} \\ \hat{G}_{\text{NVR}}^t + \frac{1}{B_2} \sum_{i\in\mathcal{B}_2} ((\hat{\boldsymbol{z}}_i^{t+1})^\top \nabla f_i(\boldsymbol{y}_i^{t+1}) - (\hat{\boldsymbol{z}}_i^t)^\top \nabla f_i(\tilde{\boldsymbol{y}}_i^t)) & \text{with prob. } 1 - p_{\text{out}}. \end{cases}$$

Here we use $\tilde{\boldsymbol{y}}_i^t$ to represent an i.i.d. copy of $\boldsymbol{y}_i^t$ where $i$ is selected at time $t$.

The iterate $\boldsymbol{x}^{t+1}$ is therefore updated
$$\boldsymbol{x}^{t+1} = \boldsymbol{x}^t - \gamma G_{\text{NVR}}^{t+1}.$$

**Lemma 9** *The error between $G_{\text{NVR}}^{t+1}$ and $\hat{G}_{\text{NVR}}^{t+1}$ can be upper bounded as follows*
$$\tfrac{1}{T} \sum_{t=0}^{T-1} \mathbb{E}\left[\left\|G_{\text{NVR}}^{t+1} - \hat{G}_{\text{NVR}}^{t+1}\right\|_2^2\right] \le \tfrac{1}{B_1} \tfrac{C_f^2 \sigma_g^2}{m} + \tfrac{4(1-p_{out})}{B_2 m p_{out}} \tfrac{1}{T} \sum_{t=0}^{T-1} \left(\mathbb{E}[\|G_i^{t+1} - G_i^t\|_2^2]\right).$$

**Proof:** In this proof, we ignore the subscript in $G_{\text{NVR}}^{t+1}$ and $\hat{G}_{\text{NVR}}^{t+1}$, we bound the error between $G^{t+1}$ and associated $\hat{G}^{t+1}$ where
$$G_i^{t+1} = (\tfrac{1}{m} \sum_{\tilde{\eta}\in\tilde{H}_i} \nabla g_{\tilde{\eta}}(\boldsymbol{x}))^\top \nabla f_i(\boldsymbol{y}_i^{t+1}),$$
$$\hat{G}_i^{t+1} = (\mathbb{E}_{\tilde{\eta}|i}[\nabla g_{\tilde{\eta}}(\boldsymbol{x})])^\top \nabla f_i(\boldsymbol{y}_i^{t+1}).$$
Let's only consider the expectation over the randomness of $\nabla g_{\tilde{\eta}}$,
$$\mathbb{E}_{H_i}\left[\left\|G_i^{t+1} - \hat{G}_i^{t+1}\right\|_2^2\right] \le \mathbb{E}_{\tilde{\eta}|i}\left[\left\|(\tfrac{1}{m} \sum_{\tilde{\eta}\in\tilde{H}_i} \nabla g_{\tilde{\eta}}(\boldsymbol{x}) - \mathbb{E}_{\tilde{\eta}|i}[\nabla g_{\tilde{\eta}}(\boldsymbol{x})])\right\|_2^2\right] \mathbb{E}[\|\nabla f_i(\boldsymbol{y}_i^{t+1})\|_2^2]$$
$$\le \tfrac{C_f^2}{m} \mathbb{E}_{\tilde{\eta}|i}\left[\left\|\nabla g_{\tilde{\eta}}(\boldsymbol{x}) - \mathbb{E}_{\tilde{\eta}|i}[\nabla g_{\tilde{\eta}}(\boldsymbol{x})]\right\|_2^2\right]$$
$$\le \tfrac{C_f^2 \sigma_g^2}{m}.$$
Then we can bound the error as follows
$$\mathbb{E}\left[\mathbb{E}_{H_i}\left[\left\|G^{t+1} - \hat{G}^{t+1}\right\|_2^2\right]\right] = \tfrac{p_{\text{out}}}{B_1} \mathbb{E}\left[\mathbb{E}_{H_i}\left[\left\|G_i^{t+1} - \hat{G}_i^{t+1}\right\|_2^2\right]\right]$$
$$+ (1 - p_{\text{out}})\left(\left\|G^t - \hat{G}^t\right\|_2^2 + \tfrac{1}{B_2} \mathbb{E}\left[\mathbb{E}_{H_i}\left[\left\|G_i^{t+1} - G_i^t - \hat{G}_i^{t+1} - \hat{G}_i^t\right\|_2^2\right]\right]\right)$$
$$\le \tfrac{p_{\text{out}}}{B_1} \tfrac{C_f^2 \sigma_g^2}{m} + (1 - p_{\text{out}})\left\|G^t - \hat{G}^t\right\|_2^2$$
$$+ \tfrac{(1-p_{\text{out}})}{B_2 m}\left(\mathbb{E}\left[\|G_i^{t+1} - G_i^t\|_2^2\right]\right)$$
$$\le \tfrac{p_{\text{out}}}{B_1} \tfrac{C_f^2 \sigma_g^2}{m} + (1 - p_{\text{out}})\left\|G^t - \hat{G}^t\right\|_2^2 + \tfrac{(1-p_{\text{out}})}{B_2 m}\left(\mathbb{E}[\|G_i^{t+1} - G_i^t\|_2^2]\right).$$
Unroll the recursion gives
$$\tfrac{1}{T} \sum_{t=0}^{T-1} \mathbb{E}\left[\left\|G^{t+1} - \hat{G}^{t+1}\right\|_2^2\right] \le \tfrac{1}{B_1} \tfrac{C_f^2 \sigma_g^2}{m} + \tfrac{4(1-p_{\text{out}})}{B_2 m p_{\text{out}}} \tfrac{1}{T} \sum_{t=0}^{T-1} \mathbb{E}[\|G_i^{t+1} - G_i^t\|_2^2].$$

$\square$

**Lemma 10 (Staleness)** *Define the staleness of iterates at time $t$ as $\Xi^t := \frac{1}{n}\sum_{j=1}^n \|\boldsymbol{x}^t - \boldsymbol{\phi}_j^t\|_2^2$ and let $G^{t+1}$ be the gradient estimate, then*

$$\frac{1}{T}\sum_{t=0}^{T-1}\mathbb{E}[\Xi^t] \leq \frac{6n^2}{B_2^2}\gamma^2 \frac{1}{T}\sum_{t=0}^{T-1}\mathbb{E}[\|G^{t+1}\|_2^2]. \tag{23}$$

**Proof:** Like previously (Lemma 5), let $\mathbb{E}[\cdot]$ denote the expectation conditioned on all previous randomness until $t-1$. It is clear that $\Xi^0 = 0$, so we only consider $t > 0$. We upper bound $\mathbb{E}[\Xi^t]$ as follows,

$$\mathbb{E}[\Xi^t] = (1 - p_{\text{out}})\underbrace{\frac{1}{n}\sum_{j=1}^n \mathbb{E}[\|\boldsymbol{x}^t - \boldsymbol{\phi}_j^t\|_2^2]}_{\text{if time } t \text{ takes } \mathcal{B}_2} + p_{\text{out}}\underbrace{\frac{1}{n}\sum_{j=1}^n \mathbb{E}[\|\boldsymbol{x}^t - \boldsymbol{\phi}_j^t\|_2^2]}_{\text{if time } t \text{ takes } \mathcal{B}_1(\boldsymbol{\phi}_j^t = \boldsymbol{x}^{t-1})}.$$

Then we can expand $\mathbb{E}[\Xi^t]$ as follows

$$\mathbb{E}[\Xi^t] = \frac{1 - p_{\text{out}}}{n}\sum_{j=1}^n \mathbb{E}[\|\boldsymbol{x}^t - \boldsymbol{\phi}_j^t\|_2^2] + \frac{p_{\text{out}}}{n}\sum_{j=1}^n \mathbb{E}[\|\boldsymbol{x}^t - \boldsymbol{x}^{t-1}\|_2^2]$$

$$\leq \frac{1 - p_{\text{out}}}{n}\sum_{j=1}^n \left((1 + \tfrac{1}{\beta})\,\mathbb{E}_i[\|\boldsymbol{x}^{t-1} - \boldsymbol{\phi}_j^t\|_2^2] + (1 + \beta)\,\|\boldsymbol{x}^{t-1} - \boldsymbol{x}^t\|_2^2\right) + p_{\text{out}}\gamma^2\,\mathbb{E}[\|G^t\|_2^2]$$

$$\leq \frac{1}{n}\sum_{j=1}^n (1 + \tfrac{1}{\beta})\,\mathbb{E}_i[\|\boldsymbol{x}^{t-1} - \boldsymbol{\phi}_j^t\|_2^2] + (1 + \beta)\gamma^2\,\mathbb{E}[\|G^t\|_2^2]$$

where we use Cauchy-Schwarz inequality with coefficient $\beta > 0$. By the definition of $\boldsymbol{\phi}_j^t$,

$$\mathbb{E}[\Xi^t] \leq \frac{1}{n}\sum_{j=1}^n (1 + \tfrac{1}{\beta})\left(\frac{n - B_2}{n}\,\|\boldsymbol{x}^{t-1} - \boldsymbol{\phi}_j^{t-1}\|_2^2 + \frac{B_2}{n}\,\|\boldsymbol{x}^{t-1} - \boldsymbol{x}^{t-1}\|_2^2\right) + (1 + \beta)\gamma^2\,\mathbb{E}[\|G^t\|_2^2]$$

$$= (1 + \tfrac{1}{\beta})(1 - \tfrac{B_2}{n})\Xi^{t-1} + (1 + \beta)\gamma^2\,\mathbb{E}[\|G^t\|_2^2].$$

By taking $\beta = 2n/B_2$, we have that $(1 + \frac{1}{\beta})(1 - \frac{B_2}{n}) \leq 1 - \frac{B_2}{2n}$ and thus

$$\mathbb{E}[\Xi^t] \leq (1 - \tfrac{B_2}{2n})\Xi^{t-1} + (1 + \tfrac{2n}{B_2})\gamma^2\,\mathbb{E}[\|G^t\|_2^2].$$

Note that $\mathbb{E}[\Xi^0] = 0$.

$$\frac{1}{T}\sum_{t=0}^{T-1}\mathbb{E}[\Xi^t] \leq \frac{2n}{B_2}(1 + \tfrac{2n}{B_2})\gamma^2 \frac{1}{T}\sum_{t=0}^{T-1}\mathbb{E}[\|G^{t+1}\|_2^2]$$

$$\leq \frac{6n^2}{B_2^2}\gamma^2 \frac{1}{T}\sum_{t=0}^{T-1}\mathbb{E}[\|G^{t+1}\|_2^2].$$

$\square$

The following lemma describes how the inner variable changes inside the variance.

**Lemma 11** *Denote $\mathcal{E}_y^{t+1} := \mathbb{E}\left[\|\boldsymbol{y}_i^{t+1} - \mathbb{E}_{\eta|i}[g_\eta(\boldsymbol{x}^t)]\|_2^2\right]$ to be the error from inner variance and $p_{out}T \leq 1$. Then*

$$\frac{1}{T}\sum_{t=0}^{T-1}\mathcal{E}_y^{t+1} \leq \frac{(1 - p_{in})C_g^2}{p_{in}S_2}\frac{1}{T}\sum_{t=0}^{T-1}\Xi^t + \frac{2\sigma_g^2}{S_1}.$$

*Meanwhile, $\mathcal{E}_y^1 = \mathbb{E}[\|\boldsymbol{y}_i^1 - \mathbb{E}_{\eta|i}[g_\eta(\boldsymbol{x}^0)]\|_2^2] = \frac{\sigma_g^2}{S_1}$.*

**Proof:**

$$\mathcal{E}_y^{t+1} \leq p_{\text{in}}\frac{\sigma_g^2}{S_1} + (1 - p_{\text{in}})\,\mathbb{E}_i[\mathbb{E}_{\eta|i}[\|\boldsymbol{y}_i^t - \mathbb{E}_{\eta|i}[g_\eta(\boldsymbol{\phi}_i^t)]\|_2^2]]$$

$$+ \frac{1 - p_{\text{in}}}{S_2}\,\mathbb{E}_i[\mathbb{E}_{\eta|i}[\|g_\eta(\boldsymbol{x}^t) - g_\eta(\boldsymbol{\phi}_i^t)\|_2^2]]$$

$$\leq (1 - p_{\text{in}})\mathcal{E}_y^t + \frac{(1 - p_{\text{in}})C_g^2}{S_2}\,\mathbb{E}_i[\mathbb{E}_{\eta|i}[\|\boldsymbol{x}^t - \boldsymbol{\phi}_i^t\|_2^2]] + p_{\text{in}}\frac{\sigma_g^2}{S_1}.$$

As $t = 0$ always uses the large batch, $\mathcal{E}_y^1 = \mathbb{E}[\|\boldsymbol{y}_i^1 - \mathbb{E}_{\eta|i}[g_\eta(\boldsymbol{x}^0)]\|_2^2] = \frac{\sigma_g^2}{S_1}$. Then

$$\frac{1}{T}\sum_{t=0}^{T-1}\mathcal{E}_y^{t+1} \leq \frac{(1-p_{\text{in}})C_g^2}{p_{\text{in}}S_2}\frac{1}{T}\sum_{t=0}^{T-1}\mathbb{E}_i[\mathbb{E}_{\eta|i}[\|\boldsymbol{x}^t - \boldsymbol{\phi}_i^t\|_2^2]] + \frac{\sigma_g^2}{S_1} + \frac{\mathcal{E}_y^1}{p_{\text{in}}T}$$
$$\leq \frac{(1-p_{\text{in}})C_g^2}{p_{\text{in}}S_2}\frac{1}{T}\sum_{t=0}^{T-1}\Xi^t + \frac{2\sigma_g^2}{S_1}.$$

$\square$

**Lemma 12** *The error* $\mathbb{E}_i[\mathbb{E}_{p_{in}}[\mathbb{E}_{\eta|i}[\|\boldsymbol{y}_i^{t+1} - \tilde{\boldsymbol{y}}_i^t\|_2^2]]]$ *satisfies*

$$\frac{1}{T}\sum_{t=1}^{T-1}\mathbb{E}_i[\mathbb{E}_{p_{in}}[\mathbb{E}_{\eta|i}[\|\boldsymbol{y}_i^{t+1} - \tilde{\boldsymbol{y}}_i^t\|_2^2]]] \leq \frac{4C_g^2\gamma^2}{T}\sum_{t=0}^{T-1}\mathbb{E}[\|G^{t+1}\|_2^2] + \frac{4(1-p_{in})C_g^2}{S_2}\frac{1}{T}\sum_{t=0}^{T-1}\Xi^{t+1}$$
$$+ \frac{6(1-p_{in})}{T}\sum_{t=0}^{T-1}\mathcal{E}_y^{t+1}.$$

Note that when $p_{\text{in}} = 1$ and $S_1 = S_2 = m$, we recover the following

$$\frac{1}{T}\sum_{t=1}^{T-1}\mathbb{E}_i[\mathbb{E}_{p_{in}}[\mathbb{E}_{\eta|i}[\|\boldsymbol{y}_i^{t+1} - \tilde{\boldsymbol{y}}_i^t\|_2^2]]] = \frac{1}{T}\sum_{t=1}^{T-1}\mathbb{E}_i[\mathbb{E}_{p_{in}}[\mathbb{E}_{\eta|i}[\|\frac{1}{m}\sum_{\eta\in H_\xi}g_\eta(\boldsymbol{x}^t) - g_\eta(\boldsymbol{x}^{t-1})\|_2^2]]]$$
$$\leq \frac{4C_g^2\gamma^2}{T}\sum_{t=0}^{T-1}\mathbb{E}[\|G^{t+1}\|_2^2].$$

**Proof:** For $t \geq 2$, $\mathbb{E}_i[\mathbb{E}_{p_{in}}[\mathbb{E}_{\eta|i}[\|\boldsymbol{y}_i^{t+1} - \tilde{\boldsymbol{y}}_i^t\|_2^2]]]$ can be upper bounded as follows

$$\mathbb{E}_i[\mathbb{E}_{p_{in}}[\mathbb{E}_{\eta|i}[\|\boldsymbol{y}_i^{t+1} - \tilde{\boldsymbol{y}}_i^t\|_2^2]]] = p_{\text{in}}\mathbb{E}_i\left[\mathbb{E}_{\eta|i}\left[\left\|\frac{1}{S_1}\sum_{\eta\in H_i}(g_\eta(\boldsymbol{x}^t) - g_\eta(\boldsymbol{x}^{t-1}))\right\|_2^2\right]\right]$$

$$+ (1-p_{\text{in}})\mathbb{E}_i\left[\mathbb{E}_{\eta|i}\left[\left\|\boldsymbol{y}_i^t - \boldsymbol{y}_i^{t-1} + \frac{1}{S_2}\sum_{\eta\in H_i}(g_\eta(\boldsymbol{x}^t) - g_\eta(\boldsymbol{\phi}_i^t)) - (g_\eta(\boldsymbol{x}^{t-1}) - g_\eta(\boldsymbol{\phi}_i^{t-1}))\right\|_2^2\right]\right]$$

$$\leq p_{\text{in}}C_g^2\|\boldsymbol{x}^t - \boldsymbol{x}^{t-1}\|_2^2 + \frac{1-p_{\text{in}}}{S_2}\mathbb{E}_i\left[\mathbb{E}_{\eta|i}\left[\|(g_\eta(\boldsymbol{x}^t) - g_\eta(\boldsymbol{\phi}_i^t)) - (g_\eta(\boldsymbol{x}^{t-1}) - g_\eta(\boldsymbol{\phi}_i^{t-1}))\|_2^2\right]\right]$$

$$+ 3(1-p_{\text{in}})\left(\mathbb{E}_i\left[\|\boldsymbol{y}_i^t - \mathbb{E}_{\eta|i}[g_\eta(\boldsymbol{\phi}_i^t)]\|_2^2\right] + \mathbb{E}_i\left[\|\boldsymbol{y}_i^{t-1} - \mathbb{E}_{\eta|i}[g_\eta(\boldsymbol{\phi}_i^{t-1})]\|_2^2\right] + C_g^2\|\boldsymbol{x}^t - \boldsymbol{x}^{t-1}\|_2^2\right)$$

$$\leq p_{\text{in}}C_g^2\|\boldsymbol{x}^t - \boldsymbol{x}^{t-1}\|_2^2 + \frac{2(1-p_{\text{in}})C_g^2}{S_2}\left(\Xi^t + \Xi^{t-1}\right) + 3(1-p_{\text{in}})\left(\mathcal{E}_y^t + \mathcal{E}_y^{t-1} + C_g^2\|\boldsymbol{x}^t - \boldsymbol{x}^{t-1}\|_2^2\right)$$

$$\leq (p_{\text{in}} + 3(1-p_{\text{in}}))C_g^2\|\boldsymbol{x}^t - \boldsymbol{x}^{t-1}\|_2^2 + \frac{2(1-p_{\text{in}})C_g^2}{S_2}\left(\Xi^t + \Xi^{t-1}\right) + 3(1-p_{\text{in}})\left(\mathcal{E}_y^t + \mathcal{E}_y^{t-1}\right).$$

For $t = 1$, we choose $\tilde{\boldsymbol{y}}_i^1 = \boldsymbol{y}_i^1$

$$\mathbb{E}_i[\mathbb{E}_{p_{in}}[\mathbb{E}_{\eta|i}[\|\boldsymbol{y}_i^2 - \tilde{\boldsymbol{y}}_i^1\|_2^2]]] = p_{\text{in}}\mathbb{E}_i\left[\mathbb{E}_{\eta|i}\left[\left\|\frac{1}{S_1}\sum_{\eta\in H_i}(g_\eta(\boldsymbol{x}^1) - g_\eta(\boldsymbol{x}^0))\right\|_2^2\right]\right]$$

$$+ (1-p_{\text{in}})\mathbb{E}_i\left[\mathbb{E}_{\eta|i}\left[\left\|\boldsymbol{y}_i^1 - \frac{1}{S_2}\sum_{\eta\in H_i}(g_\eta(\boldsymbol{x}^1) - g_\eta(\boldsymbol{x}^0)) - \tilde{\boldsymbol{y}}_i^1\right\|_2^2\right]\right]$$

$$\leq C_g^2\|\boldsymbol{x}^1 - \boldsymbol{x}^0\|_2^2.$$

Then for summing up $t = 1$ to $T - 1$

$$\sum_{t=2}^{T-1}\mathbb{E}_i[\mathbb{E}_{p_{in}}[\mathbb{E}_{\eta|i}[\|\boldsymbol{y}_i^{t+1} - \tilde{\boldsymbol{y}}_i^t\|_2^2]]] + \mathbb{E}_i[\mathbb{E}_{p_{in}}[\mathbb{E}_{\eta|i}[\|\boldsymbol{y}_i^2 - \tilde{\boldsymbol{y}}_i^1\|_2^2]]]$$

$$\leq (p_{\text{in}} + 3(1-p_{\text{in}}))C_g^2\sum_{t=2}^{T-1}\|\boldsymbol{x}^t - \boldsymbol{x}^{t-1}\|_2^2 + \frac{2(1-p_{\text{in}})C_g^2}{S_2}\left(\sum_{t=2}^{T-1}\Xi^t + \sum_{t=2}^{T-1}\Xi^{t-1}\right)$$

$$+ 3(1-p_{\text{in}})\left(\sum_{t=2}^{T-1}\mathcal{E}_y^t + \sum_{t=2}^{T-1}\mathcal{E}_y^{t-1}\right) + C_g^2\|\boldsymbol{x}^1 - \boldsymbol{x}^0\|_2^2$$

$$\leq 4C_g^2\sum_{t=1}^{T-1}\|\boldsymbol{x}^t - \boldsymbol{x}^{t-1}\|_2^2 + \frac{4(1-p_{\text{in}})C_g^2}{S_2}\sum_{t=0}^{T-1}\Xi^{t+1} + 6(1-p_{\text{in}})\sum_{t=0}^{T-1}\mathcal{E}_y^{t+1}.$$

Finally, the error has the following upper bound

$$\frac{1}{T}\sum_{t=1}^{T-1}\mathbb{E}_i[\mathbb{E}_{p_{in}}[\mathbb{E}_{\eta|i}[\|\boldsymbol{y}_i^{t+1} - \tilde{\boldsymbol{y}}_i^t\|_2^2]]]$$
$$\leq \frac{4C_g^2\gamma^2}{T}\sum_{t=0}^{T-1}\mathbb{E}[\|G^{t+1}\|_2^2] + \frac{4(1-p_{\text{in}})C_g^2}{S_2}\frac{1}{T}\sum_{t=0}^{T-1}\Xi^{t+1} + \frac{6(1-p_{\text{in}})}{T}\sum_{t=0}^{T-1}\mathcal{E}_y^{t+1}.$$

$\square$

**Lemma 13 (Bias and Variance of NestedVR)** *If the step size $\gamma$ satisfies,*

$$\gamma^2 L_F^2 \max\left\{ \frac{(1-p_{in})}{p_{in}S_2} \frac{18}{B_2}, \frac{1-p_{out}}{B_2} \frac{(1-p_{in})}{p_{in}S_2} \frac{18n^2}{B_2^2}, \frac{(1-p_{out})}{B_2} \right\} \le \frac{1}{16} \cdot \frac{1}{6}$$

*then the variance and bias of NestedVR are*

$$\frac{1}{T}\sum_{t=0}^{T-1} \mathcal{E}_{var}^{t+1} \le 32 \left( \left( \frac{p_{out}}{B_1} + \frac{1-p_{out}}{B_2} \right) \frac{(1-p_{in})}{p_{in}S_2} \frac{18n^2}{B_2^2} + \frac{(1-p_{out})}{B_2} \right) \frac{\gamma^2 L_F^2}{T} \sum_{t=0}^{T-1} \left\| \mathbb{E}[G^{t+1}] \right\|_2^2$$

$$+ 96 \left( \frac{p_{out}}{B_1} + \frac{(1-p_{in})(1-p_{out})}{B_2} \right) \frac{\tilde{L}_F^2}{S_1} + \frac{(1-p_{out})}{T} \frac{8\tilde{L}_F^2}{B_1 S_1}$$

$$\frac{1}{T}\sum_{t=0}^{T-1} \mathcal{E}_{bias}^{t+1} \le \frac{12(1-p_{in})}{p_{in}S_2} \frac{n^2}{B_2^2} \frac{L_F^2 \gamma^2}{T} \sum_{t=0}^{T-1} \left\| \mathbb{E}[G^{t+1}] \right\|_2^2 + \frac{4\tilde{L}_F^2}{S_1}$$

$$+ \left( \frac{12(1-p_{in})}{p_{in}S_2} \frac{n^2}{B_2^2} L_F^2 \gamma^2 + \frac{2(1-p_{out})^2}{p_{out}} \right) \frac{1}{T} \sum_{t=0}^{T-1} \mathcal{E}_{var}^{t+1}.$$

**Proof: Notations.** Let us define the following terms,

$$G_i^{t+1} := (z_i^{t+1})^\top \nabla f_i(y_i^{t+1}), \quad \tilde{G}_i^t := (z_i^t)^\top \nabla f_i(\tilde{y}_i^t).$$

Note that the $\tilde{G}^t$ computed at time $t+1$ has same expectation as $G^t$

$$\mathbb{E}^{t+1}[\tilde{G}^t|t] = \mathbb{E}^t[G^t|t-1]. \tag{24}$$

**Computing the bias.** First consider the two cases in the outer loop

$$\mathcal{E}_{bias}^{t+1} = \left\| \nabla F(x^t) - \mathbb{E}^{t+1}[G^{t+1}|t] \right\|_2^2$$

$$\le 2 \underbrace{\left\| \nabla F(x^t) - \mathbb{E}^{t+1}[G_i^{t+1}|t] \right\|_2^2}_{A_1^{t+1}} + 2 \underbrace{\left\| \mathbb{E}^{t+1}[G_i^{t+1}|t] - \mathbb{E}^{t+1}[G^{t+1}|t] \right\|_2^2}_{A_2^{t+1}}.$$

We expand $A_2^{t+1}$ as follows

$$A_2^{t+1} = \left\| \mathbb{E}^{t+1}[G_i^{t+1}|t] - \mathbb{E}^{t+1}[G^{t+1}|t] \right\|_2^2$$

$$= \left\| \mathbb{E}^{t+1}[G_i^{t+1}|t] - p_{out}\,\mathbb{E}^{t+1}[G_i^{t+1}|t] - (1-p_{out})(G^t + \mathbb{E}^{t+1}[G_i^{t+1} - \tilde{G}_i^t|t]) \right\|_2^2$$

$$= (1-p_{out})^2 \left\| G^t - \mathbb{E}^{t+1}[\tilde{G}_i^t|t] \right\|_2^2$$

$$= (1-p_{out})^2 \left\| G^t - \mathbb{E}^t[G_i^t|t-1] \right\|_2^2$$

where we use (24) in the last equality. Now we take expectation with respect to randomness at $t$ such that $G^t$ is a random variable, then

$$A_2^{t+1} = (1-p_{out})^2\, \mathbb{E}^t\left[ \left\| G^t - \mathbb{E}^t[G_i^t|t-1] \right\|_2^2 |t-1 \right]$$

$$= (1-p_{out})^2 \left( \left\| \mathbb{E}^t[G^t|t-1] - \mathbb{E}^t[G_i^t|t-1] \right\|_2^2 + \mathcal{E}_{var}^t \right)$$

$$= (1-p_{out})^2 \left( A_2^t + \mathcal{E}_{var}^t \right)$$

while at initialization we always use large batch

$$A_2^1 = \left\| \mathbb{E}^1[G_i^1] - \mathbb{E}^1[G^1] \right\|_2^2 = \left\| \mathbb{E}^1[G_i^1] - \mathbb{E}^1[G_i^1] \right\|_2^2 = 0.$$

Therefore, when we average over time $t$

$$\frac{1}{T}\sum_{t=0}^{T-1} A_2^{t+1} \le \frac{(1-p_{out})^2}{p_{out}} \frac{1}{T}\sum_{t=0}^{T-1} \mathcal{E}_{var}^{t+1}. \tag{25}$$

On the other hand, let us consider the upper bound on $A_1^{t+1}$

$$A_1^{t+1} \le C_g^2 L_f^2\, \mathbb{E}[\| y_i^{t+1} - \mathbb{E}_{\eta|i}[g_\eta(x^t)] \|_2^2] = C_g^2 L_f^2 \mathcal{E}_y^{t+1}.$$

From Lemma 11 we know that

$$\frac{1}{T}\sum_{t=0}^{T-1} A_1^{t+1} \le C_g^2 L_f^2 \left( \frac{(1-p_{in})C_g^2}{p_{in}S_2} \frac{1}{T}\sum_{t=0}^{T-1} \Xi^t + \frac{2\sigma_g^2}{S_1} \right)$$

$$\le \frac{(1-p_{in})L_F^2}{p_{in}S_2} \frac{1}{T}\sum_{t=0}^{T-1} \Xi^t + \frac{2\tilde{L}_F^2}{S_1}.$$

From Lemma 10 we know that

$$\frac{1}{T}\sum_{t=0}^{T-1} A_1^{t+1} \leq \frac{(1-p_{\text{in}})L_F^2}{p_{\text{in}}S_2}\left(\frac{6n^2}{B_2^2}\gamma^2 \frac{1}{T}\sum_{t=0}^{T-1}\mathbb{E}[\|G^{t+1}\|_2^2]\right) + \frac{2\tilde{L}_F^2}{S_1}$$

$$= \frac{6(1-p_{\text{in}})}{p_{\text{in}}S_2}\frac{n^2}{B_2^2}\frac{L_F^2\gamma^2}{T}\sum_{t=0}^{T-1}\|\mathbb{E}[G^{t+1}]\|_2^2 + \frac{2\tilde{L}_F^2}{S_1} + \frac{6(1-p_{\text{in}})}{p_{\text{in}}S_2}\frac{n^2}{B_2^2}\frac{L_F^2\gamma^2}{T}\sum_{t=0}^{T-1}\mathcal{E}_{\text{var}}^{t+1}.$$

Therefore, the bias has the following bound

$$\frac{1}{T}\sum_{t=0}^{T-1}\mathcal{E}_{\text{bias}}^{t+1} \leq \frac{12(1-p_{\text{in}})}{p_{\text{in}}S_2}\frac{n^2}{B_2^2}\frac{L_F^2\gamma^2}{T}\sum_{t=0}^{T-1}\|\mathbb{E}[G^{t+1}]\|_2^2 + \frac{4\tilde{L}_F^2}{S_1}$$

$$+ \left(\frac{12(1-p_{\text{in}})}{p_{\text{in}}S_2}\frac{n^2}{B_2^2}L_F^2\gamma^2 + \frac{2(1-p_{\text{out}})^2}{p_{\text{out}}}\right)\frac{1}{T}\sum_{t=0}^{T-1}\mathcal{E}_{\text{var}}^{t+1}. \tag{26}$$

Note that when $p_{\text{in}} = 1$ and $S_1 = S_2 = m$, then this bias recovers BSpiderBoost in (16)

$$\frac{1}{T}\sum_{t=0}^{T-1}\mathcal{E}_{\text{bias}}^{t+1} \leq \frac{4\tilde{L}_F^2}{m} + \frac{2(1-p_{\text{out}})^2}{p_{\text{out}}}\frac{1}{T}\sum_{t=0}^{T-1}\mathcal{E}_{\text{var}}^{t+1}.$$

**Computing the variance.** Let us decompose the variance into 3 parts:

$$\mathcal{E}_{\text{var}}^{t+1} = \mathbb{E}\left[\|G^{t+1} - \mathbb{E}[G^{t+1}]\|_2^2\right]$$

$$= \mathbb{E}\left[\|G^{t+1} \pm \hat{G}^{t+1} \pm \mathbb{E}_{\eta|i}[\hat{G}^{t+1}] - \mathbb{E}_i[\mathbb{E}_{\eta|i}[\hat{G}^{t+1}]]\|_2^2\right]$$

$$= \underbrace{\mathbb{E}\left[\|G^{t+1} - \hat{G}^{t+1}\|_2^2\right]}_{\mathcal{E}_{\nabla g}^{t+1}} + \underbrace{\mathbb{E}_i[\|\mathbb{E}_{\eta|i}[G^{t+1}] - \mathbb{E}_i[\mathbb{E}_{\eta|i}[G^{t+1}]]\|_2^2]}_{\mathcal{E}_{\text{var,out}}^{t+1}} + \underbrace{\mathbb{E}[\|G^{t+1} - \mathbb{E}_{\eta|i}[G^{t+1}]\|_2^2]}_{\mathcal{E}_{\text{var,in}}^{t+1}}$$

where $\mathcal{E}_{\text{var,out}}^{t+1}$ and $\mathcal{E}_{\text{var,in}}^{t+1}$ are the variance of outer loop and inner loop.

**Inner Variance.** For $t \geq 1$, we expand the inner variance

$$\mathcal{E}_{\text{var,in}}^{t+1} = p_{\text{out}}\,\mathbb{E}\left[\left\|\frac{1}{B_1}\sum_i(\mathbb{E}_{\tilde{\eta}|i}[\nabla g_{\tilde{\eta}}(\boldsymbol{x}^t)])^\top(\nabla f_i(\boldsymbol{y}_i^{t+1}) - \mathbb{E}_{\eta|i}[\nabla f_i(\boldsymbol{y}_i^{t+1})])\right\|_2^2\right]$$

$$+ (1 - p_{\text{out}})\,\mathbb{E}\left[\left\|\frac{1}{B_2}\sum_i(G_i^{t+1} - \tilde{G}_i^t) - \mathbb{E}_{\eta|i}[G_i^{t+1} - \tilde{G}_i^t]\right\|_2^2\right]$$

$$\leq \frac{p_{\text{out}}}{B_1}C_g^2\,\mathbb{E}\left[\|\nabla f_i(\boldsymbol{y}_i^{t+1}) - \mathbb{E}_{\eta|i}[\nabla f_i(\boldsymbol{y}_i^{t+1})]\|_2^2\right] + \frac{1-p_{\text{out}}}{B_2}\mathbb{E}_i\left[\mathbb{E}_{\eta|i}\left[\|G_i^{t+1} - \tilde{G}_i^t\|_2^2\right]\right] \tag{27}$$

$$\leq \frac{p_{\text{out}}}{B_1}4C_g^2 L_f^2 \mathcal{E}_y^{t+1} + \frac{1-p_{\text{out}}}{B_2}\mathbb{E}_i\left[\mathbb{E}_{p_{\text{in}}}\left[\mathbb{E}_{\eta|i}\left[\|G_i^{t+1} - \tilde{G}_i^t\|_2^2\right]\right]\right].$$

We bound the outer variance as

$$\mathbb{E}_i\left[\mathbb{E}_{p_{\text{in}}}\left[\mathbb{E}_{\eta|i}\left[\|G_i^{t+1} - \tilde{G}_i^t\|_2^2\right]\right]\right] = \mathbb{E}_i\left[\mathbb{E}_{p_{\text{in}}}\left[\mathbb{E}_{\eta|i}\left[\|G_i^{t+1} \pm (\mathbb{E}_{\tilde{\eta}|i}[\nabla g_{\tilde{\eta}}(\boldsymbol{x}^{t-1})])^\top\nabla f_i(\boldsymbol{y}_i^{t+1}) - \tilde{G}_i^t\|_2^2\right]\right]\right]$$

$$\leq 2\,\mathbb{E}_i\left[\mathbb{E}_{p_{\text{in}}}\left[\mathbb{E}_{\eta|i}\left[\|G_i^{t+1} - (\mathbb{E}_{\tilde{\eta}|i}[\nabla g_{\tilde{\eta}}(\boldsymbol{x}^{t-1})])^\top\nabla f_i(\boldsymbol{y}_i^{t+1})\|_2^2\right]\right]\right]$$

$$+ 2\,\mathbb{E}_i\left[\mathbb{E}_{p_{\text{in}}}\left[\mathbb{E}_{\eta|i}\left[\|(\mathbb{E}_{\tilde{\eta}|i}[\nabla g_{\tilde{\eta}}(\boldsymbol{x}^{t-1})])^\top\nabla f_i(\boldsymbol{y}_i^{t+1}) - \tilde{G}_i^t\|_2^2\right]\right]\right]$$

$$\leq 2C_f^2 L_g^2 \|\boldsymbol{x}^t - \boldsymbol{x}^{t-1}\|_2^2 + 2C_g^2 L_f^2 \,\mathbb{E}_i[\mathbb{E}_{p_{\text{in}}}[\mathbb{E}_{\eta|i}[\|\boldsymbol{y}_i^{t+1} - \tilde{\boldsymbol{y}}_i^t\|_2^2]]]. \tag{28}$$

For $t = 0$, as we only use large and small batch in the

$$\mathcal{E}_{\text{var,in}}^1 = \mathbb{E}\left[\left\|\frac{1}{B_1}\sum_i(\mathbb{E}_{\tilde{\eta}|i}[\nabla g_{\tilde{\eta}}(\boldsymbol{x}^0)])^\top(\nabla f_i(\boldsymbol{y}_i^1) - \mathbb{E}_{\eta|i}[\nabla f_i(\boldsymbol{y}_i^1)])\right\|_2^2\right]$$

$$\leq \frac{1}{B_1}C_g^2\,\mathbb{E}[\|\nabla f_i(\boldsymbol{y}_i^1) - \mathbb{E}_{\eta|i}[\nabla f_i(\boldsymbol{y}_i^1)]\|_2^2]$$

$$\leq \frac{1}{B_1}4C_g^2 L_f^2 \mathcal{E}_y^1 \tag{29}$$

$$\leq \frac{1}{B_1}4C_g^2 L_f^2 \frac{\sigma_g^2}{S_1}$$

$$\leq \frac{4\tilde{L}_F^2}{B_1 S_1}.$$

Therefore, average over time $t = 0, \ldots T - 1$ gives

$$\frac{1}{T}\sum_{t=0}^{T-1}\mathcal{E}_{\text{var,in}}^{t+1} \le \frac{p_{\text{out}}}{B_1}4C_g^2 L_f^2 \frac{1}{T}\sum_{t=1}^{T-1}\mathcal{E}_y^{t+1} + \frac{1-p_{\text{out}}}{B_2}\frac{1}{T}\sum_{t=1}^{T-1}\mathbb{E}_i[\mathbb{E}_{p_{\text{in}}}[\mathbb{E}_{\eta|i}[\|G_i^{t+1} - \tilde{G}_i^t\|_2^2]]] + \frac{\mathcal{E}_{\text{var,in}}^1}{T}$$

$$= \frac{p_{\text{out}}}{B_1}4C_g^2 L_f^2 \frac{1}{T}\sum_{t=0}^{T-1}\mathcal{E}_y^{t+1} + \frac{1-p_{\text{out}}}{B_2}\frac{1}{T}\sum_{t=1}^{T-1}\mathbb{E}_i\left[\mathbb{E}_{p_{\text{in}}}\left[\mathbb{E}_{\eta|i}\left[\|G_i^{t+1} - \tilde{G}_i^t\|_2^2\right]\right]\right] + \frac{(1-p_{\text{out}})\mathcal{E}_{\text{var,in}}^1}{T}$$

$$\le \frac{p_{\text{out}}}{B_1}4C_g^2 L_f^2 \frac{1}{T}\sum_{t=0}^{T-1}\mathcal{E}_y^{t+1} + \frac{(1-p_{\text{out}})\mathcal{E}_{\text{var,in}}^1}{T}$$
$$+ \frac{2(1-p_{\text{out}})}{B_2}\left(\frac{C_f^2 L_g^2}{T}\sum_{t=1}^{T-1}\|x^t - x^{t-1}\|_2^2 + C_g^2 L_f^2 \frac{1}{T}\sum_{t=1}^{T-1}\mathbb{E}_i[\mathbb{E}_{p_{\text{in}}}[\mathbb{E}_{\eta|i}[\|y_i^{t+1} - \tilde{y}_i^t\|_2^2]]]\right)$$

$$\le \frac{p_{\text{out}}}{B_1}4C_g^2 L_f^2 \frac{1}{T}\sum_{t=0}^{T-1}\mathcal{E}_y^{t+1} + \frac{(1-p_{\text{out}})\mathcal{E}_{\text{var,in}}^1}{T}$$
$$+ \frac{2(1-p_{\text{out}})}{B_2}\frac{C_f^2 L_g^2 \gamma^2}{T}\sum_{t=0}^{T-1}\mathbb{E}[\|G^{t+1}\|_2^2]$$
$$+ \frac{2(1-p_{\text{out}})}{B_2}C_g^2 L_f^2 \frac{1}{T}\sum_{t=1}^{T-1}\mathbb{E}_i[\mathbb{E}_{p_{\text{in}}}[\mathbb{E}_{\eta|i}[\|y_i^{t+1} - \tilde{y}_i^t\|_2^2]]].$$

Let us first apply Lemma 12

$$\frac{1}{T}\sum_{t=0}^{T-1}\mathcal{E}_{\text{var,in}}^{t+1} \le \frac{p_{\text{out}}}{B_1}4C_g^2 L_f^2 \frac{1}{T}\sum_{t=0}^{T-1}\mathcal{E}_y^{t+1} + \frac{(1-p_{\text{out}})\mathcal{E}_{\text{var,in}}^1}{T} + \frac{2(1-p_{\text{out}})}{B_2}\frac{C_f^2 L_g^2 \gamma^2}{T}\sum_{t=0}^{T-1}\mathbb{E}[\|G^{t+1}\|_2^2]$$
$$+ \frac{2(1-p_{\text{out}})C_g^2 L_f^2}{B_2}\left(\frac{4C_g^2\gamma^2}{T}\sum_{t=0}^{T-1}\mathbb{E}[\|G^{t+1}\|_2^2] + \frac{4(1-p_{\text{in}})C_g^2}{S_2}\frac{1}{T}\sum_{t=0}^{T-1}\Xi^t + \frac{6(1-p_{\text{in}})}{T}\sum_{t=0}^{T-1}\mathcal{E}_y^{t+1}\right)$$
$$\le \left(\frac{p_{\text{out}}}{B_1} + \frac{(1-p_{\text{in}})(1-p_{\text{out}})}{B_2}\right)\frac{12C_g^2 L_f^2}{T}\sum_{t=0}^{T-1}\mathcal{E}_y^{t+1} + \frac{8(1-p_{\text{out}})L_F^2 \gamma^2}{B_2}\frac{1}{T}\sum_{t=0}^{T-1}\mathbb{E}[\|G^{t+1}\|_2^2]$$
$$+ \frac{8(1-p_{\text{out}})(1-p_{\text{in}})C_g^4 L_f^2}{B_2 S_2}\frac{1}{T}\sum_{t=0}^{T-1}\Xi^t + \frac{(1-p_{\text{out}})\mathcal{E}_{\text{var,in}}^1}{T}$$

Then we apply Lemma 11 on the bound of $\frac{1}{T}\sum_{t=0}^{T-1}\mathcal{E}_y^{t+1}$

$$\frac{1}{T}\sum_{t=0}^{T-1}\mathcal{E}_{\text{var,in}}^{t+1} \le 24\left(\frac{p_{\text{out}}}{B_1} + \frac{(1-p_{\text{in}})(1-p_{\text{out}})}{B_2}\right)\left(\frac{(1-p_{\text{in}})L_F^2}{p_{\text{in}}S_2}\frac{1}{T}\sum_{t=0}^{T-1}\Xi^t + \frac{\tilde{L}_F^2}{S_1}\right)$$
$$+ \frac{8(1-p_{\text{out}})L_F^2 \gamma^2}{B_2}\frac{1}{T}\sum_{t=0}^{T-1}\mathbb{E}[\|G^{t+1}\|_2^2]$$
$$+ \frac{8(1-p_{\text{out}})(1-p_{\text{in}})C_g^4 L_f^2}{B_2 S_2}\frac{1}{T}\sum_{t=0}^{T-1}\Xi^t + \frac{(1-p_{\text{out}})\mathcal{E}_{\text{var,in}}^1}{T}$$
$$\le 24\left(\frac{p_{\text{out}}}{B_1} + \frac{(1-p_{\text{in}})(1-p_{\text{out}})}{B_2} + \frac{p_{\text{in}}(1-p_{\text{out}})}{B_2}\right)\frac{(1-p_{\text{in}})L_F^2}{p_{\text{in}}S_2}\frac{1}{T}\sum_{t=0}^{T-1}\Xi^t$$
$$+ \frac{8(1-p_{\text{out}})L_F^2 \gamma^2}{B_2}\frac{1}{T}\sum_{t=0}^{T-1}\mathbb{E}[\|G^{t+1}\|_2^2]$$
$$+ 24\left(\frac{p_{\text{out}}}{B_1} + \frac{(1-p_{\text{in}})(1-p_{\text{out}})}{B_2}\right)\frac{\tilde{L}_F^2}{S_1} + \frac{(1-p_{\text{out}})\mathcal{E}_{\text{var,in}}^1}{T}$$
$$\le 24\left(\frac{p_{\text{out}}}{B_1} + \frac{1-p_{\text{out}}}{B_2}\right)\frac{(1-p_{\text{in}})L_F^2}{p_{\text{in}}S_2}\frac{1}{T}\sum_{t=0}^{T-1}\Xi^t$$
$$+ \frac{8(1-p_{\text{out}})L_F^2 \gamma^2}{B_2}\frac{1}{T}\sum_{t=0}^{T-1}\mathbb{E}[\|G^{t+1}\|_2^2]$$
$$+ 24\left(\frac{p_{\text{out}}}{B_1} + \frac{(1-p_{\text{in}})(1-p_{\text{out}})}{B_2}\right)\frac{\tilde{L}_F^2}{S_1} + \frac{(1-p_{\text{out}})}{T}\mathcal{E}_{\text{var,in}}^1.$$

From Lemma 10, we plug in the upper bound of $\frac{1}{T}\sum_{t=0}^{T-1}\Xi^t$

$$\frac{1}{T}\sum_{t=0}^{T-1}\mathcal{E}_{\text{var,in}}^{t+1} \le 24\left(\frac{p_{\text{out}}}{B_1} + \frac{1-p_{\text{out}}}{B_2}\right)\frac{(1-p_{\text{in}})L_F^2}{p_{\text{in}}S_2}\left(\frac{6n^2}{B_2^2}\gamma^2 \frac{1}{T}\sum_{t=0}^{T-1}\mathbb{E}[\|G^{t+1}\|_2^2]\right)$$
$$+ \frac{8(1-p_{\text{out}})L_F^2 \gamma^2}{B_2}\frac{1}{T}\sum_{t=0}^{T-1}\mathbb{E}[\|G^{t+1}\|_2^2]$$
$$+ 24\left(\frac{p_{\text{out}}}{B_1} + \frac{(1-p_{\text{in}})(1-p_{\text{out}})}{B_2}\right)\frac{\tilde{L}_F^2}{S_1} + \frac{(1-p_{\text{out}})}{T}\mathcal{E}_{\text{var,in}}^1$$
$$\le 8\left(\left(\frac{p_{\text{out}}}{B_1} + \frac{1-p_{\text{out}}}{B_2}\right)\frac{(1-p_{\text{in}})}{p_{\text{in}}S_2}\frac{18n^2}{B_2^2} + \frac{(1-p_{\text{out}})}{B_2}\right)\frac{\gamma^2 L_F^2}{T}\sum_{t=0}^{T-1}\mathbb{E}[\|G^{t+1}\|_2^2]$$
$$+ 24\left(\frac{p_{\text{out}}}{B_1} + \frac{(1-p_{\text{in}})(1-p_{\text{out}})}{B_2}\right)\frac{\tilde{L}_F^2}{S_1} + \frac{(1-p_{\text{out}})}{T}\mathcal{E}_{\text{var,in}}^1.$$

Finally, we add the upper bound on with $\mathcal{E}_{\text{var,in}}^1$ with (29)

$$\frac{1}{T}\sum_{t=0}^{T-1}\mathcal{E}_{\text{var,in}}^{t+1} \le 8\left(\left(\frac{p_{\text{out}}}{B_1} + \frac{1-p_{\text{out}}}{B_2}\right)\frac{(1-p_{\text{in}})}{p_{\text{in}}S_2}\frac{18n^2}{B_2^2} + \frac{(1-p_{\text{out}})}{B_2}\right)\frac{\gamma^2 L_F^2}{T}\sum_{t=0}^{T-1}\mathbb{E}[\|G^{t+1}\|_2^2]$$
$$+ 24\left(\frac{p_{\text{out}}}{B_1} + \frac{(1-p_{\text{in}})(1-p_{\text{out}})}{B_2}\right)\frac{\tilde{L}_F^2}{S_1} + \frac{(1-p_{\text{out}})}{T}\frac{4\tilde{L}_F^2}{B_1 S_1}. \tag{30}$$

**Outer Variance.** Now we consider the outer variance for $t \geq 1$

$$\mathcal{E}_{\text{var,out}}^{t+1} \leq \frac{(1-p_{\text{out}})^2}{B_2} \mathbb{E}_i \left[ \left\| \mathbb{E}_{\eta|i}[G_i^{t+1}] - \mathbb{E}_{\eta|i}[\tilde{G}_i^t] \right\|_2^2 \right]$$

$$\leq \frac{(1-p_{\text{out}})^2}{B_2} \mathbb{E}_i \left[ \mathbb{E}_{\eta|i} \left[ \left\| G_i^{t+1} - \tilde{G}_i^t \right\|_2^2 \right] \right].$$

Compared to (27) we know that the upper bound of is smaller than that of $\mathcal{E}_{\text{var,in}}^{t+1}$. Besides, whereas $\mathcal{E}_{\text{var,out}}^1 = 0$ as we use large batch at $t=0$. Therefore, the upper bound of $\mathcal{E}_{\text{var}}^{t+1}$ is upper bounded by $2*(30)$.

**Variance of $\nabla g_{\tilde{\eta}}$.** From Lemma 9, we know that

$$\frac{1}{T}\sum_{t=0}^{T-1} \mathbb{E}[\mathcal{E}_{\nabla g}^{t+1}] \leq \frac{1}{B_1} \frac{C_f^2 \sigma_g^2}{m} + \frac{4(1-p_{\text{out}})}{B_2 m p_{\text{out}}} \frac{1}{T}\sum_{t=0}^{T-1}\left( \mathbb{E}\left[ \left\| G_i^{t+1} - \tilde{G}_i^t \right\|_2^2 \right]\right)$$

$$\leq \frac{1}{B_1} \frac{C_f^2 \sigma_g^2}{m} + \frac{1}{m}\mathcal{E}_{\text{var}}^{t+1}$$

Finally, we use $\mathbb{E}[\|G^{t+1}\|_2^2] = \|\mathbb{E}[G^{t+1}]\|_2^2 + \mathcal{E}_{\text{var}}^{t+1}$.

$$\frac{1}{T}\sum_{t=0}^{T-1} \mathcal{E}_{\text{var}}^{t+1} \leq 16\left(\left(\frac{p_{\text{out}}}{B_1} + \frac{1-p_{\text{out}}}{B_2}\right)\frac{(1-p_{\text{in}})}{p_{\text{in}}S_2}\frac{18n^2}{B_2^2} + \frac{(1-p_{\text{out}})}{B_2}\right)\frac{\gamma^2 L_F^2}{T}\sum_{t=0}^{T-1}\left\|\mathbb{E}[G^{t+1}]\right\|_2^2$$

$$+ 16\left(\left(\frac{p_{\text{out}}}{B_1} + \frac{1-p_{\text{out}}}{B_2}\right)\frac{(1-p_{\text{in}})}{p_{\text{in}}S_2}\frac{18n^2}{B_2^2} + \frac{(1-p_{\text{out}})}{B_2}\right)\frac{\gamma^2 L_F^2}{T}\sum_{t=0}^{T-1}\mathcal{E}_{\text{var}}^{t+1}$$

$$+ 48\left(\frac{p_{\text{out}}}{B_1} + \frac{(1-p_{\text{in}})(1-p_{\text{out}})}{B_2}\right)\frac{\tilde{L}_F^2}{S_1} + \frac{(1-p_{\text{out}})}{T}\frac{8\tilde{L}_F^2}{B_1 S_1}.$$

By taking step size $\gamma$ to satisfy

$$\gamma^2 L_F^2 \max\left\{\frac{p_{\text{out}}}{B_1}\frac{(1-p_{\text{in}})}{p_{\text{in}}S_2}\frac{18n^2}{B_2^2}, \frac{1-p_{\text{out}}}{B_2}\frac{(1-p_{\text{in}})}{p_{\text{in}}S_2}\frac{18n^2}{B_2^2}, \frac{(1-p_{\text{out}})}{B_2}\right\} \leq \frac{1}{16} \cdot \frac{1}{6}$$

which can be simplified to

$$\gamma^2 L_F^2 \max\left\{\frac{(1-p_{\text{in}})}{p_{\text{in}}S_2}\frac{18}{B_2}, \frac{1-p_{\text{out}}}{B_2}\frac{(1-p_{\text{in}})}{p_{\text{in}}S_2}\frac{18n^2}{B_2^2}, \frac{(1-p_{\text{out}})}{B_2}\right\} \leq \frac{1}{16} \cdot \frac{1}{6}.$$

Then the coefficient of $\frac{1}{T}\sum_{t=0}^{T-1}\mathcal{E}_{\text{var}}^{t+1}$ is bounded by $\frac{1}{2}$

$$16\left(\left(\frac{p_{\text{out}}}{B_1} + \frac{1-p_{\text{out}}}{B_2}\right)\frac{(1-p_{\text{in}})}{p_{\text{in}}S_2}\frac{18n^2}{B_2^2} + \frac{(1-p_{\text{out}})}{B_2}\right)\gamma^2 L_F^2 \leq \frac{1}{2}.$$

The the variance has the following bound

$$\frac{1}{T}\sum_{t=0}^{T-1} \mathcal{E}_{\text{var}}^{t+1} \leq 32\left(\left(\frac{p_{\text{out}}}{B_1} + \frac{1-p_{\text{out}}}{B_2}\right)\frac{(1-p_{\text{in}})}{p_{\text{in}}S_2}\frac{18n^2}{B_2^2} + \frac{(1-p_{\text{out}})}{B_2}\right)\frac{\gamma^2 L_F^2}{T}\sum_{t=0}^{T-1}\left\|\mathbb{E}[G^{t+1}]\right\|_2^2$$

$$+ 96\left(\frac{p_{\text{out}}}{B_1} + \frac{(1-p_{\text{in}})(1-p_{\text{out}})}{B_2}\right)\frac{\tilde{L}_F^2}{S_1} + \frac{(1-p_{\text{out}})}{T}\frac{8\tilde{L}_F^2}{B_1 S_1}.$$

$\square$

**Theorem 10** *Consider the* (FCCO) *problem. Suppose Assumptions 3, 4, 5 holds true. Let step size $\gamma = \mathcal{O}(\frac{1}{\sqrt{n}L_F})$. Then for NestedVR, $x^s$ picked uniformly at random among $\{x^t\}_{t=0}^{T-1}$ satisfies: $\mathbb{E}[\|\nabla F(x^s)\|_2^2] \leq \varepsilon^2$, for nonconvex $F$, if the hyperparameters of the inner loop $S_1 = \mathcal{O}(\tilde{L}_F^2 \varepsilon^{-2}), S_2 = \mathcal{O}(\tilde{L}_F \varepsilon^{-1}), p_{in} = \mathcal{O}(1/S_2)$, the hyperparameters of the outer loop $B_1 = n, B_2 = \sqrt{n}, p_{out} = 1/B_2$, and the number of iterations*

$$T = \Omega\left(\frac{\sqrt{n}L_F(F(x^0)-F^\star)}{\varepsilon^2}\right).$$

*The resulting sample complexity is*

$$\mathcal{O}\left(\frac{nL_F\tilde{L}_F(F(x^0)-F^\star)}{\varepsilon^3}\right).$$

*In fact, it reaches this sample complexity for all $\frac{p_{in}p_{out}}{\sqrt{1-p_{in}}} \lesssim \varepsilon$.*

**Proof:** Using descent lemma (Lemma 4) and bias-variance bounds of NestedVR (Lemma 13)

$$\frac{1}{T}\sum_{t=0}^{T-1}\|\nabla F(\boldsymbol{x}^t)\|_2^2 + \frac{1}{2T}\sum_{t=0}^{T-1}\left\|\mathbb{E}[G^{t+1}]\right\|_2^2$$

$$\leq \frac{2(F(\boldsymbol{x}^0)-F^\star)}{\gamma T} + \frac{L_F\gamma}{T}\sum_{t=0}^{T-1}\mathcal{E}_{\text{var}}^{t+1} + \frac{1}{T}\sum_{t=0}^{T-1}\mathcal{E}_{\text{bias}}^{t+1}$$

$$\leq \underbrace{\frac{2(F(\boldsymbol{x}^0)-F^\star)}{\gamma T}}_{\mathcal{T}_0} + \underbrace{\frac{4\tilde{L}_F^2}{S_1}}_{\mathcal{T}_1} + \underbrace{\frac{12(1-p_{\text{in}})}{p_{\text{in}}S_2}\frac{n^2}{B_2^2}\frac{L_F^2\gamma^2}{T}\sum_{t=0}^{T-1}\left\|\mathbb{E}[G^{t+1}]\right\|_2^2}_{\mathcal{T}_2}$$

$$+ \underbrace{\left(\frac{12(1-p_{\text{in}})}{p_{\text{in}}S_2}\frac{n^2}{B_2^2}L_F^2\gamma^2 + \frac{2(1-p_{\text{out}})^2}{p_{\text{out}}} + \gamma L_F\right)\frac{1}{T}\sum_{t=0}^{T-1}\mathcal{E}_{\text{var}}^{t+1}}_{\mathcal{T}_3}.$$

**Compute $\mathcal{T}_0$.** In order to let $\mathcal{T}_0 \leq \varepsilon^2$, we require that

$$\gamma T \geq \varepsilon^{-2}. \tag{31}$$

**Compute $\mathcal{T}_1$.** In order to let $\mathcal{T}_1$ to be smaller than $\varepsilon^2$, we need

$$S_1 = \frac{4\tilde{L}_F^2}{\varepsilon^2}.$$

**Compute $\mathcal{T}_2$.** In order to let the coefficient of $\frac{1}{T}\sum_{t=0}^{T-1}\left\|\mathbb{E}[G^{t+1}]\right\|_2^2$ in $\mathcal{T}_2$ to be less than $\frac{1}{4}$, i.e.

$$\frac{12(1-p_{\text{in}})}{p_{\text{in}}S_2}\frac{n^2}{B_2^2}L_F^2\gamma^2 \leq \frac{1}{4}, \tag{32}$$

which requires $\gamma$

$$\gamma \leq \frac{B_2\sqrt{p_{\text{in}}S_2}}{7L_F n\sqrt{1-p_{\text{in}}}} = \frac{p_{\text{out}}p_{\text{in}}\tilde{L}_F}{7\varepsilon L_F\sqrt{1-p_{\text{in}}}}. \tag{33}$$

**Compute $\mathcal{T}_3$.** Let us now focus on $\mathcal{T}_3$ and notice that the middle term $\frac{2(1-p_{\text{out}})^2}{p_{\text{out}}}$

$$\frac{2(1-p_{\text{out}})^2}{p_{\text{out}}}\frac{1}{T}\sum_{t=0}^{T-1}\mathcal{E}_{\text{var}}^{t+1}.$$

Using Lemma 13 we have that

$$\frac{2(1-p_{\text{out}})^2}{p_{\text{out}}}\frac{1}{T}\sum_{t=0}^{T-1}\mathcal{E}_{\text{var}}^{t+1}$$

$$\leq \underbrace{32\frac{2(1-p_{\text{out}})^2}{p_{\text{out}}}\left(\left(\frac{p_{\text{out}}}{B_1}+\frac{1-p_{\text{out}}}{B_2}\right)\frac{(1-p_{\text{in}})}{p_{\text{in}}S_2}\frac{18n^2}{B_2^2}+\frac{(1-p_{\text{out}})}{B_2}\right)\gamma^2 L_F^2\frac{1}{T}\sum_{t=0}^{T-1}\left\|\mathbb{E}[G^{t+1}]\right\|_2^2}_{\mathcal{T}_{3,1}}$$

$$+ \underbrace{96\frac{2(1-p_{\text{out}})^2}{p_{\text{out}}}\left(\frac{p_{\text{out}}}{B_1}+\frac{(1-p_{\text{in}})(1-p_{\text{out}})}{B_2}\right)\frac{\tilde{L}_F^2}{S_1}}_{\mathcal{T}_{3,2}} + \underbrace{\frac{2(1-p_{\text{out}})^2}{p_{\text{out}}}\frac{(1-p_{\text{out}})}{T}\frac{8\tilde{L}_F^2}{B_1 S_1}}_{\mathcal{T}_{3,3}}.$$

- Compute $\mathcal{T}_{3,3}$: As we already know that $S_1 = \mathcal{O}(\varepsilon^{-2})$ and $T \geq 1$ and $B_1 p_{\text{out}} \geq 1$. This imposes no more constraints, i.e.

$$S_1 = \mathcal{O}\left(\frac{\tilde{L}_F^2}{\varepsilon^2}\right).$$

- Compute $\mathcal{T}_{3,2}$: As $S_1 = \mathcal{O}(\varepsilon^{-2})$ and $B_1 = n$ and $B_2 = B_1 p_{\text{out}}$, then it requires

$$\frac{(1-p_{\text{in}})(1-p_{\text{out}})^3}{p_{\text{out}}^2} \leq n.$$

- Compute $\mathcal{T}_{3,1}$: In order to satisfy the following

$$32\frac{2(1-p_{\text{out}})^2}{p_{\text{out}}}\left(\left(\frac{p_{\text{out}}}{B_1}+\frac{1-p_{\text{out}}}{B_2}\right)\frac{(1-p_{\text{in}})}{p_{\text{in}}S_2}\frac{18n^2}{B_2^2}+\frac{(1-p_{\text{out}})}{B_2}\right)\gamma^2 L_F^2 \leq \frac{1}{12}$$

we need to enforce

$$\gamma \leq \frac{p_{\text{in}}p_{\text{out}}\tilde{L}_F}{\varepsilon L_F(1-p_{\text{in}})^{1/2}(1-p_{\text{out}})^{3/2}}. \tag{34}$$

Now we go back to $\mathcal{T}_3$ and compare the other two coefficients

$$\frac{12(1-p_{\text{in}})}{p_{\text{in}}S_2}\frac{n^2}{B_2^2}L_F^2\gamma^2 + \frac{2(1-p_{\text{out}})^2}{p_{\text{out}}} + \gamma L_F.$$

As $\gamma L_F \leq \frac{1}{2} \lesssim \frac{2(1-p_{\text{out}})^2}{p_{\text{out}}}$ we can safely ignore $\gamma L_F$. On the other hand, from (32) we know that the first term is also have

$$\frac{12(1-p_{\text{in}})}{p_{\text{in}}S_2}\frac{n^2}{B_2^2}L_F^2\gamma^2 \leq \frac{1}{4} \lesssim \frac{2(1-p_{\text{out}})^2}{p_{\text{out}}}.$$

**Constraints from the Bias-Variance Lemma (Lemma 13).** By setting $B_1 = n$ and $S_1 = \mathcal{O}(\frac{\tilde{L}_F^2}{\varepsilon^2})$, this constraint translates to

$$\gamma^2 L_F^2 \max\left\{\frac{(1-p_{\text{in}})}{p_{\text{in}}^2}\frac{\varepsilon^2}{B_2}, \frac{1-p_{\text{out}}}{B_2}\frac{(1-p_{\text{in}})\varepsilon^2}{p_{\text{in}}^2}\frac{1}{p_{\text{out}}^2}, \frac{(1-p_{\text{out}})}{B_2}\right\} \lesssim 1$$

which is weaker than (33).

**Summary on the Limit on $\gamma$.** Combine (33) and (34) and $\gamma \leq \frac{1}{2L_F}$, we have a final limit on step size $\gamma$

$$\gamma \lesssim \min\left\{\frac{p_{\text{out}}p_{\text{in}}\tilde{L}_F}{\varepsilon L_F\sqrt{1-p_{\text{in}}}}, \frac{1}{L_F}\right\} \tag{35}$$

Then the total sample complexity of NestedVR can be computed as

(# of iters $T$) × (Avg. outer batch size $B_2 = B_1 p_{\text{out}}$) × (Avg. inner batch size $S_2 = S_1 p_{\text{in}}$).

This sample complexity has the following requirement

$$B_2 S_2 T = \frac{B_2 S_2 (T\gamma)}{\gamma} \overset{(31)}{\geq} \frac{B_2 S_2}{\varepsilon^2 \gamma} = \frac{n\varepsilon^{-2}}{\varepsilon^2}\frac{p_{\text{in}}p_{\text{out}}}{\gamma} \overset{(35)}{\gtrsim} n\varepsilon^{-3}.$$

The lower bound $n\varepsilon^{-3}$ is reached when in (35) we have

$$\frac{p_{\text{out}}p_{\text{in}}\tilde{L}_F}{\varepsilon L_F\sqrt{1-p_{\text{in}}}} \lesssim \frac{1}{L_F}.$$

That is, $\frac{p_{\text{out}}p_{\text{in}}}{\sqrt{1-p_{\text{in}}}} \lesssim \varepsilon$.

In particular, we can choose the following hyperparameters to reach $\mathcal{O}(n\varepsilon^{-3})$ sample complexity

$$B_1 = n, \quad B_2 = \sqrt{n}, \quad p_{\text{out}} = \frac{1}{\sqrt{n}}, \quad S_1 = \mathcal{O}(\tilde{L}_F^2\varepsilon^{-2}), \quad S_2 = \mathcal{O}(\tilde{L}_F\varepsilon^{-1}), \quad p_{\text{in}} = \mathcal{O}(\tilde{L}_F^{-1}\varepsilon)$$

The step size $\gamma$ can be chosen as

$$\gamma \lesssim \frac{1}{\sqrt{n}L_F}.$$

and the iteration complexity

$$T = \Omega\left(\frac{\sqrt{n}L_F(F(\boldsymbol{x}^0) - F^\star)}{\varepsilon^2}\right).$$

Putting these together gives the claimed sample complexity bound. By picking $\boldsymbol{x}^s$ uniformly at random among $\{\boldsymbol{x}^t\}_{t=0}^{T-1}$, we get the desired guarantee. $\qquad\square$

### E.3 Convergence of E-NestedVR

In this section, we analyze the sample complexity of Algorithm 1 (E-NestedVR) for the FCCO problem with

$$G_{\text{E-NVR}}^{t+1} = \begin{cases} \frac{1}{B_1}\sum_i(\boldsymbol{z}_i^{t+1})^\top \mathcal{L}_{\mathcal{D}_{\boldsymbol{y},i}^{t+1}}^{(2)}\nabla f_i(0) & \text{with prob. } p_{\text{out}} \\ G_{\text{E-NVR}}^t + \frac{1}{B_2}\sum_i\left((\boldsymbol{z}_i^{t+1})^\top \mathcal{L}_{\mathcal{D}_{\boldsymbol{y},i}^{t+1}}^{(2)}\nabla f_i(0) - (\boldsymbol{z}_i^t)^\top \mathcal{L}_{\mathcal{D}_{\boldsymbol{y},i}^t}^{(2)}\nabla f_i(0)\right) & \text{with prob. } 1 - p_{\text{out}}. \end{cases} \tag{36}$$

**Lemma 14 (Bias and Variance of E-NestedVR)** *If the step size $\gamma$ satisfies*

$$\gamma^2 L_F^2 \max\left\{ \frac{(1-p_{in})}{p_{in}S_2}\frac{18}{B_2}, \frac{1-p_{out}}{B_2}\frac{(1-p_{in})}{p_{in}S_2}\frac{18n^2}{B_2^2}, \frac{(1-p_{out})}{B_2} \right\} \le \frac{1}{16}\cdot\frac{1}{6}$$

*then the variance and bias of E-NestedVR are*

$$\frac{1}{T}\sum_{t=0}^{T-1}\mathcal{E}_{var}^{t+1} \le 14\cdot 32\left(\left(\frac{p_{out}}{B_1}+\frac{1-p_{out}}{B_2}\right)\frac{(1-p_{in})}{p_{in}S_2}\frac{18n^2}{B_2^2}+\frac{(1-p_{out})}{B_2}\right)\frac{\gamma^2 L_F^2}{T}\sum_{t=0}^{T-1}\left\|\mathbb{E}[G^{t+1}]\right\|_2^2$$

$$+ 14\cdot 96\left(\frac{p_{out}}{B_1}+\frac{(1-p_{in})(1-p_{out})}{B_2}\right)\frac{\tilde{L}_F^2}{S_1}+\frac{(1-p_{out})}{T}\frac{8\tilde{L}_F^2}{B_1 S_1}.$$

$$\frac{1}{T}\sum_{t=0}^{T-1}\mathcal{E}_{bias}^{t+1} \le \frac{(1-p_{in})^3\tilde{L}_F^2}{p_{in}S_2}\frac{6n^2}{B_2^2}\gamma^2\frac{1}{T}\sum_{t=0}^{T-1}\left\|\mathbb{E}[G^{t+1}]\right\|_2^2 + \frac{2(1-p_{in})^2\tilde{L}_F^2}{S_1}+\frac{C_e^2}{S_2^4}$$

$$+ \left(\frac{(1-p_{in})^3\tilde{L}_F^2}{p_{in}S_2}\frac{6n^2}{B_2^2}\gamma^2 + \frac{(1-p_{out})^2}{p_{out}}\right)\frac{1}{T}\sum_{t=0}^{T-1}\mathcal{E}_{var}^{t+1}.$$

**Proof:** Note that this proof is very similar to NestedVR so we highlight the differences. Let $G^{t+1}=G_{\text{E-NVR}}^{t+1}$ (36) be the E-NestedVR update and define

$$G_i^{t+1} := (z_i^{t+1})^\top \mathcal{L}_{\mathcal{D}_{y,i}^{t+1}}^{(2)}\nabla f_i(0)$$

We expand the bias by inserting $\mathbb{E}_{i,p_{in},\eta|i}[G_i^{t+1}]$

$$\mathcal{E}_{\text{bias}}^{t+1} = \left\|\nabla F(x^{t+1})-\mathbb{E}[G^{t+1}]\right\|_2^2$$

$$\le 2\underbrace{\left\|\nabla F(x^{t+1})-\mathbb{E}_{i,p_{in},\eta,\tilde{\eta}|i}[G_i^{t+1}]\right\|_2^2}_{A_1^{t+1}}+2\underbrace{\left\|\mathbb{E}_{i,p_{in},\eta,\tilde{\eta}|i}[G_i^{t+1}]-\mathbb{E}[G^{t+1}]\right\|_2^2}_{A_2^{t+1}}.$$

**Consider $A_1^{t+1}$.** The term $A_1^{t+1}$ captures the difference between full gradient and extrapolated gradient

$$A_1^{t+1} = \left\|\mathbb{E}_i\left[(\mathbb{E}_{\tilde{\eta}|i}[\nabla g_{\tilde{\eta}}(x^t)])^\top \nabla f_i(\mathbb{E}[g_\eta(x^t)])-\mathbb{E}_{p_{in},\eta|i}\left[(\mathbb{E}_{\tilde{\eta}|i}[\nabla g_{\tilde{\eta}}(x^t)])^\top \mathcal{L}_{\mathcal{D}_{y,i}^{t+1}}^{(2)}\nabla f_i(0)\right]\right]\right\|_2^2$$

$$\le C_g^2\,\mathbb{E}_i\left[\left\|\nabla f_i(\mathbb{E}_{\eta|i}[g_\eta(x^t)])-\mathbb{E}_{p_{in},\eta|i}\left[\mathcal{L}_{\mathcal{D}_{y,i}^{t+1}}^{(2)}\nabla f_i(0)\right]\right\|_2^2\right]$$

$$\le 2C_g^2\,\mathbb{E}_i\underbrace{\left[\left\|\nabla f_i(\mathbb{E}_{\eta|i}[g_\eta(x^t)])-\mathbb{E}_{p_{in}}[\nabla f_i(\mathbb{E}_{\eta|i}[y_i^{t+1}])]\right\|_2^2\right]}_{=:A_{1,1}^{t+1}}$$

$$+ 2C_g^2\,\mathbb{E}_i\underbrace{\left[\left\|\mathbb{E}_{p_{in}}[\nabla f_i(\mathbb{E}_{\eta|i}[y_i^{t+1}])]-\mathbb{E}_{p_{in},\eta|i}\left[\mathcal{L}_{\mathcal{D}_{y,i}^{t+1}}^{(2)}\nabla f_i(0)\right]\right\|_2^2\right]}_{=:A_{1,2}^{t+1}}.$$

The first term $A_{1,1}^{t+1}$ can be upper bounded through smoothness of $f_\xi$, for $t\ge 1$

$$A_{1,1}^{t+1} = \mathbb{E}_i\left[\left\|\nabla f_i(\mathbb{E}_{\eta|i}[g_\eta(x^t)])-p_{in}\nabla f_i(\mathbb{E}_{\eta|i}[g_\eta(x^t)])-(1-p_{in})\nabla f_i(y_i^t+\mathbb{E}_{\eta|i}[g_\eta(x^t)-g_\eta(\phi_i^t)])\right\|_2^2\right]$$

$$= (1-p_{in})^2\,\mathbb{E}_i\left[\left\|\nabla f_i(\mathbb{E}[g_\eta(x^t)])-\nabla f_i(y_i^t+\mathbb{E}_{\eta|i}[g_\eta(x^t)-g_\eta(\phi_i^t)])\right\|_2^2\right]$$

$$\le (1-p_{in})^2 L_f^2\,\mathbb{E}_i\left[\left\|\mathbb{E}_{\eta|i}[g_\eta(x^t)]-(y_i^t+\mathbb{E}_{\eta|i}[g_\eta(x^t)-g_\eta(\phi_i^t)])\right\|_2^2\right]$$

$$= (1-p_{in})^2 L_f^2\,\mathbb{E}_i[\left\|y_i^t-\mathbb{E}_{\eta|i}[g_\eta(\phi_i^t)]\right\|_2^2]$$

$$= (1-p_{in})^2 L_f^2\mathcal{E}_y^t.$$

For $t=0$, $A_{1,1}^1=0$, then

$$\frac{1}{T}\sum_{t=0}^{T-1}A_{1,1}^{t+1} \le (1-p_{in})^2 L_f^2 C_g^2\frac{1}{T}\sum_{t=0}^{T-1}\mathcal{E}_y^{t+1}. \tag{37}$$

On the other hand, with Lemma 6

$$A_{1,2}^{t+1} \leq p_{\text{in}} \, \mathbb{E}_i \left[ \left\| \nabla f_i(\mathbb{E}_{\eta|i}[g_\eta(\boldsymbol{x}^t)]) - \mathbb{E}_{\eta|i} \left[ \mathcal{L}_{\mathcal{D}_{\boldsymbol{y},S_1,i}^{t+1}}^{(2)} \nabla f_i(0) \right] \right\|_2^2 \right]$$

$$+ (1 - p_{\text{in}}) \, \mathbb{E}_i \left[ \left\| \nabla f_i(\boldsymbol{y}_i^t + \mathbb{E}_{\eta|i}[g_\eta(\boldsymbol{x}^t) - g_\eta(\boldsymbol{\phi}_i^t)]) - \mathbb{E}_{\eta|i} \left[ \mathcal{L}_{\mathcal{D}_{\boldsymbol{y},S_2,i}^{t+1}}^{(2)} \nabla f_i(0) \right] \right\|_2^2 \right]$$

$$\leq \frac{p_{\text{in}} C_e^2}{S_1^4} + \frac{(1 - p_{\text{in}}) C_e^2}{S_2^4}$$

$$\leq \frac{C_e^2}{S_2^4}$$

where $\mathcal{D}_{\boldsymbol{y},S_1,i}^{t+1}$ is the distribution of $\frac{1}{S_1} \sum_{\eta \in \mathcal{S}_1} g_\eta(\boldsymbol{x}^t)$ and $\mathcal{D}_{\boldsymbol{y},S_2,i}^{t+1}$ is the distribution of

$$\boldsymbol{y}_i^t + \frac{1}{S_2} \sum_{\eta \in \mathcal{S}_2} (g_\eta(\boldsymbol{x}^t) - \mathbb{E}[g_\eta(\boldsymbol{\phi}_i^t)]).$$

Thus the $A_1^{t+1}$ has the following upper bound

$$\frac{1}{T} \sum_{t=0}^{T-1} A_1^{t+1} \leq (1 - p_{\text{in}})^2 L_f^2 C_g^2 \frac{1}{T} \sum_{t=0}^{T-1} \mathcal{E}_y^{t+1} + \frac{C_e^2}{S_2^4}. \tag{38}$$

**Consider $A_2^{t+1}$.** Let us expand $A_2^{t+1}$ through recursion

$$A_2^{t+1} = \left\| \mathbb{E}_{i,p_{\text{in}},\eta,\tilde{\eta}|i}[G_i^{t+1}] - \mathbb{E}[G^{t+1}] \right\|_2^2$$

$$= (1 - p_{\text{out}})^2 \left\| G^t - \mathbb{E}_i[\mathbb{E}_{\eta,\tilde{\eta}|i}[\tilde{G}_i^t]] \right\|_2^2$$

$$= (1 - p_{\text{out}})^2 \left( \left\| \mathbb{E}[G^t] - \mathbb{E}_i[\mathbb{E}_{\eta,\tilde{\eta}|i}[\tilde{G}_i^t]] \right\|_2^2 + \mathcal{E}_{\text{var}}^t \right)$$

$$= (1 - p_{\text{out}})^2 \left( A_2^t + \mathcal{E}_{\text{var}}^t \right).$$

For $t = 0$, we have that $A_2^1 = 0$, then average over time gives

$$\frac{1}{T} \sum_{t=0}^{T-1} A_2^{t+1} \leq \frac{(1 - p_{\text{out}})^2}{p_{\text{out}}} \frac{1}{T} \sum_{t=0}^{T-1} \mathcal{E}_{\text{var}}^{t+1}.$$

Therefore, the bias has the following bound

$$\frac{1}{T} \sum_{t=0}^{T-1} \mathcal{E}_{\text{bias}}^{t+1} \leq (1 - p_{\text{in}})^2 L_f^2 C_g^2 \frac{1}{T} \sum_{t=0}^{T-1} \mathcal{E}_y^{t+1} + \frac{C_e^2}{S_2^4} + \frac{(1 - p_{\text{out}})^2}{p_{\text{out}}} \frac{1}{T} \sum_{t=0}^{T-1} \mathcal{E}_{\text{var}}^{t+1}.$$

Using Lemma 11

$$\frac{1}{T} \sum_{t=0}^{T-1} \mathcal{E}_{\text{bias}}^{t+1} \leq (1 - p_{\text{in}})^2 L_f^2 C_g^2 \left( \frac{(1 - p_{\text{in}}) C_g^2}{p_{\text{in}} S_2} \frac{1}{T} \sum_{t=0}^{T-1} \Xi^t + \frac{2\sigma_g^2}{S_1} \right)$$

$$+ \frac{C_e^2}{S_2^4} + \frac{(1 - p_{\text{out}})^2}{p_{\text{out}}} \frac{1}{T} \sum_{t=0}^{T-1} \mathcal{E}_{\text{var}}^{t+1}$$

$$\leq \frac{(1 - p_{\text{in}})^3 \tilde{L}_F^2}{p_{\text{in}} S_2} \frac{1}{T} \sum_{t=0}^{T-1} \Xi^t + \frac{2(1 - p_{\text{in}})^2 \tilde{L}_F^2}{S_1} + \frac{C_e^2}{S_2^4} + \frac{(1 - p_{\text{out}})^2}{p_{\text{out}}} \frac{1}{T} \sum_{t=0}^{T-1} \mathcal{E}_{\text{var}}^{t+1}.$$

Using Lemma 10 we have that

$$\frac{1}{T} \sum_{t=0}^{T-1} \mathcal{E}_{\text{bias}}^{t+1} \leq \frac{(1 - p_{\text{in}})^3 \tilde{L}_F^2}{p_{\text{in}} S_2} \left( \frac{6n^2}{B_2^2} \gamma^2 \frac{1}{T} \sum_{t=0}^{T-1} \mathbb{E}[\|G^{t+1}\|_2^2] \right) + \frac{2(1 - p_{\text{in}})^2 \tilde{L}_F^2}{S_1} + \frac{C_e^2}{S_2^4} + \frac{(1 - p_{\text{out}})^2}{p_{\text{out}}} \frac{1}{T} \sum_{t=0}^{T-1} \mathcal{E}_{\text{var}}^{t+1}$$

$$\leq \frac{(1 - p_{\text{in}})^3 \tilde{L}_F^2}{p_{\text{in}} S_2} \frac{6n^2}{B_2^2} \gamma^2 \frac{1}{T} \sum_{t=0}^{T-1} \|\mathbb{E}[G^{t+1}]\|_2^2 + \frac{2(1 - p_{\text{in}})^2 \tilde{L}_F^2}{S_1} + \frac{C_e^2}{S_2^4}$$

$$+ \left( \frac{(1 - p_{\text{in}})^3 \tilde{L}_F^2}{p_{\text{in}} S_2} \frac{6n^2}{B_2^2} \gamma^2 + \frac{(1 - p_{\text{out}})^2}{p_{\text{out}}} \right) \frac{1}{T} \sum_{t=0}^{T-1} \mathcal{E}_{\text{var}}^{t+1}.$$

**Variance.** Combine the variance of NestedVR in Lemma 13 and Lemma 2 gives

$$\frac{1}{T} \sum_{t=0}^{T-1} \mathcal{E}_{\text{var}}^{t+1} \leq 14 \cdot 32 \left( \left( \frac{p_{\text{out}}}{B_1} + \frac{1 - p_{\text{out}}}{B_2} \right) \frac{(1 - p_{\text{in}})}{p_{\text{in}} S_2} \frac{18n^2}{B_2^2} + \frac{(1 - p_{\text{out}})}{B_2} \right) \frac{\gamma^2 L_F^2}{T} \sum_{t=0}^{T-1} \|\mathbb{E}[G^{t+1}]\|_2^2$$

$$+ 14 \cdot 96 \left( \frac{p_{\text{out}}}{B_1} + \frac{(1 - p_{\text{in}})(1 - p_{\text{out}})}{B_2} \right) \frac{\tilde{L}_F^2}{S_1} + \frac{(1 - p_{\text{out}})}{T} \frac{8\tilde{L}_F^2}{B_1 S_1}.$$

$\square$

**Theorem 5** *[E-NestedVR Convergence] Consider the* (FCCO) *problem. Under the same assumptions as Theorem 3.*

- *If $n = \mathcal{O}(\varepsilon^{-2/3})$, then we choose the hyperaparameters of E-NestedVR (Algorithm 1) as $B_1 = B_2 = n, p_{out} = 1, S_1 = \tilde{L}_F^2 \varepsilon^{-2}, S_2 = \tilde{L}_F \varepsilon^{-1}, p_{in} = \tilde{L}_F^{-1}\varepsilon, \gamma = \mathcal{O}(\frac{1}{L_F})$.*

- *If $n = \Omega(\varepsilon^{-2/3})$, then we choose the hyperaparameters of E-NestedVR as $B_1 = n, B_2 = \sqrt{n}, p_{out} = 1/\sqrt{n}, S_1 = S_2 = \max\left\{C_e C_g \varepsilon^{-1/2}, \tilde{L}_F^2/(n\varepsilon^2)\right\}, p_{in} = 1, \gamma = \mathcal{O}(\frac{1}{L_F})$.*

*Then the output $\boldsymbol{x}^s$ of E-NestedVR satisfies: $\mathbb{E}[\|\nabla F(\boldsymbol{x}^s)\|_2^2] \leq \varepsilon^2$, for nonconvex F with iterations*

$$T = \Omega\left(L_F(F(\boldsymbol{x}^0) - F^\star)\varepsilon^{-2}\right).$$

**Proof:** Denote. $G^{t+1} = G_{\text{E-NVR}}^{t+1}$ (36). Using descent lemma (Lemma 3) and bias-variance of E-NestedVR (Lemma 14)

$$\frac{1}{T}\sum_{t=0}^{T-1}\|\nabla F(\boldsymbol{x}^t)\|_2^2 + \frac{1}{2T}\sum_{t=0}^{T-1}\left\|\mathbb{E}[G^{t+1}]\right\|_2^2$$

$$\leq \frac{2(F(\boldsymbol{x}^0)-F^\star)}{\gamma T} + \frac{L_F\gamma}{T}\sum_{t=0}^{T-1}\mathcal{E}_{\text{var}}^{t+1} + \frac{1}{T}\sum_{t=0}^{T-1}\mathcal{E}_{\text{bias}}^{t+1}$$

$$\leq \frac{2(F(\boldsymbol{x}^0)-F^\star)}{\gamma T} + \frac{(1-p_{\text{in}})^3\tilde{L}_F^2}{p_{\text{in}}S_2}\frac{6n^2}{B_2^2}\gamma^2\frac{1}{T}\sum_{t=0}^{T-1}\left\|\mathbb{E}[G^{t+1}]\right\|_2^2 + \frac{2(1-p_{\text{in}})^2\tilde{L}_F^2}{S_1} + \frac{C_e^2}{S_2^4}$$

$$+ \left(\frac{(1-p_{\text{in}})^3\tilde{L}_F^2}{p_{\text{in}}S_2}\frac{6n^2}{B_2^2}\gamma^2 + \frac{(1-p_{\text{out}})^2}{p_{\text{out}}} + L_F\gamma\right)\frac{1}{T}\sum_{t=0}^{T-1}\mathcal{E}_{\text{var}}^{t+1}.$$

As we would like the right-hand side to be bounded by either $\frac{1}{T}\sum_{t=0}^{T-1}\left\|\mathbb{E}[G^{t+1}]\right\|_2^2$ or $\varepsilon^2$.

- **Bound on $\frac{2(F(\boldsymbol{x}^0)-F^\star)}{\gamma T}$ with $\varepsilon^2$ , i.e.**

$$\gamma T \gtrsim (F(\boldsymbol{x}^0) - F^\star)\varepsilon^{-2} \tag{39}$$

- **Coefficient of $\frac{1}{T}\sum_{t=0}^{T-1}\left\|\mathbb{E}[G^{t+1}]\right\|_2^2$ is bounded by $\frac{1}{4}$, i.e.**

$$\frac{(1-p_{\text{in}})^3\tilde{L}_F^2}{p_{\text{in}}S_2}\frac{6n^2}{B_2^2}\gamma^2 \leq \frac{1}{4}$$

which can be achieved by choosing the following step size

$$\gamma \leq \frac{p_{\text{out}}p_{\text{in}}\sqrt{S_1}}{5\tilde{L}_F(1-p_{\text{in}})^{3/2}}. \tag{40}$$

- **Bound on $\frac{2(1-p_{\text{in}})^2\tilde{L}_F^2}{S_1}$ with $\varepsilon^2$**

$$\frac{2(1-p_{\text{in}})^2\tilde{L}_F^2}{S_1} \leq \varepsilon^2. \tag{41}$$

- **Bound $\frac{C_e^2}{S_2^4}$ with $\varepsilon^2$.** This leads to

$$S_2 \geq \sqrt{\frac{C_e}{\varepsilon}}. \tag{42}$$

- **Bound on the variance.** First notice from (40) and $\gamma \leq \frac{1}{2L_F}$,

$$\frac{(1-p_{\text{in}})^3\tilde{L}_F^2}{p_{\text{in}}S_2}\frac{6n^2}{B_2^2}\gamma^2 \leq \frac{1}{4} \lesssim \frac{(1-p_{\text{out}})^2}{p_{\text{out}}}$$

$$L_F\gamma \leq \frac{1}{2} \lesssim \frac{(1-p_{\text{out}})^2}{p_{\text{out}}}.$$

Therefore, we only need to consider the upper bound on

$$\frac{(1-p_{\text{out}})^2}{p_{\text{out}}}\frac{1}{T}\sum_{t=0}^{T-1}\mathcal{E}_{\text{var}}^{t+1}$$

$$\leq 14\cdot 32\frac{(1-p_{\text{out}})^2}{p_{\text{out}}}\left(\left(\frac{p_{\text{out}}}{B_1} + \frac{1-p_{\text{out}}}{B_2}\right)\frac{(1-p_{\text{in}})}{p_{\text{in}}S_2}\frac{18n^2}{B_2^2} + \frac{(1-p_{\text{out}})}{B_2}\right)\frac{\gamma^2 L_F^2}{T}\sum_{t=0}^{T-1}\left\|\mathbb{E}[G^{t+1}]\right\|_2^2$$

$$+ 14\cdot 96\frac{(1-p_{\text{out}})^2}{p_{\text{out}}}\left(\frac{p_{\text{out}}}{B_1} + \frac{(1-p_{\text{in}})(1-p_{\text{out}})}{B_2}\right)\frac{\tilde{L}_F^2}{S_1} + \frac{(1-p_{\text{out}})^3}{p_{\text{out}}T}\frac{8\tilde{L}_F^2}{B_1 S_1}.$$

We impose the constraints for each term

$$\frac{(1-p_{\text{out}})^2}{p_{\text{out}}} \frac{p_{\text{out}}}{B_1} \frac{(1-p_{\text{in}})}{p_{\text{in}} S_2} \frac{18n^2}{B_2^2} L_F^2 \gamma^2 \lesssim 1$$

$$\frac{(1-p_{\text{out}})^2}{p_{\text{out}}} \frac{1-p_{\text{out}}}{B_2} \frac{(1-p_{\text{in}})}{p_{\text{in}} S_2} \frac{18n^2}{B_2^2} L_F^2 \gamma^2 \lesssim 1$$

$$\frac{(1-p_{\text{out}})^2}{p_{\text{out}}} \frac{1-p_{\text{out}}}{B_2} L_F^2 \gamma^2 \lesssim 1$$

$$\frac{(1-p_{\text{out}})^2}{p_{\text{out}}} \frac{p_{\text{out}}}{B_1} \frac{\tilde{L}_F^2}{S_1} \lesssim \varepsilon^2$$

$$\frac{(1-p_{\text{out}})^2}{p_{\text{out}}} \frac{(1-p_{\text{in}})(1-p_{\text{out}})}{B_2} \frac{\tilde{L}_F^2}{S_1} \lesssim \varepsilon^2$$

$$\frac{(1-p_{\text{out}})^3}{p_{\text{out}} T} \frac{8\tilde{L}_F^2}{B_1 S_1} \lesssim \varepsilon^2.$$

These can be simplified as

$$\gamma \lesssim \frac{p_{\text{in}} p_{\text{out}} \sqrt{B_1} \sqrt{S_1}}{(1-p_{\text{out}}) \sqrt{1-p_{\text{in}}}} \frac{1}{L_F} \tag{43}$$

$$\gamma \lesssim \frac{p_{\text{in}} p_{\text{out}}^2 \sqrt{B_1} \sqrt{S_1}}{(1-p_{\text{out}})^{3/2} \sqrt{1-p_{\text{in}}}} \frac{1}{L_F} \tag{44}$$

$$\gamma \lesssim \frac{\sqrt{B_1}}{(1-p_{\text{out}})^{3/2}} \frac{1}{L_F} \tag{45}$$

$$B_1 S_1 \gtrsim \frac{(1-p_{\text{out}})^2 \tilde{L}_F^2}{\varepsilon^2} \tag{46}$$

$$B_1 S_1 \gtrsim \frac{(1-p_{\text{out}})^3 (1-p_{\text{in}}) \tilde{L}_F^2}{\varepsilon^2 p_{\text{out}}^2} \tag{47}$$

$$B_1 S_1 \gtrsim \frac{(1-p_{\text{out}})^3 \tilde{L}_F^2}{T \varepsilon^2 p_{\text{out}}}. \tag{48}$$

- Constraints from Lemma 14

$$\gamma^2 L_F^2 \max\left\{ \frac{(1-p_{\text{in}})}{p_{\text{in}} S_2} \frac{18}{B_2}, \frac{1-p_{\text{out}}}{B_2} \frac{(1-p_{\text{in}})}{p_{\text{in}} S_2} \frac{18n^2}{B_2^2}, \frac{(1-p_{\text{out}})}{B_2} \right\} \leq \frac{1}{16} \cdot \frac{1}{6}$$

which can be translated to

$$\gamma \lesssim \frac{p_{\text{in}} \sqrt{S_1} \sqrt{B_2}}{L_F \sqrt{1-p_{\text{in}}}} \tag{49}$$

$$\gamma \lesssim \frac{p_{\text{in}} p_{\text{out}} \sqrt{S_1} \sqrt{B_2}}{L_F \sqrt{1-p_{\text{in}}} \sqrt{1-p_{\text{out}}}} \tag{50}$$

$$\gamma \lesssim \frac{\sqrt{B_2}}{L_F \sqrt{1-p_{\text{out}}}} \tag{51}$$

- Constraint from sufficient decrease lemma:

$$\gamma \leq \frac{1}{2L_F}. \tag{52}$$

We simplify the conditions noticing that 1) (48) is weaker than (46); 2) (45) and (51) are weaker than (52). Combine all the constraints on $\gamma$, i.e. (43), (44), (49), (50), (52)

$$\gamma \lesssim \frac{1}{L_F} \min\left\{ \min\left\{1, \frac{p_{\text{out}}}{\sqrt{1-p_{\text{out}}}}\right\} \frac{p_{\text{in}} p_{\text{out}} \sqrt{B_1} \sqrt{S_1}}{(1-p_{\text{out}}) \sqrt{1-p_{\text{in}}}} \frac{1}{L_F}, \min\left\{1, \frac{p_{\text{out}}}{\sqrt{1-p_{\text{out}}}}\right\} \frac{p_{\text{in}} \sqrt{S_1} \sqrt{B_2}}{\sqrt{1-p_{\text{in}}}}, 1, \frac{p_{\text{out}} p_{\text{in}} \sqrt{S_1}}{5\tilde{L}_F (1-p_{\text{in}})^{3/2}} \right\}.$$

This can be simplified as an upper bound

$$\gamma \lesssim \frac{1}{L_F} \min\left\{ \frac{p_{\text{in}} p_{\text{out}} \sqrt{S_1}}{\sqrt{1-p_{\text{in}}}}, \frac{p_{\text{in}} p_{\text{out}} \sqrt{S_1} \sqrt{B_1}}{\sqrt{1-p_{\text{out}}}}, \frac{p_{\text{in}} p_{\text{out}}^2 \sqrt{S_1} \sqrt{B_1}}{\sqrt{1-p_{\text{in}}} \sqrt{1-p_{\text{out}}}}, 1 \right\}.$$

Now we consider two sets of hyperparameters depending on the size of $n$ **Case 1:** For $n = \mathcal{O}(\varepsilon^{-2/3})$, we choose the following set of hyperparameters

$$B_1 = B_2 = n, \quad p_{\text{out}} = 1, \quad S_1 = \tilde{L}_F^2 \varepsilon^{-2}, \quad S_2 = \tilde{L}_F \varepsilon^{-1}, \quad p_{\text{in}} = \tilde{L}_F^{-1} \varepsilon.$$

Then we have $\gamma \lesssim \frac{1}{L_F} \min\{\frac{p_{\text{in}} \sqrt{S_1}}{\sqrt{1-p_{\text{in}}}}, 1\} = \frac{1}{L_F}$, we have the total sample complexity of

$$B_2 S_2 T = \frac{B_2 S_2 T \gamma}{\gamma} \overset{(39)}{=} \frac{F(\boldsymbol{x}^0) - F^\star}{\varepsilon^2} \frac{B_2 S_2}{\gamma} = \frac{(F(\boldsymbol{x}^0) - F^\star) n \tilde{L}_F L_F}{\varepsilon^3}$$

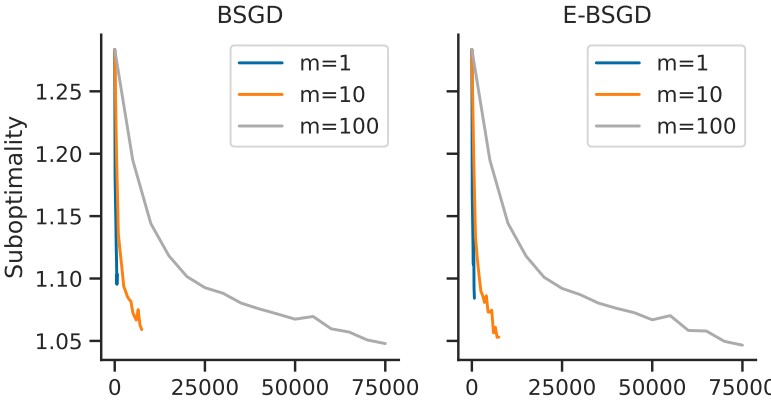

Figure 4: Performance of BSGD vs. E-BSGD on the few-shot sinsuoid regression task.

**Case 2:** For $n = \Omega(\varepsilon^{-2/3})$, we choose the following set of hyperparameters

$$B_1 = n, \quad B_2 = \sqrt{n}, \quad p_{\text{out}} = \frac{1}{\sqrt{n}}.$$

In this case, (46) is stronger than (47) which requires $S_1 \gtrsim \frac{\tilde{L}_F^2}{n\varepsilon^2}$

$$S_1 = S_2 = \max\left\{\tilde{\sigma}_{\text{bias}}^{1/2}\varepsilon^{-1/2}, \frac{\sigma_{\text{in}}^2}{n\varepsilon^2}\right\}, \quad p_{\text{in}} = 1$$

Then we have $\gamma \lesssim \frac{1}{L_F}\min\{\frac{p_{\text{out}}\sqrt{n}\sqrt{S_1}}{\sqrt{1-p_{\text{out}}}}, 1\} = \frac{1}{L_F}$, we have the total sample complexity of

$$B_2 S_2 T = \frac{B_2 S_2 T \gamma}{\gamma} \overset{(39)}{=} \frac{F(\boldsymbol{x}^0) - F^\star}{\varepsilon^2} \frac{B_2 S_2}{\gamma} = (F(\boldsymbol{x}^0) - F^\star)\max\left\{\frac{\sqrt{n}\tilde{\sigma}_{\text{bias}}^{1/2}}{\varepsilon^{2.5}}, \frac{\sigma_{\text{in}}^2}{\sqrt{n}\varepsilon^4}\right\}.$$

By picking $\boldsymbol{x}^s$ uniformly at random among $\{\boldsymbol{x}^t\}_{t=0}^{T-1}$, we get the desired guarantee. $\qquad\square$

# F  Missing Details from Section 5

## F.1  Application of First-order MAML

Over the past few years, the MAML framework [11] has become quite popular for few-shot supervised learning and meta reinforcement learning tasks. The first-order Model-Agnostic Meta-Learning (MAML) can be formulated mathematically as follows:

$$\min_{\boldsymbol{x}} \mathbb{E}_{i\sim p, \mathcal{D}_{\text{query}}^i} \ell_i\left(\mathbb{E}_{\mathcal{D}_{\text{supp}}^i}(\boldsymbol{x} - \alpha\nabla\ell_i(\boldsymbol{x}, \mathcal{D}_{\text{supp}}^i)), \mathcal{D}_{\text{query}}^i\right)$$

where $\alpha$ is the step size, $\mathcal{D}_{\text{supp}}^i$ and $\mathcal{D}_{\text{query}}^i$ are meta-training and meta-testing data respectively and $\ell_i$ being the loss function of task $i$. Stated in the CSO framework, $f_\xi(\boldsymbol{x}) := \ell_i(\boldsymbol{x}, \mathcal{D}_{\text{query}}^i)$ and $g_\eta(\boldsymbol{x}, \xi) := \boldsymbol{x} - \alpha\nabla\ell_i(\boldsymbol{x}, \mathcal{D}_{\text{supp}}^i)$ where $\xi = (i, \mathcal{D}_{\text{query}}^i)$ and $\eta = \mathcal{D}_{\text{supp}}^i$.

In this context, lots of popular choices for $f_\xi$ are smooth. For illustration purposes, we now discuss a widely used sine-wave few-shot regression task as appearing from the work of Finn et al. [11], where the goal is to do a few-shot learning of a sine wave, $A\sin(t - \phi)$, using a neural network $\Phi_{\boldsymbol{x}}(t)$ with smooth activations, where $A$ and $\phi$ represent the unknown amplitude and phase, and $\boldsymbol{x}$ denotes the model weight. Each task $i$ is characterized by $(A^i, \phi^i, \mathcal{D}_{\text{query}}^i)$. In the first-order MAML training process, we randomly select a task $i$, and draw training data $\eta = \mathcal{D}_{\text{supp}}^i$. Define the loss function for a given dataset $\mathcal{D}$ as $\ell_i(\Phi_{\boldsymbol{x}}; \mathcal{D}) = \frac{1}{2}\mathbb{E}_{t\sim\mathcal{D}}\left\|A^i\sin(t - \phi^i) - \Phi_{\boldsymbol{x}}(t)\right\|_2^2$. We then establish the outer function $f_i(\boldsymbol{x}) = \ell_i(\Phi_{\boldsymbol{x}}; \mathcal{D}_{\text{query}}^i)$ and inner function $g_\eta(\boldsymbol{x}) = \boldsymbol{x} - \alpha\nabla_{\boldsymbol{x}}\ell_i(\Phi_{\boldsymbol{x}}; \mathcal{D}_{\text{supp}}^i)$. As $f_i$ is smooth, our results are applicable.

In Figure 4, we show the results of BSGD and E-BSGD applied to this problem. In this experiment, the amplitude $A$ is drawn from a uniform distribution $\mathcal{U}(0.1, 5)$ and the phase $\phi$ is drawn from $\mathcal{U}(0, \pi)$. Both $\mathcal{D}_{\text{supp}}$ and $\mathcal{D}_{\text{query}}$ are independently drawn from $\mathcal{U}(-5, 5)$. The step size is set to

$\alpha = 0.01$. The batch size is fixed to 10. The performances of BSGD and E-BSGD are very close. This is not surprising because finetuning step size $\alpha$ is chosen to be small which significantly reduces the variance of $g_\eta$, making the bias of meta gradient to be very small ($\mathcal{O}(\alpha^2)$). Therefore, we observe similar performance of BSGD and E-BSGD. Similar trend also holds for BSpiderBoost and NestedVR compared to their extrapolated variants.

## F.2 Application of Deep Average Precision Maximization

The areas under precision-recall curve (AUPRC) has an unbiased point estimator that maximizes average precision (AP) [26, 34]. Let $\mathcal{S}_+$ and $\mathcal{S}_-$ be the set of positive and negative samples and $\mathcal{S} = \mathcal{S}_- \cup \mathcal{S}_+$. Let $h_{\boldsymbol{w}}(\cdot)$ be a classifier parameterized with $\boldsymbol{w}$ and $\ell$ be a surrogate function, such as logistic or sigmoid. A smooth surrogate objective for maximizing average precision can be formulated as [33]:

$$F(\boldsymbol{w}) = -\frac{1}{|\mathcal{S}_+|} \sum_{\boldsymbol{x}_i \in \mathcal{S}_+} \frac{\sum_{\boldsymbol{x} \in \mathcal{S}_+} \ell(h_{\boldsymbol{w}}(\boldsymbol{x}) - h_{\boldsymbol{w}}(\boldsymbol{x}_i))}{\sum_{\boldsymbol{x} \in \mathcal{S}} \ell(h_{\boldsymbol{w}}(\boldsymbol{x}) - h_{\boldsymbol{w}}(\boldsymbol{x}_i))}$$

This problem can be seen as a conditional stochastic optimization problem with $g_i(\boldsymbol{w}) = [\sum_{\boldsymbol{x} \in \mathcal{S}_+} \ell(h_{\boldsymbol{w}}(\boldsymbol{x}) - h_{\boldsymbol{w}}(\boldsymbol{x}_i)), \sum_{\boldsymbol{x} \in \mathcal{S}} \ell(h_{\boldsymbol{w}}(\boldsymbol{x}) - h_{\boldsymbol{w}}(\boldsymbol{x}_i))]$ and $f_i : \mathbb{R} \times \mathbb{R} \backslash \{0\} \to \mathbb{R}$ is defined as $f_i(\boldsymbol{y}) = -\frac{[\boldsymbol{y}]_1}{[\boldsymbol{y}]_2}$ where $[\boldsymbol{y}]_k$ denotes the $k$th coordinate of a vector $\boldsymbol{y} \in \mathbb{R} \times \mathbb{R} \backslash \{0\}$. During the stochastic optimization of this objective, we draw uniformly at random $\xi := \boldsymbol{x}_i$ (drawn from the set $\mathcal{S}_+$) as a positive sample and $\eta | \xi = [\mathcal{F}_{\boldsymbol{x}_1}, \mathcal{F}_{\boldsymbol{x}_2}]$ where set $\boldsymbol{x}_1$ is drawn uniformly at random from $\mathcal{S}_+$ and $\boldsymbol{x}_2$ is drawn uniformly at random from $\mathcal{S}$ and functional $\mathcal{F}_{\boldsymbol{x}}(\boldsymbol{w}) := \ell(h_w(\boldsymbol{x}) - h_w(\boldsymbol{x}_i))$. Note that $f_i \in \mathcal{C}^\infty$ is smooth with gradient

$$\nabla f_i(\boldsymbol{y}) = \begin{bmatrix} -\frac{1}{[\boldsymbol{y}]_2} \\ \frac{[\boldsymbol{y}]_1}{([\boldsymbol{y}]_2)^2} \end{bmatrix}.$$

Therefore, our results from Sections 3 and 4 again apply.

## F.3 Necessity of Additional Smoothness Conditions

Throughout the paper, we assume bounded moments (Assumption 1) and a smoothness condition (Assumption 2) to derive our extrapolation technique. However, it is worth noting that the technique itself does not explicitly depend on higher-order derivatives. Our theoretical framework does not address the behavior of extrapolation in the absence of these smoothness constraints. In this section, we investigate the application of extrapolation to two non-smooth functions:

- ReLU function given by $q(x) = \max\{x, 0\}$;
- Perturbed quadratics represented as $q(x) = x^2/2 + \text{TriangleWave(x)} + 1$. The function $\text{TriangleWave}(x)$ has a period of 2 and spans the range [-1,1], defined as:

$$\text{TriangleWave}(x) = 2 \left| 2 \left( \frac{x}{2} - \left\lfloor \frac{x}{2} + \frac{1}{2} \right\rfloor \right) \right| - 1$$

Visual representations of these functions can be found in Figure 5c. We set $s = 0$ and consider a random variable $\delta \sim \mathcal{N}(10, 100)$ with $m = 1$. We then apply first-, second-, and third-order extrapolation. The outcomes are depicted in Figure 5. Remarkably, both the ReLU and the perturbed quadratic functions do not conform to the differentiability assumptions inherent to our stochastic extrapolation schemes. Nonetheless, as indicated by Figure 5a and Figure 5b, our proposed second- and third-order extrapolation techniques yield a superior approximation of $q(\mathbb{E}[\delta])$.

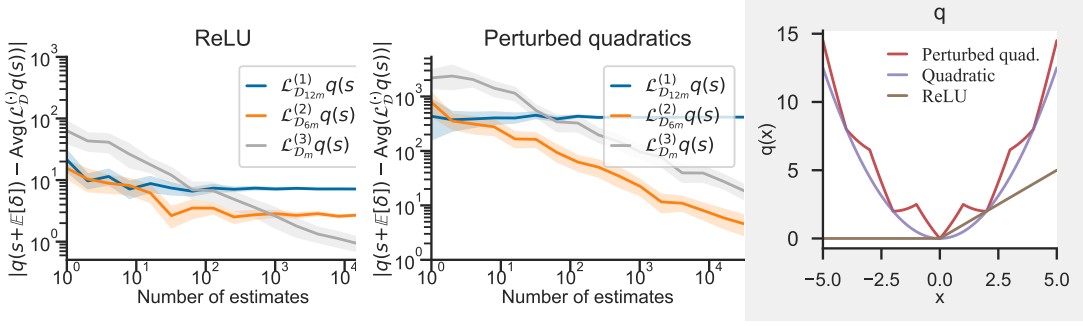

Figure 5a                     Figure 5b                     Figure 5c

Figure 5: (a) Fig. 5a: Error in estimating $q(s + \mathbb{E}[\delta])$ for our proposed first-, second-, and third-order extrapolation schemes applied to ReLU $q(x) = \max\{x, 0\}$, $s = 0$, $\delta \sim \mathcal{N}(10, 100)$, $m = 1$. (b) Fig 5b: Error in estimating $q(s + \mathbb{E}[\delta])$ for our proposed first-, second-, and third-order extrapolation schemes applied to a perturbed quadratic $q(x) = x^2/2 + \text{TriangleWave(x)} + 1$, $s = 0$, $\delta \sim \mathcal{N}(10, 100)$, $m = 1$. The TriangleWave($x$) has a period of 2 and spans the range [-1,1], i.e. $2|2\left(\frac{x}{2} - \lfloor \frac{x}{2} + \frac{1}{2} \rfloor\right)| - 1$. (c) Fig 5c: The ReLU and perturbed quadratic used in the Fig. 5a and 5b along with quadratic curves.

