# OpenReview forum: "Debiasing Conditional Stochastic Optimization"
_NeurIPS.cc/2023/Conference — NeurIPS 2023 poster_

### Official Review · Reviewer_B49C · 2023-07-06

**Soundness:** 3 good
**Presentation:** 2 fair
**Contribution:** 2 fair
**Rating:** 6
**Confidence:** 4

**Summary:**

This paper considers CSO problems and its finite-sum variant FCCO, and proposed new algorithms incorporating extrapolation and variance reduction. The proposed algorithms are shown to enjoy a better sample size in the inner layer which induces an improved sample complexity compared to existing CSO literature. The authors also furthered their study in the FCCO regime, the proposed algorithm enjoys a better complexity in moderately large $n$ regime.

**Strengths:**

1. Improved sample complexities for CSO and FCCO problems
2. The algorithm design incorporates extrapolation which is new to the CSO literature I think, and it brings an improvement in the sample complexity in the inner layer, which reveals some innovation.

**Weaknesses:**

1. The additional regularity assumption (compared to [15]) on the bounded higher-order gradients/moments, even though authors tried to rationalize it, reduces the significance of the result.
2. You mentioned "... $\Omega(\epsilon^{-3})$ is a sample complexity lower bound for standard stochastic nonconvex optimization [2]..." in Table 1, but as far as I can see, with your higher-order Lipschitz condition, the statement may not hold. Your function class is more restricted compared to the setting in [2]. I think authors may refer to  Could you please comment more on it?
   - Carmon, Yair, et al. "Lower bounds for finding stationary points ii: first-order methods." Mathematical Programming 185.1-2 (2021): 315-355.
   - Arjevani, Yossi, et al. "Second-order information in non-convex stochastic optimization: Power and limitations." Conference on Learning Theory. PMLR, 2020.
3. Some writing issues, I believe the paper's writing style could benefit from a few adjustments to enhance its readability, there .
   - Line 141, $T_m$ (vs. $T_{D_m}$)
   - Line 148, "lesser"
   - Line 173, "to (be) $\nabla$..."
   - Line 224, "... some time $t$...", I think such statements can be further revised and formalized.

With that, I think the paper provides some interesting results, especially in terms of the algorithm design, but I am not that convinced on the significance of the results (regarding the a bit unfair comparison) and feel that the writing can be further revised for a better flow.

**Questions:**

1. For clarification, with the extrapolation framework you need the constant term $s$, but when applied to CSO, you just set $s=0$, I am not sure why it is important to include the $s$ into your framework if there is no nontrivial instantiation. Could you please comment more on it?
2. For clarification, in Theorem 3, you set $m=O(\epsilon^{-0.5})$, but in Line 247, you mentioned the straightforward extension of E-BSpiderBoost requires a different $m$ by taking a maximum, I am wondering why the setting of $m$ in Theorem 3 does not directly apply here.

**Limitations:**

N/A.

---

> ### Author Rebuttal · Authors · 2023-08-09
>
> We thank the reviewer for careful proofreading and suggestions. We will incorporate the suggestions about writing. Please find below our responses to the specific questions and weaknesses that you mention.
>
>
> >**The additional regularity assumption (compared to [15]) on the bounded higher-order gradients/moments, even though authors tried to rationalize it, reduces the significance of the result.**
>
> We agree with the reviewer that it will be great to remove the regularity assumption. We would like to point out:
> 1) For all the CSO/FCCO examples considered by Hu et al. [15] (see https://arxiv.org/pdf/2002.10790.pdf) including invariant logistic regression, instrumental variable regression, first-order MAML for sine-wave few shot regression considered, our additional assumption is satisfied. Hence, our results are widely applicable and could still be considered significant.
> 2) Beyond CSO/FCCO rates, we also introduce a novel bias reduction technique, that we believe is of independent interest.
> 3) Our bias reduction technique seems to have promise even the regularity conditions fail (e.g., for non-smooth functions) as suggested by our attached experiments (see global response).  We leave it as future work to theoretically weaken the regularity assumption.
>
> >**You mentioned $\Omega(\varepsilon^{-3})$ is a sample complexity lower bound for standard stochastic nonconvex optimization [2]..." in Table 1, but as far as I can see, with your higher-order Lipschitz condition, the statement may not hold. Your function class is more restricted compared to the setting in [2].**
>
>
> We thank reviewer for these references. Your second reference [Arjevani et al., 2020] gives a highly non-trivial result that even when stochastic higher-order oracles are available, $\Omega(\epsilon^{-3})$ remains a lower bound in sample complexity for nonconvex optimization. Because of this, the $\Omega(\epsilon^{-3})$ lower bound result also applies to our setting with the restricted function class (note that stochastic optimization is just a special case of conditional stochastic optimization). This in contrast with the situation in the deterministic setting where Carmon et al. "Lower bounds for finding stationary points I", 2019, show that sample complexity upper bound improves with access to the $p$th-order oracles with Lipschitz derivatives to $\Theta(\epsilon^{-(1+1/p)})$. We will add these references in the main text.
>
>
> >**For clarification, with the extrapolation framework you need the constant term, but when applied to CSO, you just set $s=0$, I am not sure why it is important to include the into your framework if there is no nontrivial instantiation. Could you please comment more on it?**
>
> Yes, in Section 2, we introduce a constant $s$ to emphasize the generality of our scheme. We do use $s=0$ in the subsequent invocations, and would clarify this.
>
>
> >**For clarification, in Theorem 3, you set $m=O(\epsilon^{-0.5})$, but in Line 247, you mentioned the straightforward extension of E-BSpiderBoost requires a different  by taking a maximum $m$, I am wondering why the setting of  in Theorem 3 does not directly apply here.**
>
>
> FCCO is a special case of CSO where the outer random variable is discrete and takes value in a finite set. Compared to CSO, there is an extra hyperparameter $n$ (the number of outer samples), when $n$ is large, then we recover the choice of Theorem 3. But when $n$ is small, for example, there is only 1 outer sample, then there is better choice of $m$ than the choice from Theorem 3.

---

> > ### Comment · Reviewer_B49C · 2023-08-17
> >
> > Thank you for your reply, which basically addressed my problems. With the additional experiments, I will raise my score. Thank you.

---

### Official Review · Reviewer_STJD · 2023-07-07

**Soundness:** 4 excellent
**Presentation:** 3 good
**Contribution:** 3 good
**Rating:** 6
**Confidence:** 3

**Summary:**

This paper presents a stochastic extrapolation technique with variance reduction to obtain improved sample complexity bounds for conditional stochastic optimization problem. Numerical study also support the effectiveness of the proposed algorithm.

**Strengths:**

- The stochastic extrapolation is a useful tool for CSO problem. The technique is new for solving the CSO problem, though it requires higher-order smoothness assumption of the loss function. I think it could be potentially useful for other stochastic optimization formulations. I am also curious about how the authors come up with this approach? Do authors inspired from any existing literature?
- The overall presentation is very clear and the theoretical results with technique assumptions are presented in detail.
- Numerical study is very strong to support the effectiveness of the algorithm.

**Weaknesses:**

- From my perspective, some notations could be confusing. For example, in Theorem 2 the authors use $\Omega(\cdot)$ notation for batch size $m$ and number of iterations $T$, but in line 235 the authors use $\mathcal{O}$ instead. I would encourage the authors check those notations throughout the paper to make it consistent.
- Please check and revise the first sentence in line 287.
- In the theoretical analysis, the authors present iteration complexity together with sample compelxity per-iteration. However, in related work comparison, the authors only compare the overall total sample complexity of baseline methods. I encourage the authors also check and compare iteration complexity together with sample compelxity per-iteration.
- In Table 1, I think the authors should also add V-MLMC method (from [16]) for comparison. Also, in numerical study, I am confusing why the authors do not compare with RT-MLMC method but only V-MLMC method?

**Questions:**

N/A

---

> ### Author Rebuttal · Authors · 2023-08-09
>
> We thank the reviewer for careful proofreading and suggestions. We will add the iteration complexities in the revision.
>
> >**I am also curious about how the authors come up with this approach? Do authors inspired from any existing literature?**
>
> We independently came up with this stochastic extrapolation approach. The closest to our approach in statistics is the idea of sample splitting. Generally, the techniques in statistics are designed to be applicable to certain statistical models, while our approach seems more general. Besides, compared to previous bias correction approaches, ours is significantly different in terms of both setup and analysis. We have briefly discussed these differences in Appendix B. We fully agree with the reviewer that these techniques could be potentially used in other stochastic optimization problems.
>
> >**why the authors do not compare with RT-MLMC method but only V-MLMC method?**
>
> According to analysis in the previous work [16], when applied to CSO problem, V-MLMC and RT-MLMC enjoy same sample complexity (Table 3 in [16]). Experimentally too, [16] observed V-MLMC behaves better than RT-MLMC with a smaller variance, therefore, we choose to compare against the more robust V-MLMC approach. However, we will be happy to add RT-MLMC as a baseline.

---

> > ### Comment · Reviewer_STJD · 2023-08-19
> > **After reading rebuttal**
> >
> > Thank you for your reply. I will raise my score to 6 because of the efforts made in rebuttal

---

### Official Review · Reviewer_me2Y · 2023-07-13

**Soundness:** 3 good
**Presentation:** 3 good
**Contribution:** 3 good
**Rating:** 6
**Confidence:** 2

**Summary:**

The paper highlights that optimizing conditional stochastic optimization problems is challenging as the sample-averaged gradient is biased. The biased gradient increases the sample complexity needed to achieve convergence. To account for the bias, the paper proposes stochastic extrapolation. The method shows that the bias can be expressed as a sum of Taylor series expansion terms and the bias can be estimated by constructing estimates for each of the terms in the expansion. The key insight is that the size of the $k^{th}$ order term of the series is of size $O(1/m^k)$ where $m$ is the batch size. They propose the $k^{th}$ order extrapolation operator which computes up to the $k^{th}$ order term in the Taylor series and uses the terms to correct for the bias. The paper then uses the the stochastic extrapolation technique to bias correct the gradients for the conditional stochastic optimization problem and finite-sum coupled compositional optimization problem. They then prove convergence guarantees for optimization algorithms applied on the two problem types, showing stochastic extrapolation improves the sample complexity of the algorithms. Finally, they perform a numeric experiment on invariant logistic regression and show the extrapolated version of the algorithms achieves lower sub-optimality.


**Strengths:**

The paper proposes a fairly novel approach to address the issue of biased sub-gradients. The problem and approach are described clearly and intuitively. The paper is also able to highlight some important optimization problems that satisfy the regularity conditions leveraged by the paper. The paper’s significance is being able to leverage these regularity assumptions in order to prove stronger sample complexity bounds.

**Weaknesses:**

The paper is hard to read since authors i) often reference the appendix with limited context, ii) do not point out if the reference is in the appendix or not, and iii) do not define notation.
	- As an example for the first point, Lemma 1, Lemma 2, and Theorem 7 are all in the appendix but it is not noted.
	- For the second point, Lemma 2 also is referenced in line 155 to quantify the variance of the operator applied to the function, but it’s unclear what Lemma 2 shows precisely. In most cases, it seems that the authors are trying to make a statement and reference the proof, however, the language is not clear that is the case. As evidence, when referencing Lemma 2, the authors state “a consequence of Lemma 2 is that” when it could be more clear by stating “Lemma 2 (in Appendix Section B) proves”.
	-For the third point, it is hard to find a definition for V in Lemma 2.
The above three points make reading and verifying the statements in the paper challenging.

The paper has minimal numeric studies. For example, it would be interesting to see how the approach performs for problems that do not satisfy the regularity conditions. It may also be interesting to see the improvement as the level of bias varies due to underlying problem parameters.

**Questions:**

1. The paper mentions a burn-in rate for higher order extrapolation. Is there any intuition of controls the burn-in rate or if the burn-in rate is theoretically quantifiable?
2. Does the stochastic extrapolation perform better or worse when the number of samples is small? The numerics seem to indicate the performance is the same for BSGD and BSpiderBoost when the # of samples is small.

**Limitations:**

From a technical perspective, the authors do note that their proposed approach requires the target optimization problem to satisfy certain regularity conditions. However, it is unclear what is the cost of applying the stochastic extrapolation and if there is any trade-offs.

---

> ### Author Rebuttal · Authors · 2023-08-09
>
> Thank you for your positive feedback. We also thank the reviewer for the suggestions on the writing and readability. We will incorporate the suggestions accordingly. Please find below our responses to the specific questions and weaknesses that you mention.
>
>
> >**The paper has minimal numeric studies. For example, it would be interesting to see how the approach performs for problems that do not satisfy the regularity conditions. It may also be interesting to see the improvement as the level of bias varies due to underlying problem parameters.**
>
> We have provided a pdf in the rebuttal that explore other dimensions of the extrapolation as suggested by the reviewer. In global response, we discuss the case where the differentiability assumption fails. We now discuss Fig. 6 in the attached pdf where we vary the level of bias.
>
> **Description of Attached Fig. 6**: We vary the magnitude of the second order moment ($\sigma_2$) of the random variable. The function being estimated is $q(x)=x^2/2$. In this setting, both second- and third-order extrapolation schemes yield unbiased estimates. On the other hand, the bias of first-order method grows with $\sigma_2$ according to Proposition 2 in the Appendix C. The experiment results confirm that with increasing second moments, our proposed second- and third-order extrapolation schemes still enjoy the same convergence rate.
>
> > **burn-in rate for higher order extrapolation**
>
> The burn-in phase depends on the constants in the problem (i.e., smoothness constants ($a_k$) and magnitude of moments ($\sigma_k$)). If resources permit, we can estimate these constants to roughly draw the burn-in region and rate.
>
> >**numerics seem to indicate the performance is the same for BSGD and BSpiderBoost when the # of samples is small.**
>
> This is also empirically observed in the previous work [15]. The BSpiderBoost is much harder to tune in practice, and for some tasks the advantage of BSpiderBoost over BSGD can be marginal. So in this paper, we pay more attention to the improvements of extrapolation rather non-extrapolation version (e.g., BSGD vs. E-BSGD, BSpiderBoost vs. E-BSpiderBoost).

---

> > ### Comment · Reviewer_me2Y · 2023-08-20
> > **Thank you for your response**
> >
> > I appreciate the response and additional experimental results. I will keep my score as I think it definitely falls within an acceptance category.

---

### Official Review · Reviewer_eDyu · 2023-07-26

**Soundness:** 3 good
**Presentation:** 3 good
**Contribution:** 3 good
**Rating:** 6
**Confidence:** 3

**Summary:**

The authors study stochastic optimization problems with nested conditional expectations, with access to first-order samples from the involved functions. With such an oracle access, it is non-trivial to build unbiased estimates of the gradient, hence normal stochastic optimization methods don't work.

They authors show that under a third-order differentiability assumption, which they argue to be "mild", one could adapt existing extrapolation techniques to this problem and obtain an estimate of the gradient with a reduced bias. This in turn results in substantial improvement in the convergence rate of algorithm fed with this bias-reduced estimate. The authors demonstrate this using state-of-the-art algorithms, both with and without finite-sum variance reduction.

**Strengths:**

The claimed improvements are substantial. The paper is modular, in the sense that there is one clear goal: reducing the bias of the gradient estimate, and that is carried out separately. Then, the result is fed into existing best algorithms. This makes the contribution very clear.

**Weaknesses:**

There seems to be some issue in the last term of Eq (2): since $\phi_\delta$ depends on $\delta$, we cannot take it outside expectation, neither can we decompose the resulting expectation in such a way that $E[(\delta - h)^{2k-1}]$ comes out and multiplies the expectation of the term involving $\phi_\delta$. I am not sure if this will have a major adverse effect on the rest of the results in this section, but I will keep looking deeper in the proofs.

In terms of related work, I am surprised that the use of extrapolation for bias reduction in Stochastic Composite Optimization is missed. The central work here is:
M. Wang and J. Liu. Accelerating Stochastic Composition Optimization. 2016
The authors need to discuss the relationship between the techniques.

I also find it problematic that the authors call the assumption of third-order differentiability a "mild assumption". Whether this is mild or not has to be left to the reader to decide. Currently, the introduction and Table 1 give an incorrect impression. This needs to be made clear way earlier in the introduction, and also laid out clearly in the caption of Table 1. Also, the assumption $n = O(\epsilon^{-2})$ is certainly worth more explanation that the footnote given.

**Questions:**

Please add any comments you may have about the points raised under "Weaknesses", including the relationship to the work on accelerated stochastic composition optimization, as well as the issue with Eq (2).

**Limitations:**

The paper can be improved in terms of clarifying the limitations of its results, as explained under "Weaknesses" above.

This is a theoretical paper. As such, greater societal impact is not addressable at this level.

---

> ### Author Rebuttal · Authors · 2023-08-09
>
> Thank you very much for your encouraging comments and valuable feedback. Please find below our responses to the specific weaknesses that you mention.
>
> >**last term of Eq (2)**
>
> The reviewer is correct, there is a typo in Eq. (2), $\phi_\delta$ should be inside the expectation, and we will fix this. To clarify, the idea behind our extrapolation scheme is that  we cancel out the lower order terms and keep the remainder small. The lower order terms cancel out, while the remainder is upper bounded by the uniform bound on $q^{(\cdot)}$ and the moments of $\delta$. By construction, the moments decrease with increasing $m$ of $\delta$, c.f., Lemma 1 in Appendix C.
>
> >**Reference: M. Wang and J. Liu. Accelerating Stochastic Composition Optimization. 2016**
>
> This is a very interesting paper, and we thank the reviewer for pointing it out. The problem they considered is the same as considered by Zhang et al. "Multilevel composite stochastic optimization via nested variance reduction" [38]. Both of them try to estimate the inner stochastic function $g_\eta(x)$ with no bias. This paper use extrapolation to estimate $g_\eta(x)$ with an iteration complexity of $O(T^{-4/9})$ while Zhang et al. use variance reduction and yield a better $O(T^{-1/2})$. However, both methods cannot be applied to conditional stochastic optimization, considered here, because the inner variable $\eta$ depends on the outer, which makes their estimation biased.
>
> Regarding the similarities and differences between our extrapolation technique and [Wang and Liu, 2016].
> - Similarities: Both these papers use affine combinations in the construction where some coefficients are negative. That is why both methods are termed extrapolation.
> - Differences:
>     - Wang and Liu apply extrapolation to iterates to help estimate the inner function, while we apply extrapolation to the outer function to reduce the bias in CSO problem.
>     - Our affine combination coefficients are constants while their coefficients depend on iteration (e.g. $\beta_k=2k^{-b}$ in their Theorem 1).
>     - Different motivations behind picking the affine combination coefficients. We choose coefficients based on the motivation to cancel out lower order terms in Taylor series while they don't.
>     - As explained in Lines 160-164, our extrapolation technique naturally generalizes to higher orders, while [Wang and Liu, 2016] does not.
>
> We will add this to our discussion in Appendix B.
>
> >**problematic that the authors call the assumption of third-order differentiability a "mild assumption".**
>
> We meant to say that even with this assumption we still cover lots of known application of CSO/FCCO problems appeared in the closely related CSO/FCCO works [15,31], including invariant logistic regression, instrumental variable regression, first-order MAML for sine-wave few-shot regression task, deep average precision maximization etc. We will rephrase this discussion.

---

> > ### Comment · Reviewer_eDyu · 2023-08-16
> > **Thank you for your response**
> >
> > Thank you for your response. I have read the other reviews and the rebuttals. The response addresses my question about the work of Wang and Liu [2016]. I am still assessing the paper in terms of the proof and in light of the other reviews. I will post any concerns here.

---

### Official Review · Reviewer_VftD · 2023-07-27

**Soundness:** 2 fair
**Presentation:** 3 good
**Contribution:** 2 fair
**Rating:** 6
**Confidence:** 3

**Summary:**

The paper addresses the conditional stochastic optimization (CSO) problem. A noted drawback in CSO is the bias in its sample-averaged gradient, an offshoot of its nested structure, which necessitates an extensive sample complexity to achieve convergence.
The authors introduces a new stochastic extrapolation technique address this bias effectively. They propose that by merging this extrapolation with variance reduction techniques, improvements in sample complexity can be realized compared to existing results for non-convex smooth objectives. The authors claim that the proposed methods improve the sample complexities for both the CSO and FCCO under some reguarlity.

**Strengths:**

- Overall writing is clear, and organization is well-structured.
- The paper introduces an extrapolation-based scheme to alleviate bias in gradient estimations. The proposed method is used for gradient estimations for CSO and FCCO problems.
- For CSO problem, the authors claim that the proposed methods, E-BSGD and E-BSpiderBoost, achieve an improved sample complexity compared to the existing bounds under some additional reguarlity.
For FCCO problem, the authors proposed an algorithm, E-NestedVR, combining the extrapolation-based scheme with a multi-level variance reduction which achieves an improved (or at least matching) sample complexity compared to the existing results.
- The techniques of the anlysis are interesting, and worth credit.

**Weaknesses:**

- While the new bounds show improved rates, the results are based on additional regularity assumptions. The authors acknowledge the discrepancy in the set of assumptions used in the current paper and those in the existing results. They state that "the bounds derived from previous studies do not improve when incorporating this additional regularity assumption" of the 3rd-order differentiability, and hence argue that their results present an improved bound.
However, I am not sure whether such an assertion is valid. The new regularity condition imposed by this paper is used for the authors' technical convenience, not required by the existing methods. Hence, it is very possible and often obvious that the existing methods do not improve even with the new additional regularity assumption because these methods did not even need such regularity.
- So I am not sure whether the claim of the "improved" rate is valid. Aren't they rather apples and oranges? For example, can you precisely pinpoint where Assumtpino 6 is needed? What rate (just a ballpark) would you get if you do not have this assumption?

**Questions:**

Please see the comment in the weakness section.

**Limitations:**

Please see the comment in the weakness section.

---

> ### Author Rebuttal · Authors · 2023-08-09
>
> Thank you for appreciating our work. Please find below our responses to the specific weaknesses that you mention.
>
> >**While the new bounds show improved rates, the results are based on additional regularity assumptions ...**
>
> We agree that the regularity assumptions are for technical convenience and apologize if our phrasing was unclear. This regularity condition allows us to use Taylor expansion to derive extrapolation schemes, but none of our algorithms (E-BSGD, E-BSpiderBoost, and E-NestedVR) explicitly use the high order derivatives.
>
> In Lines 88-89, we only wanted to clarify that if we add our additional 3rd-order differentiability on $\nabla f_\xi$ assumption to existing methods, they do not automatically yield better rates. We will rephrase the discussion here to be clear. Btw, one of the reasons why we brought this up is because we conjecture that with even stronger differentiability assumption, our bounds automatically improve. The reason here being that then we can construct higher-order (> 2) extrapolation operators and use them in our proposed algorithms. So this might be a case where stronger assumptions directly improve the results.
>
> >**So I am not sure whether the claim of the "improved" rate is valid. Aren't they rather apples and oranges?**
>
> We again agree with the reviewer that because of additional assumption, a fair comparison is hard. However, we would like to point out two things:
> 1) As we point out in the paper for many common CSO/FCCO problems our additional assumption is satisfied. This includes invariant logistic regression, instrumental variable regression, first-order MAML for sine-wave few shot regression considered by Hu et al. [15] (see https://arxiv.org/pdf/2002.10790.pdf) while proposing BSGD/BSpiderBoost, and similarly, deep average precision maximization considered by Wang et al. [31] while proposing SOX. So in these cases there is a direct improvement.
> 2) Beyond CSO/FCCO rates, we also introduce a novel bias reduction technique, that we believe is of independent interest.
>
> >**For example, can you precisely pinpoint where Assumtpino 6 is needed? What rate (just a ballpark) would you get if you do not have this assumption?..**
>
> Throughout our analysis, we provide bias bounds (e.g., Lemma 6)  for various algorithms, which use Corollary 1 which in turn is built on Proposition 1 under the Assumption 6. Now, as an extension, we give an experimental result (see the global response and attached pdf) that might indicate that our proposed stochastic extrapolation technique also corrects biases for even non-smooth objectives. We leave it as future work to theoretically weaken the regularity assumption.

---

> > ### Comment · Reviewer_VftD · 2023-08-19
> > **Thanks for the responses.**
> >
> > I have read the authors' responses and other reviews. I think I have a better understanding of the paper. Thanks for the responses.

---

### Author Rebuttal · Authors · 2023-08-09

We thank all the reviewers for constructive and actionable feedback on the paper. We have additional experimental results, in particular, addressing the question of what happens if the regularity  assumption gets violated.


**Description of Fig. 5 in the attached pdf**: We consider an experimental  setting similar to Fig. 1, but where the differentiability assumption is violated. The goal again, for a fixed $s$, is to draw samples of $q(s+\delta)$ to estimate $q(s+\mathbb{E}[\delta])$.


1. Relu function ($q(x)=\max(x,0)$). This function is Lipschitz continuous but does not satisfy the smoothness condition as it is not differentiable at $0$. The Fig. 5a shows that with more estimates, both second- and third-order extrapolation outperform first-order extrapolation.
2. Quadratic function perturbed by triangle waveform ($q(x)=x^2/2+$TriangleWave$(x)+1$). The resulting function $q$ is non-convex and Lipschitz continuous. As noticed in Fig. 5b, despite the fact that this function violates the differentiability assumption, applying extrapolation to this problem still exhibits similar curve as the quadratic one (Fig. 1a in the paper).

In Fig. 5c, we plot the above considered $q$ functions (Relu, perturbed quadratic).

We gave a possible explanation to the above observations without rigorous mathematical proof. Given a non-smooth function $q$, if there exists a smooth function (say $\tilde{q}$) which is close to $q$ with an approximation error much smaller than the magnitude of bias, then the bias of higher-order extrapolation applied to $q$ can be decomposed into the bias of applying higher-order extrapolation applied to $\tilde{q}$ plus approximation errors. As the bias dominates approximation error, the bias correction on non-smooth $q$ look similar to that on smooth $\tilde{q}$.

In our paper, the differentiability assumption is only used to ensure the existence of Taylor series, which motivates the extrapolation scheme. But higher order derivatives are not explicitly used in the extrapolation. The above experiments indicate that the Assumption 6 may be further relaxed while extrapolation still work.  We will add these experimental results to the paper. We leave the formalization of this argument and relaxation of Assumption 6 as future work.

---

### Decision · Program_Chairs · 2023-09-21

**Decision:**

Accept (poster)

**Comment:**

This paper proposes stochastic extrapolation schemes to address bias in gradient estimates in conditional stochastic optimization. The paper makes meaningful theoretical contributions by assuming smoothness conditions. Reviewers raised concerns about the clarity of exposition, especially regarding the mathematical framework. I recommend carefully addressing the comments on better contextualizing the additional assumptions used in the paper, and discussing the limitations of the framework more.